

# Estimating time in quantum chaotic systems and black holes

Haifeng Tang⋆, Shreya Vardhan† and Jinzhao Wang‡

Stanford Institute for Theoretical Physics, Stanford University, Stanford, CA 94305, USA

⋆ hftang@stanford.edu , † shreyavar08@gmail.com , ‡ jinzhao@stanford.edu

## Abstract

We characterize new universal features of the dynamics of chaotic quantum many-body systems, by considering a hypothetical task of "time estimation". Most macroscopic observables in a chaotic system equilibrate to nearly constant late-time values. Intuitively, it should become increasingly difficult to estimate the precise value of time by making measurements on the state. We use a quantity called the Fisher information from quantum metrology to quantify the minimum uncertainty in estimating time. Due to unitarity, the uncertainty in the time estimate does not grow with time if we have access to optimal measurements on the full system. Restricting the measurements to act on a small subsystem or to have low computational complexity leads to results expected from equilibration, where the time uncertainty becomes large at late times. With optimal measurements on a subsystem larger than half of the system, we regain the ability to estimate the time very precisely, even at late times. Hawking's calculation for the reduced density matrix of the black hole radiation in semiclassical gravity contradicts our general predictions for unitary quantum chaotic systems. Hawking's state always has a large uncertainty for attempts to estimate the time using the radiation, whereas our general results imply that the uncertainty should become small after the Page time. This gives a new version of the black hole information loss paradox in terms of the time estimation task. By restricting to simple measurements on the radiation, the time uncertainty becomes large. This indicates from a new perspective that the observations of computationally bounded agents are consistent with the semiclassical effective description of gravity.


# 1  Introduction

At late times in chaotic quantum many-body systems, most simple macroscopic observables relax to steady-state values, which are constant with time up to small fluctuations. Intuitively, from the perspective of such observables, the evolution of the state $|\psi(t)\rangle = e^{-iHt}|\psi_0\rangle$ by the chaotic Hamiltonian $H$ from any initial state $|\psi_0\rangle$ slows down with time. This intuition is captured by the fact that the late-time state $|\psi(t)\rangle$ macroscopically resembles an equilibrium density matrix $\rho^{(\text{eq})}$, which commutes with the Hamiltonian and does not evolve at all. In this paper, we will explore the extent to which the state $|\psi(t)\rangle$ evolves with $t$ using a hypothetical task of "time estimation." We imagine that we are given an $\mathcal{O}(1)$ number of copies of $|\psi(t)\rangle$, but do not know the precise value of $t$ and want to estimate it using measurements on the state. How effectively can we perform this task?

    The setup of this thought experiment is inspired by the general task of parameter estimation in quantum metrology [1, 2]. We use a quantity called the Fisher information [3–5] to quantify the minimum uncertainty in the time estimate. Studying the behaviour of the Fisher information for time estimation will allow us to identify certain universal features of thermalization beyond those captured by standard observables including correlation functions and

entanglement entropy. This approach contributes a new perspective to recent discussions on characterizing various fine-grained aspects of the structure of thermalizing quantum states, such as complexity growth [6], deep thermalization [7], subsystem entropy fluctuations [8], and pseudoentanglement [9,10]. Further, by treating black holes as examples of chaotic systems, this quantity will allow us to identify a new version of Hawking's information loss paradox, and to make new predictions based on unitarity for the black hole evaporation process.

The Fisher information we will study can be seen as an intrinsic velocity associated with changes in the state with time. The minimum uncertainty $(\delta t)^2$ of an attempted time estimate is inversely proportional to the Fisher information. Intuition from thermalization suggests that the Fisher information should decay and go to zero at late times, corresponding to a large uncertainty of time estimation. However, there is a competing intuition from unitarity that $|\psi(t)\rangle$ cannot stop evolving with time at a fundamental microscopic level. We will see that the latter intuition is captured by the fact that an optimal version of the Fisher information, known as the quantum Fisher information (QFI), is a constant with time if we have access to the full system. This constant value is proportional to the energy variance of the state. Hence, the full microscopic state $|\psi(t)\rangle$ evolves just as fast at any later time as it does initially.

With optimal measurements on the full system, we therefore always retain the ability to estimate time accurately if we could do so at early times. We consider two natural kinds of sub-optimal measurements. The uncertainty in time on considering optimal measurements on a subsystem is quantified by the *subsystem quantum Fisher information $F_A(t)$*. On the other hand, by restricting the *computational complexity* of measurements, either on the full system or on a subsystem, we naturally arrive at another coarse-grained Fisher information called the *classical Fisher information* in the computational basis, $f_A^{\text{comp}}(t)$. $F_A(t)$ for a subsystem smaller than half of the system, and $f_A^{\text{comp}}(t)$ even for more than half of the system, will both turn out to reproduce expectations from thermalization. We will define these quantities and introduce the task of time estimation more explicitly in Sec. 2.

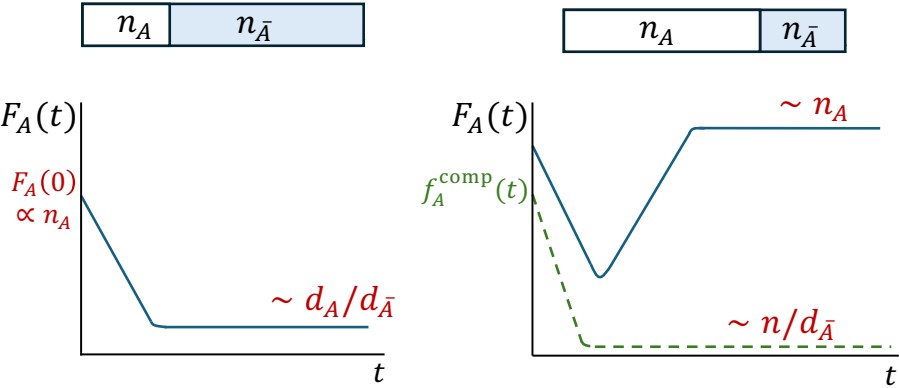

Figure 1: We consider the evolution of the subsystem Fisher information associated with time estimation in a chaotic quantum many-body system. We divide the system into two parts $A$ and $\bar{A}$, and $n_A$, $d_A$ respectively denote the number of degrees of freedom and Hilbert space dimension of $A$. For $n_A < n_{\bar{A}}$ (left), the subsystem quantum Fisher information $F_A(t)$ decays monotonically from an extensive value to an exponentially small value at late times, consistent with the expectation from thermalization. For $n_A > n_{\bar{A}}$ (right, blue curve), $F_A(t)$ shows a surprising non-monotonic evolution and saturates to an extensive value. On restricting to simple measurements, the associated classical Fisher information $f_A^{\text{comp}}(t)$ decays monotonically and becomes small even for $n_A > n/2$ (right, green dashed curve).

We summarize our main results in the schematic plots in Fig. 1. These results apply to pure initial states with extensive energy variance, which also have extensive initial values of $F_A$ and $f_A^{\text{comp}}$. Based on numerical results in spin chains, analytic results in random pure states, and other numerical as well as analytic toy models, we expect that these behaviours should be universal for unitary chaotic quantum many-body systems with local interactions.

Let $n_A, n_{\bar{A}}$ denote the number of degrees of freedom in a subsystem $A$ and its complement $\bar{A}$. The full system size is $n$. We find two striking differences between the subsystem quantum Fisher information $F_A(t)$ between the cases $n_A < n_{\bar{A}}$ and $n_A > n_{\bar{A}}$ as shown in Fig. 1, which are consequences of the interplay between thermalization and unitarity:

1. The saturation value of $F_A(t)$ is exponentially suppressed in $n$ for $n_A < n_{\bar{A}}$, and proportional to $n$ for $n_A > n_{\bar{A}}$. This transition can be understood analytically by using a Haar-random pure state as a toy model for the late-time state, and is also observed numerically in chaotic spin chain models. Like previous transitions at $n_A = n/2$ found by Page [11] and Hayden and Preskill [12], this transition indicates a qualitative difference between $\text{Tr}_{\bar{A}}[|\psi(t)\rangle\langle\psi(t)|]$ and the equilibrium density matrix $\text{Tr}_{\bar{A}}[\rho^{(\text{eq})}]$ when we consider a subsystem larger than half of the system. The new transition indicates a difference in the *speed of evolution* of the two states, rather than in their *information content* which is captured by entanglement entropy.

2. For $n_A < n_{\bar{A}}$, $F_A(t)$ shows a monotonic decay with time, consistent with the physical expectation that the time-evolution is increasingly slowing down. For $n_A > n_{\bar{A}}$, $F_A(t)$ shows a non-monotonic time-evolution.[1] We will understand the increasing behaviour at intermediate times, which is counterintuitive from the perspective of thermalization, as follows. $\rho_A(t)$ is not full-rank for $n_A > n_{\bar{A}}$, and its support rotates within the full Hilbert space of $A$. As the state gains access to more and more of the Hilbert space, its speed of rotation increases, until it saturates to a universal late-time value which gives the dominant contribution to $F_A(t)$. See Fig. 2.

   Note that more conventional probes of thermalization such as entanglement entropy are insensitive to any properties of the eigenstates of the reduced density matrices and cannot capture this physical phenomenon.

   The main evidence for these behaviors at intermediate times comes from numerical results in chaotic spin chains, and is further supported by analytic calculations in a toy model for the time-evolving state based on the Brownian GUE model.

We also study the behavior of $F_A(t)$ in a free-fermion integrable system and find a remarkably different behaviour, indicating that this quantity is a sharp probe of the transition from free-fermion integrability to chaos. In interacting integrable systems, the behavior of $F_A(t)$ is qualitatively similar to that in chaotic systems.

In a chaotic system, even for $n_A > n_{\bar{A}}$, we find that the Fisher information $f_A^{\text{comp}}$ associated with simple measurements shows a monotonic decay and saturates to an exponentially small value in $n$. This is true as long as $n_A < n - \mathcal{O}(\log n)$, after which point even $f_A^{\text{comp}}$ is no longer small.

Let us now turn to the implications of these results for black holes, by treating them as examples of highly chaotic quantum many-body systems. The process of formation of a black hole from a star corresponds to the approach to equilibrium in general chaotic systems. Hawking [13, 14] found that black holes subsequently emit thermal radiation and evaporate. However, Hawking's calculation of the state of the black hole radiation in semiclassical gravity leads to a contradiction with the above general predictions for unitary evolutions. This can be seen

---

[1]The decay is monotonic up to small fluctuations which go away on averaging over initial states or small time intervals.

as a new version of the black hole information paradox, now relating to the time-evolution properties of the state rather than its entropy. The task we consider in this setting is to estimate the time that has elapsed during the evaporation process by making measurements on the radiation emitted by the black hole, and we want to understand the minimum uncertainty $\delta t$ in making this estimate.

In Hawking's calculation, the black hole evaporation process is modeled by a considering a sequence of Schwarzschild black hole solutions in 3+1 dimensions, where the mass $M$ of the black hole gradually decreases due to the emission of radiation. The spectrum of the radiation emitted at a given stage in the evaporation process, where the black hole has mass $M$, approximately matches the Planck spectrum for black-body radiation at temperature

$$T_H = 1/(8\pi G_N M). \tag{1}$$

In this calculation, the state $\rho_R$ of the radiation does not change over times much shorter than the time scale $t_{\text{evap}} \equiv G_N M$. To see this, note that the total number of particles in the radiation changes with time as $\frac{dN}{dt} \sim \frac{1}{t_{\text{evap}}}$ (see [15] for an explicit calculation of this emission rate), so that the average time between the emission of two particles into the radiation is $t_{\text{evap}}$. Hence, the semiclassical gravity calculation predicts a minimum uncertainty of $\delta t \sim t_{\text{evap}}$ in estimating the time by detecting the emission of new particles.

If we consider the fundamental quantum description of the black hole as opposed to its semiclassical description, then based on similar reasoning to earlier works of Page [11,16] and Hayden and Preskill [12], we expect that the universal behaviours that we find for the quantum Fisher information in chaotic quantum many-body systems should also apply to black holes. This will turn out to imply that in the fundamental description, Hawking's result $\delta t \sim t_{\text{evap}}$ cannot be true after the Page time, when the effective Hilbert space dimension of the radiation becomes larger than that of the black hole. Instead, we predict that the subsystem QFI of the radiation should become $\mathcal{O}(1/G_N)$ after the Page time in the fundamental description of the black hole, which implies that $\delta t \sim \sqrt{G_N}$ using optimal measurements on the radiation after the Page time. In particular, this uncertainty is much smaller than $t_{\text{evap}}$, as long as the black hole mass is much larger than the Planck mass ($M \gg 1/\sqrt{G_N}$). It would be interesting to see whether and how our predictions for the subsystem QFI can be checked by including corrections to semiclassical gravity, such as those that were recently used to obtain the Page curve in [17–20].

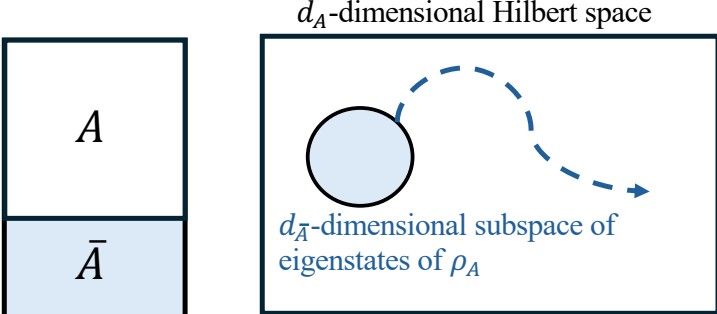

Figure 2: For a pure state $|\psi(t)\rangle$ in a chaotic system, for a subsystem $A$ larger than half of the system, the QFI of the reduced density matrix $\rho_A(t)$ has a large universal late-time value. In this case, $\rho_A(t)$ is not full-rank. The late-time value of the QFI captures the speed of rotation of the support of $\rho_A(t)$ within the full Hilbert space of $A$.

The optimal measurements on the radiation that lead to a small uncertainty of the time estimate after the Page time are likely to be highly complex. There is a growing body of evidence in the literature that observations made by computationally bounded observers with access to the radiation are consistent with Hawking's semiclassical calculation for the reduced density matrix of the radiation [21–26]. Our results for the computational basis classical Fisher information $f_A^{\text{comp}}$ provide yet another perspective that confirms these ideas. We predict that the value of this quantity is $\mathcal{O}(e^{-1/G_N})$ at all but very late times in the evaporation process. Hence, it is effectively indistinguishable from the result of zero Fisher information from Hawking's state.

The plan of the paper is as follows. In Sec. 2, we explain the setup of the time estimation task, introduce the relevant quantities, and discuss general constraints on them from unitary Hamiltonian evolution. In Sec. 3, we discuss results for the quantum Fisher information associated with subsystems using a variety of models for chaotic systems, as well as certain examples of integrable systems. In Sec. 4, we discuss the classical Fisher information for simple measurements in the computational basis. In Sec. 5, we carry out an explicit numerical simulation where we estimate the value of time by making measurements on the state, using a method called the maximum likelihood estimate. In addition to being a fun experiment, this clarifies a conceptual question about whether it is possible to come up with a global (as opposed to merely local) estimate of time when the Fisher information is large. In Sec. 6, we phrase our question about time estimation in the language of quantum error-correction in order to make an explicit comparison between the task we consider here and the Hayden-Preskill protocol. Most technical details are presented in the Appendices.

We expect the results to be of equal interest from the perspective of quantum many-body physics and quantum gravity. The sections most relevant in the context of black hole physics are Sec. 3.3 on results in random pure states, Sec. 3.4 about the implications of these results for black holes, and the discussions in Sec. 4 and Sec. 5 on the complexity of distinguishing the time-evolved state from the thermal state (or equivalently, distinguishing the fundamental description of the black hole from Hawking's description).

*Relation to previous work:* The QFI has previously been applied to quantum many-body systems in the context of dynamical susceptibilities and multipartite entanglement [27–29]. In these setups, the parameter to be estimated is associated with deformations of the state by operators other than the Hamiltonian. The physical interpretation of the QFI in such cases is different from the one in this paper. The QFI associated with time estimation was previously considered in [30], with a different goal of characterizing time-energy uncertainty relations. In the context of classical stochastic dynamics, the Fisher information associated with time estimation was studied in [31]. The models in [31] can be seen as coarse-grained, classical descriptions of thermalization, which we will contrast with microscopic quantum-mechanical descriptions. In [32], a particular case of the time estimation task, using simple measurements on the full system, was studied in terms of a certain mutual information which probes changes in the state over long intervals of time and captures a different regime from the Fisher information (which quantifies changes over infinitesimal time intervals).

## 2 Setup and constraints from unitarity

In the setup of our thought experiment, we will consider two kinds of one-parameter families of states, labelled by the parameter $t$: either the time-evolved state on the full system, $\sigma(t) = e^{-iHt}\sigma_0 e^{iHt}$ for some initial state $\sigma_0$, or its reduced density matrix on some subsystem, $\sigma_A(t) = \text{Tr}_{\bar{A}}[e^{-iHt}\sigma_0 e^{iHt}]$, where $\bar{A}$ is the complement of $A$. Let us denote these two cases by the common notation $\rho(t)$. Suppose we are experimentalists who are given $N$ copies of $\rho(t_0)$,

at a particular value $t_0$ of $t$. We know the initial state $\sigma_0$ and $H$, but do not know $t_0$, and want to estimate its value by making measurements on our $N$ copies.

Consider some choice of complete projective measurement, with measurement outcomes labelled by $|\xi\rangle$. From $N$ independent measurements on $\rho(t_0)$, we obtain measurement outcomes $\xi_1, \dots, \xi_N$. Suppose we come up with an estimate $T_{\text{est}}(\xi_1, \dots, \xi_N)$ for $t_0$ based on these measurement outcomes. Our key question is the following: how well can the best possible function $T_{\text{est}}$ do at estimating $t_0$? If no estimator is able to do a good job, this indicates that $\rho(t)$ is not changing much with $t$, at least with respect to the measurement basis $\{|\xi\rangle\}$.

We can quantify the accuracy of the estimator $T_{\text{est}}$ using the following measure:

$$(\delta T_{\text{est}})^2 \equiv \overline{(T_{\text{est}}(\xi_1, \dots, \xi_N) - t_0)^2} = \sum_{\xi_1, \dots, \xi_N} p_{\xi_1}(t_0) \cdots p_{\xi_N}(t_0) \left( T_{\text{est}}(\xi_1, \dots, \xi_N) - t_0 \right)^2 , \tag{2}$$

$$p_\xi(t) \equiv \langle\xi| \rho(t) |\xi\rangle ,$$

where the sum of each $\xi_i$ is over all possible values of $\xi_i$. Based on the general theory of statistical inference of a parameter in a probability distribution using samples from the distribution, $(\delta T_{\text{est}})^2$ has a known lower bound over all possible choices of $T_{\text{est}}$. This lower bound can be expressed in terms of intrinsic properties of the family of probability distributions $p_\xi(t)$:

$$(\delta T_{\text{est}})^2 \geq \frac{1}{N f(t_0)} , \qquad f(t_0) \equiv \sum_\xi \frac{1}{p_\xi(t_0)} \left( \left. \frac{\partial p_\xi(t)}{\partial t} \right|_{t_0} \right)^2 , \tag{3}$$

where $f(t_0)$ is known as the Fisher information for the probability distribution $p_\xi(t)$ at $t = t_0$. The above inequality is known as the Cramer-Rao bound, and can be saturated by a known form of $T_{\text{est}}$ for large $N$, known as the maximum likelihood estimate (MLE) [33, 34]. The maximum likelihood estimate $T_{\text{ML}}(\xi_1, \dots, \xi_N)$ is the value of $t$ that corresponds to the global maximum of the following "log-likelihood function" over $t$:

$$\ell(\xi_1, \dots, \xi_N | t) = \sum_{i=1}^{N} \log p_{\xi_i}(t), \qquad p_\xi(t) \equiv \langle\xi| \rho(t) |\xi\rangle . \tag{4}$$

Using our knowledge of $\sigma_0$ and $H$, we can find the quantities $p_\xi(t)$ as functions of $t$ for any outcome $|\xi\rangle$, and hence $T_{\text{ML}}(\xi_1, \dots, \xi_N)$ can be evaluated in a straightforward way for a given set of measurement outcomes. We will discuss further details of the MLE and its explicit implementation for time estimation in Sec. 5. The existence of an explicit estimator function that saturates the bound (3) shows that $f(t)$ provides a quantitative measure of the lowest possible achievable uncertainty in estimating $t$.

For a quantum state $\rho(t)$, it is natural to consider the maximum value of the Fisher information over all possible choices of the measurement basis $\{|\xi\rangle\}$. This maximum value is known as the quantum Fisher information (QFI) [4], and has the following explicit formula:

$$F(t) \equiv \text{Tr}\left[ \frac{\partial \rho}{\partial t} \mathcal{R}_\rho^{-1} \left( \frac{\partial \rho}{\partial t} \right) \right], \tag{5}$$

$$\mathcal{R}_\rho^{-1}(\cdot) \equiv \sum_{\substack{i,j \text{ s.t.} \\ p_i + p_j \neq 0}} \frac{2}{p_i + p_j} |\psi_i\rangle \langle\psi_i| (\cdot) |\psi_j\rangle \langle\psi_j| . \tag{6}$$

Here $p_i$ and $|\psi_i\rangle$ are respectively the eigenvalues and eigenstates of $\rho(t)$.[2] As discussed in [4], the QFI equivalent to a certain distance measure known as the Bures distance between the

---

[2]The superoperator $\mathcal{R}_\rho^{-1}(\cdot)$ is the inverse of $\mathcal{R}_\rho(\cdot) \equiv \{\rho, \cdot\}$. $\mathcal{R}_\rho^{-1}(\partial_t \rho)$ is known as the symmetric logarithmic derivative of $\rho$.

states at $t$ and $t + dt$:

$$F(t) = 4 \lim_{dt \to 0} \frac{d_{\text{Bures}}(\rho(t), \rho(t+dt))^2}{dt^2},$$

$$d_{\text{Bures}}(\rho, \sigma) \equiv \sqrt{2} \left(1 - \text{Tr}\left[\sqrt{\sqrt{\rho}\sigma\sqrt{\rho}}\right]\right)^{\frac{1}{2}}. \tag{7}$$

It is therefore natural to view the QFI for time estimation as an intrinsic velocity associated with changes in the state.

Note that since the optimal projective measurement basis is the eigenbasis of $\mathcal{R}_\rho^{-1}(\partial\rho/\partial t|_{t_0})$, it depends on $t_0$ itself. Hence, the quantum Fisher information is the uncertainty $(\delta T_{\text{est}})^2$ associated with attempts to locally estimate values of time $t_0' = t_0 + \Delta t$ *close to* $t_0$, and assuming knowledge of $t_0$.

So far, our discussion was completely general, and would apply to any one-parameter family $\rho(X)$ labelled by some continuous variable $X$. Let us now consider turn to the value of QFI on taking $\rho(t) = e^{-iHt}\sigma_0 e^{iHt}$, i.e., the time-evolved state on the full system. In this case, we find that the QFI is constant with respect to $t$. For an initial pure state $|\psi_0\rangle$, the QFI at all times is proportional to the energy variance:

$$F(t) = F(0) = 4\left(\langle H^2\rangle_{\psi_0} - \langle H\rangle_{\psi_0}^2\right), \tag{8}$$

where $\langle \cdots \rangle_\psi$ indicates expectation values in $|\psi\rangle$. In particular, this implies that the QFI is zero at all times for an energy eigenstate, consistent with the fact that an eigenstate does not show any non-trivial time-evolution. On the other hand, a product state in a local lattice system has energy variance of $\mathcal{O}(n)$, where $n$ is the number of sites,[3] indicating that the optimal uncertainty $(\delta T_{\text{est}})^2$ is very small in the thermodynamic limit. This result is true for any Hamiltonian. It can be seen as a purely kinematical result of unitarity which is insensitive to thermalization, similar to the fact that the von Neumann entropy of the state on the full system is invariant under unitary evolution.

One can check that for a pure state, the optimal measurement basis [4], given by the eigenbasis of $\mathcal{R}_\rho^{-1}([H, |\psi(t)\rangle\langle\psi(t)|])$, consists of the states

$$\frac{1}{\sqrt{2}}(|\psi(t)\rangle \pm i|\xi(t)\rangle), \qquad |\xi(t)\rangle \equiv \frac{(H - \langle H\rangle_\psi)}{\sqrt{\langle H^2\rangle_\psi - \langle H\rangle_\psi^2}}|\psi(t)\rangle, \tag{9}$$

and any set of $2^n - 2$ states that form an orthonormal basis together with these two states. The optimal measurements on the full system, for which the time uncertainty is determined by the constant QFI (8), are therefore as complex as the state $|\psi(t)\rangle$ itself and become increasingly complex at late times.

As discussed in the introduction, from the intuition about thermalization that a state evolved by a chaotic Hamiltonian relaxes to a steady state, we may expect the Fisher information to decay monotonically and go to zero at late times. Indeed, [31] previously studied the evolution of the classical Fisher information associated with time-evolution for a probability distribution evolved by the Fokker-Planck equation, which describes relaxation dynamics, and found results consistent with this intuition. We have seen above that the QFI on the full system shows a very different behaviour due to unitarity. In order to see results that match our expectations from thermalization, it is therefore necessary to put certain restrictions on the measurements we can perform to try to estimate the time.

---

[3]Let the Hamiltonian be a sum of local terms $H = \sum_i h_i$. The energy variance is $\text{Var}(H) = \sum_{ij}\langle h_i h_j\rangle_{\psi_0} - \langle h_i\rangle_{\psi_0}\langle h_j\rangle_{\psi_0}$, and for a product state $\langle h_i h_j\rangle_{\psi_0}$ factorizes when $h_i, h_j$ have no common support. Hence, the sum is restricted to $i \approx j$ and consists of $\mathcal{O}(n)$ terms each of $\mathcal{O}(1)$ magnitude.

One natural restriction is that the measurements can only act on some subsystem $A$ of the full system, which corresponds to taking $\rho(t) = \sigma_A(t) = \text{Tr}_{\bar{A}}[e^{-iHt}\sigma_0 e^{iHt}]$ in (5), and yields the quantity

$$F_A(t) = \sum_{\substack{i,j \text{ s.t.} \\ p_i + p_j \neq 0}} \frac{2}{p_i + p_j} |\langle i|\text{Tr}_{\bar{A}}[H, e^{-iHt}\sigma_0 e^{iHt}]|j\rangle|^2, \tag{10}$$

where $p_i, |i\rangle$ are eigenvalues and eigenstates of $\sigma_A(t)$ (we do not show the time-dependence of these eigenvalues and eigenstates explicitly for conciseness). We will refer to (10) as the *subsystem QFI*. When the initial state is pure, recall that we can write a Schmidt decomposition of the time-evolved state as follows:

$$|\psi(t)\rangle = \sum_{i=1}^{\min(d_A, d_{\bar{A}})} \sqrt{p_i} |i\rangle_A |\tilde{i}\rangle_{\bar{A}}, \tag{11}$$

where $p_i, |i\rangle_A, |\tilde{i}\rangle_{\bar{A}}$ are time-dependent but we do not show the time-dependence explicitly. We give an expression for $F_A(t)$ in Appendix A that depends on $H, p_i, |i\rangle_A, |\tilde{i}\rangle_{\bar{A}}$. This expression will be useful for later calculations. It is also conceptually interesting to note that $F_A(t)$ thus depends not only on the eigenvalues of $\rho_A(t)$ but also on the global structure of the state, including the eigenstates of $\rho_A$ as well as $\rho_{\bar{A}}$ and their relation to the Hamiltonian $H$. This is in contrast to most information-theoretic quantities that are often studied for the time-evolved state such as the von Neumann and Renyi entropies, which depend only on the eigenvalues $p_i$.[4] In Section 3, we will discuss the behaviour of the subsystem QFI in quantum many-body systems.

Another natural restriction we can put on the measurements that we used to estimate the time is to require them to be simple in the sense of computational complexity. We can quantify the uncertainty of the time estimate using a general measurement basis $\{|\xi\rangle\}$ with the classical Fisher information (CFI) for that basis $f(t)$, defined in (3). The simplest choice is to take $\{|\xi\rangle\}$ to be the computational basis, which we will call $f^{\text{comp}}(t)$. We will discuss this case in Section 4.

# 3 Quantum fisher information for subsystems

In this section, we will study the behavior of the subsystem QFI $F_A(t)$ defined in (10). We will be interested mostly in chaotic systems with local interactions. We will consider initial states for which the constraints from energy conservation are not very important at late times, such that $|\psi(t)\rangle = e^{-iHt}|\psi_0\rangle$ macroscopically resembles the infinite temperature thermal density matrix $\mathbf{1}/d$ in terms of correlation functions and entanglement entropies of small subsystems. We expect that more general initial states with extensive energy variance should show similar behaviours, and will propose generalizations of some of our results to these cases.

In Fig. 1, we highlighted various qualitative features of the evolution of $F_A(t)$. Recall that we refer to the number of degrees of freedom and Hilbert space dimension of subsystem $A$ as $n_A, d_A$ respectively. Let us further summarize some additional features, which we expect should be robust in local chaotic systems:

1. $F_A(t)$ for $n_A < n_{\bar{A}}$ has an exponentially decaying regime at intermediate to late times before saturation:

$$F_A(t) \sim e^{-\alpha_{n_A} t}, \tag{12}$$

where $\alpha_{n_A}$ has an inverse power law dependence on $n_A$, $\alpha_{n_A} \sim 1/n_A^{\eta}$ for some $\eta \geq 1$.

---

[4]Another set of recently introduced quantities that probe the global structure of the state $|\psi(t)\rangle$ in a different way are "state $k$-design" frame potentials introduced in the context of deep thermalization [7, 32, 35–38].

2. For $n_A > n_{\bar{A}}$, the time scale $t^*$ corresponding to the minimum of $F_A(t)$ in Fig. 1 obeys $t^* \sim n_{\bar{A}}$.

3. The saturation value of $F_A(t)$ is

$$F_A \propto \begin{cases} n_A d_A / d_{\bar{A}}, & n_A < n_{\bar{A}}, \\ n_A, & n_A > n_{\bar{A}}, \end{cases} \tag{13}$$

where the proportionality constant has units of energy density squared.

The above features are deduced from a combination of different numerical as well as analytic models. Let us summarize the different models that we used below, and highlight which of the features of Fig. 1 and points 1-3 above can be observed in each of them:

1. The mixed-field Ising model at generic values of the coupling exhibits each of the features summarized in the figure and the above points 1-3. This spin chain model is often used in the literature to probe universal characteristics of local chaotic dynamics. We present the numerical results from this model in Sec. 3.1. We also consider an a free fermion integrable version of the spin chain, where we find a strikingly different behaviour of $F_A(t)$ for $n_A < n_{\bar{A}}$.

2. The exponential regime (12) for $n_A < n_{\bar{A}}$ is relatively short-lived in the spin chain model before it saturates for the system sizes we can access numerically. To better understand this decay, we model the smaller subsystem $A$ as an open quantum system with boundary dissipation and numerically study the behaviour of the QFI in this model. This model allows us to better understand the form and physical origin of the $n_A$-dependent decay rate in (12). By construction, this model cannot give the saturation values (13) observed in unitary systems.

3. To better understand the saturation value of $F_A(t)$ in unitary systems, we model the late-time state $|\psi(t)\rangle$ as a random pure state in the full Hilbert space drawn from the uniform (Haar) ensemble in Sec. 3.3. This analytic calculation confirms the the universality of the scaling in (13).

4. As a simple toy model for the increase of $F_A(t)$ at intermediate times for a subsystem larger than half of the system, in Appendix C, we model the time-evolved state $|\psi(t)\rangle$ using the Brownian GUE Hamiltonian [39, 40]. We find that the analytic result for $F_A(t)$ in this model qualitatively matches the increasing behavior that is numerically observed in the spin chain model. The model is too simple to reproduce features coming from locality, such as the numerical observation that $t^* \propto n_{\bar{A}}$. Such consequences of locality can likely be better understood by making use of techniques such as Lieb-Robinson bounds,[5] and should be further explored in future work.

Based on the expectation that the above results are universal, one particularly interesting example of a quantum chaotic system to which they should apply is a black hole. We discuss the implications of these results in the context of evaporating black holes in Sec. 3.4.

Before turning to specific models, let us discuss the general physical reasonableness of the results summarized above. Both the time-evolution and the saturation value of $F_A(t)$ for $n_A < n_{\bar{A}}$ is intuitive from the perspective of thermalization. In particular, the scaling $d_A / d_{\bar{A}}$ of the saturation value of $F_A(t)$ matches the expected scaling of the distance of the late-time state from the maximally mixed state, i.e., $\left( d_{\text{Bures}} \left( \rho_A(t), \frac{1}{d_A} \right) \right)^2 \sim d_A / d_{\bar{A}}$ [8, 41–43]. Note that the

---

[5]We thank Adam Bouland for this comment.

physical interpretation of the two quantities is somewhat different, and they did not necessarily have to show the same scaling based on kinematical reasoning: $F_A(t) = \left(\frac{d_{\text{Bures}}(\rho_A(t), \rho_A(t+dt))}{dt}\right)^2$ measures the *speed* of fluctuations, while $d_{\text{Bures}}\left(\rho_A(t), \frac{1}{d_A}\right)$ measures the absolute magnitude of fluctuations.

Perhaps the most striking feature of Fig. 1 is that for $n_A > n_{\bar{A}}$, we have an increasing behaviour of $F_A(t)$ at intermediate times, and an extensive late time-value. These behaviours cannot be understood purely from intuitions about thermalization, and require an understanding of new features of the dynamics of a unitary chaotic system.

We derive the result of non-monotonic time-evolution from numerical simulations in the chaotic spin chain in Sec. 3.1. To better understand its physical origin, recall that the reduced density matrix of $\rho_A(t)$ is not full-rank for $n_A > n_{\bar{A}}$. We can write the Hilbert space of $A$ as a direct sum $\mathcal{H}_A = \mathcal{H}_{\text{ent}} \oplus \mathcal{H}_{\text{null}}$, where $\mathcal{H}_{\text{ent}}$ is the subspace spanned by the eigenstates of $\rho_A$ with non-zero eigenvalues, and $\mathcal{H}_{\text{null}}$ is the complementary subspace. Then we can divide $F_A(t)$ in (10) into two non-negative terms,

$$F_A(t) = F_{A,\text{ent}}(t) + F_{A,\text{rot}}(t),\tag{14}$$

where $F_{A,\text{ent}}(t)$ includes the terms in (10) where $|i\rangle, |j\rangle \in \mathcal{H}_{\text{ent}}$, while $F_{A,\text{rot}}(t)$ includes the terms where either $|i\rangle$ or $|j\rangle$ is in $\mathcal{H}_{\text{null}}$. Moreover, $F_{A,\text{rot}}(t)$ can be shown to be equal to

$$F_{A,\text{rot}}(t) = \text{Tr}\left[\rho_A\left(\frac{dP_{\text{ent}}}{dt}\right)^2\right],\tag{15}$$

where $P_{\text{ent}}$ is the projector onto the subspace $\mathcal{H}_{\text{ent}}$. Note that no analog of this term is present in the the expression for $F_A(t)$ when $n_A \leq n/2$, as $F_A(t)$ quickly becomes full-rank in this case. On the other hand, if we take $A$ to be the full system and consider a pure state, then $F_{A,\text{rot}}(t)$ is the only contribution to $F_A(t)$.

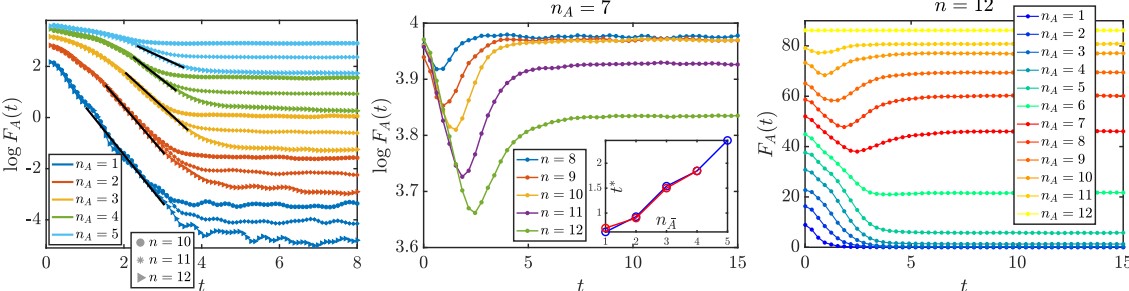

Figure 3: We plot the dynamics of $F_A(t)$ in the chaotic spin chain model (16) with various $n$ and $n_A$. We evolve the system from random product initial states and average $F_A(t)$ over 200 samples of initial states for $n \leq 11$ and 84 samples for $n = 12$. **Left:** We focus on cases with $n_A \leq n/2$. The solid black lines show the fitting to exponential decay. The rate of exponential decay decreases with $n_A$ and is independent of $n_{\bar{A}}$. (The decay rates $\alpha_{n_A}$ as a function of $n_A$ are shown in Fig. 7(right).) The saturation value decays exponentially with $n_{\bar{A}}$ for fixed $n_A$. **Middle:** We focus on the region with $n_A > n/2$, and contrast the non-monotonic time dependence with the monotonic behaviour for $n_A < n/2$ shown in left panel. The saturation value depends very weakly on $n_{\bar{A}}$ for fixed $n_A$, and we expect this weak dependence to go away in the thermodynamic limit. **Middle, inset:** We plot $t^*$ (the time at which $F_A(t)$ is minimized, with $n_A > n/2$) for various $n_A, n$ as a function of $n_{\bar{A}}$. Red/blue lines correspond to $n_A = 7, 8$. We find that $t^* \sim n_{\bar{A}}$, and does not depend on $n_A$. **Right:** We plot the dynamics of $F_A(t)$ with fixed $n = 12$ and various subsystem sizes.

By considering the behaviour of $F_{A,\text{rot}}(t)$ and $F_{A,\text{ent}}(t)$ separately, we find that $F_{A,\text{ent}}(t)$ monotonically decays with time, while $F_{A,\text{rot}}(t)$ monotonically increases with time (see Fig. 5). The competition between these two terms leads to the non-monotonic evolution of $F_A(t)$.

$F_{A,\text{ent}}(t)$ is quantitatively close to $F_{\bar{A}}(t)$ in general, and the two can be shown to be equal in the case where the spectrum of $\rho_A$ is flat (see around eq. (A.13) in Appendix A). $F_{A,\text{rot}}(t)$ is the dominant contribution to $F_A(t)$ at late times. Heuristically, its increasing behaviour captures the fact that the support of $\rho_A(t)$ gains access to more and more of full Hilbert space $\mathcal{H}_A$ to rotate within as time evolves. Eventually, it has access to the full Hilbert space of $\mathcal{H}_A$ and rotates at a large universal speed, leading to the extensive saturation value (see Fig. 2).

In Sec. 6, we will provide a further physical interpretation of the transition in the saturation value of the QFI as a function of subsystem size in (13) in terms of quantum error-correction.

## 3.1 Chaotic and integrable spin chains

Let us first study the behaviour of the subsystem QFI in the mixed-field Ising model:

$$H = \sum_i Z_i Z_{i+1} + g \sum_i X_i + h \sum_i Z_i, \tag{16}$$

with periodic boundary conditions. For generic values of $g$ and $h$, this model is chaotic by a variety of measures, including energy eigenvalue statistics [44], entanglement growth [23,45], and operator growth [46]. We take $g = -1.05$ and $h = 0.5$ as a standard representative of the chaotic case. The particular case where $h = 0$ is the integrable transverse field Ising model, which is integrable and can be written as a free fermion Hamiltonian.

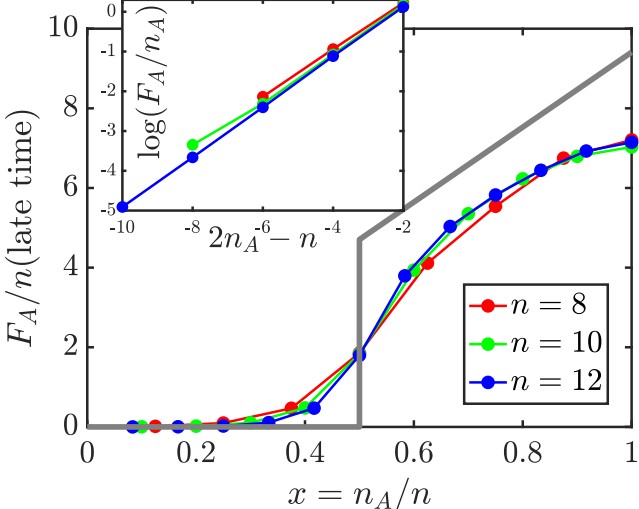

Figure 4: We plot the late time saturation value of $F_A(t)$ (averaged over 800 samples of initial states and 10 instances of time in the interval $t \in [15, 20]$) as a function of subsystem fraction $x \equiv n_A/n$, for various $n$. The solid grey line is the theoretical prediction from the random pure state model introduced in 3.3, in the thermodynamic limit $n \to \infty$. Even for finite size numerical results, we see a hint of a phase transition at $x = 1/2$. We expect based on the random pure state model that the result is likely to be extensive in $n_A$ in the thermodynamic limit, which may be obscured by finite size effects here. **Inset:** Using the same data, we plot the log of the QFI saturation value for $x < 1/2$. We see a clear collapse as a function of $2n_A - n$ showing a scaling of $\sim d_A/d_{\bar{A}}$, as predicted in (22).

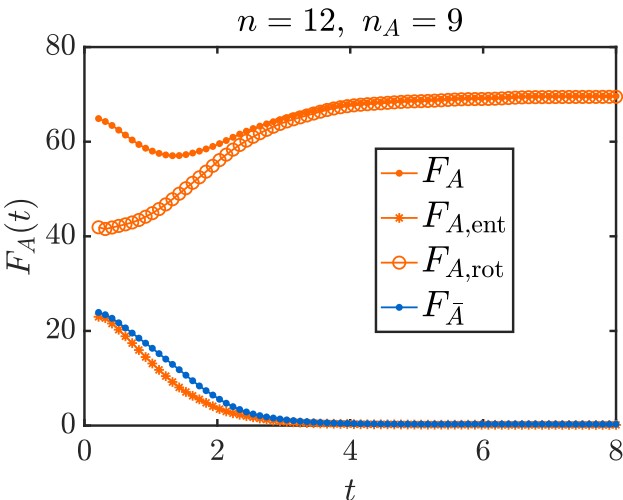

Figure 5: We plot the dynamics of $F_A(t), F_{\bar{A}}(t)$ for $n = 12, n_A = 9$ (averaged over 84 samples of initial random product states), along with the two terms $F_{A,\text{ent}}$ and $F_{A,\text{rot}}$ defined in (14). We observe that the non-monotonic behavior of $F_A(t)$ comes from the competition between $F_{A,\text{ent}}(t)$ and $F_{A,\text{rot}}(t)$.

We will consider the evolution of $F_A(t)$ for subsystem of contiguous spins of size $n_A$ on a chain of length $n$. We take the initial states to be random product states between the different sites, for which the initial value of $F_A(t)$ is extensive in $n_A$. Such states are known to macroscopically equilibrate to infinite temperature [47]. We will average over samples drawn from the uniform measure on such states. The numerical results for $F_A(t)$ for the chaotic spin case are shown in Fig. 3, 4, and 5. Let us summarize the key features of these numerical results below.

For $n_A < n/2$, $F_A(t)$ shows a monotonic decay with $t$, which is of the form $F_A(t) \sim e^{-\alpha_{n_A} t}$ at intermediate times. The decay rate $\alpha_{n_A}$ decreases with increasing $n_A$, and is independent of $n_{\bar{A}}$. See Fig. 3 (left). The saturation value of $F_A(t)$ is proportional to $d_A/d_{\bar{A}} = 2^{2n_A - n}$, as shown more explicitly in Fig. 4.

For $n_A > n/2$, $F_A(t)$ shows an initial decay up to a time $t^*$. $t^*$ depends only on $n_{\bar{A}}$, and is roughly proportional to $n_{\bar{A}}$. The fact that the behavior is qualitatively similar to the case $n_A < n/2$ up to this $t^*$ physically makes sense, as it is only on this time scale that information from the larger system can travel across the smaller subsystem in the presence of local interactions. For $t > t^*$, $F_A(t)$ increases with time and later saturates. The saturation value is independent of $n_{\bar{A}}$ and grows with $n_A$, as shown in Fig. 4.[6]

In Fig. 5, we separately plot the two terms $F_{A,\text{rot}}$ and $F_{A,\text{ent}}$ defined below (14). We find that $F_{A,\text{ent}}(t)$ is both qualitatively and quantitatively close to $F_{\bar{A}}(t)$, and shows monotonic decay. $F_{A,\text{rot}}(t)$ monotonically grows, and fully accounts for the value of $F_A(t)$ at late enough times.

In the free-fermion integrable case of the spin chain model (16), where we set $h = 0$, we find strikingly different behavior of both the time-evolution and the saturation value of $F_A(t)$ for $n_A < n/2$ compared to the chaotic case. See Fig. 6. These behaviours indicate that the state is far from equilibrating for the free-fermion integrable model, even for a small subsystem. As shown in Fig. 6, the evolution with time is non-monotonic even for $n_A$ much smaller than $n/2$. For both small and large $n_A$, the average late-time value saturation value is independent of $n_{\bar{A}}$, and grows with $n_A$. The time-evolution and saturation value of the QFI thus provides a

---

[6]We find evidence for these statements up to finite-size corrections which make the data somewhat noisy. We will provide further evidence for various features using the models in Sec. 3.2 and Sec. 3.3, which are less affected by finite-size corrections.

sharp qualitative probe of the difference between chaotic and free fermion integrable systems. In the Heisenberg XXZ spin chain, an example of an interacting integrable system, we find similar behavior to the chaotic case. The numerical results from this model are presented in Appendix D.

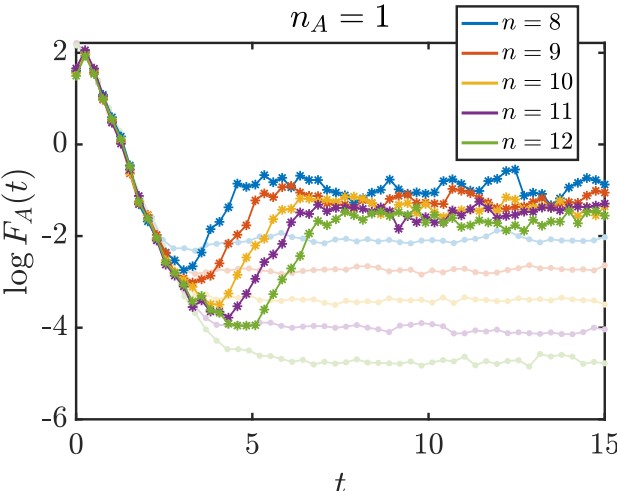

Figure 6: We plot the dynamics of $F_A(t)$ from the free fermion integrable spin chain model for $n_A = 1$ and various full system sizes, averaged over 400 samples of initial random product states. For comparison, we also plot the $F_A(t)$ from the chaotic spin chain using the same but lighter colors. We observe that the time evolution coincides initially, but differs at later times. While the saturation value in the chaotic case decays exponentially with $n$ for fixed $n_A$, the average value at late times in the integrable case is constant with respect to $n$.

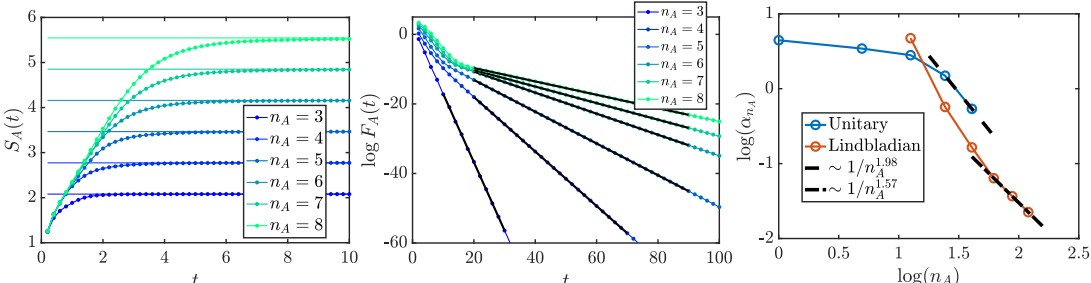

Figure 7: **Left:** The dynamics of the von Neumann entropy $S(t) = -\mathrm{Tr}[\rho(t)\log\rho(t)]$ in the Lindbladian model, which shows a size-independent linear growth rate and saturation to $\log d_A$ at late times. These features resemble the behavior of a subsystem in the unitary model of Sec. 3.1. **Middle:** We plot the QFI dynamics in Lindbladian model, again starting from random initial states. The black solid lines show the fitting to the exponential decay rate. While we use the notation $F_A$, note that we are now considering different full system sizes $n_A$ of the open quantum system. **Right:** We record the exponential decay rates $\alpha_{n_A}$ in both the unitary chaotic spin chain model of Sec. 3.1 and the non-unitary Lindbladian model. The decay rates $\alpha_{n_A}$ are obtained from fitting, as shown in the black solid lines in left panel of Fig. 7 and left panel of Fig. 3.

## 3.2 Open quantum system with boundary dissipation

In this section, we consider an open system Lindbladian dynamics with boundary dissipation. For comparison with the results of the previous section, we call the number of sites in the full system $n_A$. The state $\rho(t)$ on the full system is evolved under a time-independent Lindbladian equation, which evolves the initial pure state to a mixed state:

$$\frac{\mathrm{d}}{\mathrm{d}t}\rho(t) = \mathcal{L}(\rho(t)) = -i[H, \rho(t)] + \gamma \sum_a \left( L_a \rho(t) L_a^\dagger - \frac{1}{2}\{L_a^\dagger L_a, \rho(t)\} \right), \qquad (17)$$

where $\{L_a\}$ is a set of jump operators and $\gamma \in \mathbb{R}_+$ controls the dissipation strength. We take $H$ to be the same chaotic Ising Hamiltonian as (16), except now living on $n_A$ sites and with open boundary conditions instead of periodic boundary conditions. We take the Lindblad operators $\{L_a\}$ to act on the two boundaries of the system, and make the arbitrary choice that the dissipation channel is the depolarization channel, with order one strength ($\gamma = 1$). Then the set of jump operators $\{L_a\}_{a=1}^6$ is specified as:

$$\{L_a\}_{a=1}^6 = \{X_1, Y_1, Z_1, X_{n_A}, Y_{n_A}, Z_{n_A}\}. \qquad (18)$$

As a first check to get some intuition about this model, we show the dynamics of the von Neumann entropy of the state $\rho(t)$ on the full system in the right panel of Fig. 7. This matches the qualitative behavior of the entanglement entropy of a subsystem of a unitary chaotic spin chains very well [47]. It is thus natural to view this open system model as a way of extracting properties of the unitary system of Sec. 3.1 in the thermodynamic limit with $n_A \ll n$.

Under evolution by (17), $\rho(t)$ approaches the maximally mixed state in the following way:

$$\rho(t) \approx \frac{\mathbb{1}}{d} + \rho_1 e^{-\lambda_1 t}, \qquad (19)$$

where $-\lambda_1$ is the eigenvalue of $\mathcal{L}(\cdot)$ with the least negative real part, and $\lambda_1$ is known as the Lindbladian gap. $\rho_1$ is some traceless operator. As the spectrum of $\rho(t)$ approaches that of a maximally mixed state, we can approximate $F(t) \approx d\operatorname{Tr}(\dot\rho(t)^2)$ (here we apply the general formula (5)). Using (19), we find the exponential decay

$$F(t) \sim e^{-2\lambda_1 t}. \qquad (20)$$

In Ref. [48], the authors study the Lindbladian gap of a variety of one-dimensional chaotic spin chain models with boundary dissipation, and find that it scales as $\lambda_1 \sim 1/n_A^\eta$, where $\eta \geq 1$ is some $\mathcal{O}(1)$ number. This universal behaviour of the Lindbladian gap thus naturally explains our observation in Sec. 3.1 that the decay rate $\alpha_{n_A}$ decreases as we increase the system size $n_A$.

For concreteness, in the left panel of Fig. 7, we show the QFI of $\rho(t)$ under such a Lindbladian, and indeed find qualitatively the same behaviour of the QFI as in the previous subsection on varying the system size $n_A$. In the right panel of Fig. 7, we plot the decay rate $\alpha_{n_A}$ as a function of $n_A$, and observe that it indeed scales in a power law fashion, with power larger than one. The time evolution data is generated by Runge-Kutta method, and the decay rate from fitting the data matches twice Lindbladian gap.

## 3.3 Random pure states

A useful model for the late-time state $|\psi(t)\rangle$ in a unitary chaotic evolution on a finite-dimensional Hilbert space is a random pure state drawn from the uniform (Haar) ensemble. We will use this state $|\psi_{\mathrm{Haar}}\rangle$ to understand the saturation value of $F_A(t)$ at late times, ignoring the fact that the state up to this point was evolved by the Hamiltonian $H$. We expect this

approximation to be reasonable for states whose dynamics are not significantly constrained by the effects of energy conservation.

More explicitly, we replace $e^{-iHt}\sigma_0 e^{iHt}$ in (10) with $|\psi_{\mathrm{Haar}}\rangle\langle\psi_{\mathrm{Haar}}|$, and the eigenvalues $p_i$ and eigenvectors $|i\rangle$ of $\sigma_A(t)$ with those of $\mathrm{Tr}_{\bar{A}}|\psi_{\mathrm{Haar}}\rangle\langle\psi_{\mathrm{Haar}}|$, and then average over $|\psi_{\mathrm{Haar}}\rangle$ in the full expression.

For this calculation, the general expressions for $F_A(t)$ of a pure state in terms of the Schmidt decomposition in Appendix A will prove useful. To use these, we need to describe the ensemble of Haar-random states in terms of their Schmidt decomposition, which is given as follows (here we use $S$ to refer to the smaller subsystem out of $A$ and $\bar{A}$, and $\bar{S}$ to refer to its complement):

$$|\psi_{\mathrm{Haar}}\rangle = \sum_{i=1}^{d_S} \sqrt{p_i}\,(V\,|i\rangle)_S\,(U\,|\tilde{i}\rangle)_{\bar{S}}\,. \tag{21}$$

The $d_S$ real numbers $p_i$, the $d_S \times d_S$ unitary $V$, and the the $d_{\bar{S}} \times d_{\bar{S}}$ unitary $U$ are random and uncorrelated with each other. $U$ and $V$ are both Haar-random unitaries in their respective Hilbert spaces. $\{|i\rangle\}$ and $\{|\tilde{i}\rangle\}$ are arbitrary fixed sets of $d_S$ orthonormal states in $S$ and $\bar{S}$ respectively. $p_i$ have the statistics of the eigenvalues of normalized Wishart matrices $\frac{YY^\dagger}{\mathrm{Tr}[YY^\dagger]}$, where $Y$ is a $d_S \times d_{\bar{S}}$ matrix of independent complex Gaussian random variables drawn from the distribution $p(Y) = \mathcal{N}^{-1}e^{-d_{\bar{S}}\mathrm{Tr}[YY^\dagger]}$. Using the expressions of Appendix A and (21), we can write the expression for $F_A$ in terms of averages of just second moments $U \otimes U^* \otimes U \otimes U^*$ of Haar-random unitaries $U$ (similarly for $V$), so we get identical results for the more general class of states where $V$ and $U$ are unitary 2-designs.

Let us decompose a general geometrically local Hamiltonian $H$ as $H = H_A + H_{\bar{A}} + H_{\mathrm{int}}$, where $H_A$ consists of all terms entirely within $A$, $H_{\bar{A}}$ of all terms entirely within $\bar{A}$, and $H_{\mathrm{int}}$ of all terms across the boundary.[7] Up to small corrections which can be ignored in the thermodynamic limit, we show in Appendix B that (21) gives the following prediction for the late-time value of the subsystem QFI:

$$F_A = \begin{cases} 2\frac{d_A}{d_{\bar{A}}}\left(\frac{\mathrm{Tr}_A[H_A^2]}{d_A} - \frac{\mathrm{Tr}_A[H_A]^2}{d_A^2}\right) = c\,n_A\,d_A/d_{\bar{A}}, & n_A < n/2\,, \\ 4\left(\frac{\mathrm{Tr}_A[H_A^2]}{d_A} - \frac{\mathrm{Tr}_A[H_A]^2}{d_A^2}\right) = 2c\,n_A, & n_A > n/2\,, \end{cases} \tag{22}$$

where $c$ is a constant of units energy density squared which depends on $\mathcal{O}(1)$ microscopic couplings. This expression approximately reproduces the numerically observed late-time scaling in the chaotic case of the spin chain model, as shown in Fig. 4. We do not find a quantitative match between the two models, likely both due to finite size effects and due to the oversimplification in completely ignoring the effects of energy conservation by using (21).

The above result does not directly apply to initial states $|\psi_0\rangle$ which macroscopically resemble a finite temperature equilibrium density matrix $\rho^{(\mathrm{eq})}$ at late times (as opposed to the infinite-temperature maximally mixed state $\mathbf{1}/d$). A particular example we might have in mind is the canonical ensemble $\rho_\beta = e^{-\beta H}/Z_\beta$ at late times. We expect that a natural generalization of (22) to such states should be given by replacing the infinite-temperature energy variances in (22) with finite-temperature ones:

$$F_A = \begin{cases} 2(\langle H_A^2\rangle_{\mathrm{eq}} - \langle H_A\rangle_{\mathrm{eq}}^2)\,e^{S_{A,\mathrm{eq}}-S_{\bar{A},\mathrm{eq}}}, & S_{A,\mathrm{eq}} < S_{\bar{A},\mathrm{eq}}\,, \\ 4(\langle H_A^2\rangle_{\mathrm{eq}} - \langle H_A\rangle_{\mathrm{eq}}^2), & S_{A,\mathrm{eq}} > S_{\bar{A},\mathrm{eq}}\,, \end{cases} \tag{23}$$

---

[7]For geometrically non-local Hamiltonians such as the SYK model, the expression (22) does not apply. The result for such cases can be deduced by starting with the intermediate expressions (B.9)-(B.10) and (B.22) in Appendix B, which are completely general and hold for any Hamiltonian, and then putting in the assumptions relevant to the non-local or $k$-local Hamiltonian of interest.

where $S_{A,\text{eq}}$ and $S_{\bar{A},\text{eq}}$ are respectively the von Neumann entropies of $\rho_{\text{eq,A}} \equiv \text{Tr}_{\bar{A}}\rho^{(\text{eq})}$ and $\rho_{\text{eq},\bar{A}} \equiv \text{Tr}_A\rho^{(\text{eq})}$ (which can be seen as thermodynamic entropies of $\rho_{\text{eq}}$ in $A$ and $\bar{A}$), and $\langle \cdots \rangle_{\text{eq}}$ indicates expectation values in $\rho^{(\text{eq})}$.

## 3.4 Evaporating black holes

If we assume that the formation of a black hole and its subsequent evaporation are governed by a unitary and chaotic evolution, the combined state of the black hole and its radiation can be seen as an example of $|\psi(t)\rangle$ at late times in a chaotic quantum many-body system. The simplest model for this combined state is a Haar-random pure state (21). Results from random pure states and their natural finite-temperature generalizations have previously been used to predict the Page curve for the entropy of the black hole and radiation in [11, 16], as well as the sudden recoverability of quantum information from the black hole radiation after the Page time in [12]. Here, we will apply this model in order to predict the behaviour of the subsystem QFI of the radiation for the time estimation task.

In the evaporating black hole setup, it is natural to consider the subsystem QFI of time estimation using the radiation. For concreteness, we consider the same setup as in Hawking's calculation in [13]: the case of a non-rotating uncharged black hole in asymptotically flat space in 3+1 spacetime dimensions. The black hole is formed from spherical collapse of an initial pure state, which is the vacuum state at past infinity. In this setup, irrespective of the specific matter content (for instance, whether we consider a massless scalar field like in [13] or photons and gravitons like in [16]), the rate at which the black hole mass changes takes the form

$$\frac{dM}{dt} = -\frac{\alpha}{(G_N M)^2}, \tag{24}$$

for some dimensionless constant $\alpha$, whose value depends on the type of matter content. Integrating from $t' = 0$, when the mass is $M_0$, to $t' = t$, the time-dependent black hole mass is

$$M(t) = \left( M_0^3 - \frac{t}{G_N^2} \right)^{1/3}. \tag{25}$$

According to Hawking's semiclassical gravity calculation, the particles emitted by the black hole approximately obey the Planck spectrum at temperature

$$T_H(t) = \frac{1}{8\pi G_N M(t)}, \tag{26}$$

which grows with time during the evaporation process as $M(t)$ decreases. From (24), the fractional change in mass $dM/M$ (or equivalently the fractional change $dT_H/T_H$) becomes $\mathcal{O}(1)$ over time intervals of size $t_M = G_N^2 M^3$. Recall that from [15], the time interval between the emission of individual particles is $\sim G_N M(t)$. Hence, as long as $G_N M(t)^2 \gg 1$, which is true at all but very late times in the evaporation process, a macroscopic number of particles is emitted into the radiation in a time interval of some size $\Delta t$ around time $t$, where $G_N M(t) \ll \Delta t \ll G_N^2 M(t)^3$. Over this interval, the temperature $T_H(t)$ can be treated as a constant, and the emitted particles are approximately in the canonical ensemble at $T_H(t)$.

Based on the above discussion, we can approximate the full state of the radiation at time $t$ during the evaporation process as follows. Let us pick some discrete sequence of time intervals centered at times $t_i$ between 0 and $t$ and of width $\Delta_i$, such that $G_N M(t_i) \ll \Delta_i \ll G_N^2 M(t_i)^3$. The total number of intervals $i_{\text{max}}(t)$ is such that we cover the full range of times up to $t$, $\sum_{i=1}^{i_{\text{max}}(t)} \Delta_i = t$. Let us refer to the Hilbert space of the radiation emitted in the $i$'th such time

interval as $R_i$. Then from the assumption that the particles in $R_i$ are in the canonical ensemble at temperature $\beta_i = 1/T_H(t_i)$, the full state is

$$\rho_R^H(t) = \otimes_{i=1}^{i_{\max}(t)} \frac{e^{-\beta_i H_{R_i}}}{\text{Tr}[e^{-\beta_i H_{R_i}}]}, \tag{27}$$

where $H_{R_i}$ the Hamiltonian for the particles in $R_i$. The full Hamiltonian for the black hole and radiation is $H = H_B + \sum_i H_{R_i}$, where $H_B$ is the Hamiltonian for the remaining black hole. Now, if we assume that (27) is the true quantum state of the radiation, then due to the stationarity of (27) under $\sum_i H_{R_i}$, the state does not change at all on time scales shorter than $t_{\text{evap}} = G_N M$. (On time scales of order $t_{\text{evap}}$, the temperature does not yet change but the number of particles in the radiation grows so that $H_{R_{i_{\max}(t)}}$ changes.)

If we instead assume that a black hole is an example of a *unitary* chaotic quantum many-body system, then the true quantum state of the black hole is not given by (27), but only resembles (27) at the level of macroscopic observables. In this setup, we can apply our general formula (23), taking $\rho^{(\text{eq})} = \frac{e^{-\beta(t)H_B}}{\text{Tr}[e^{-\beta(t)H_B}]} \otimes \rho_R^H(t)$. To apply this formula, we need to find the energy variance ($\langle H_R^2 \rangle - \langle H_R \rangle^2$) of the state $\rho_R^H(t)$, which is given by sum of the energy variances for each of the canonical ensemble states $\frac{e^{-\beta_i H_{R_i}}}{\text{Tr}[e^{-\beta_i H_{R_i}}]}$. Now if we model the radiation as a $d+1$-dimensional gas of massless relativistic particles,[8] then the expectation value of the energy as a function of temperature is given by

$$\langle H_{R_i} \rangle_{\beta=1/T_i} \sim V_d^{(i)} T_i^{d+1}, \tag{28}$$

where $V_d^{(i)}$ is the $d$-dimensional spatial volume occupied by $R_i$, and $\sim$ indicates ignoring dimensionless constants, as we will continue to do in the remaining equations. The energy variance of $H_{R_i}$ is then given by

$$\langle H_{R_i}^2 \rangle_{\beta_i} - \langle H_{R_i} \rangle_{\beta_i}^2 = T^2 \frac{\partial \langle H_R \rangle_{\beta=1/T}}{\partial T}\bigg|_{T=T_i} \sim T_i \langle H_R \rangle_{\beta_i} \sim \frac{1}{G_N M(t_i)} \langle H_R \rangle_{\beta_i}, \tag{29}$$

where in the last expression we have put in $T_i = T_H(t_i)$. Now using (24) and the conservation of the total energy of the black hole and the radiation,

$$\langle H_R \rangle_{\beta_i} \approx \frac{\alpha}{(G_N M(t_i))^2} \Delta_i, \tag{30}$$

and therefore, the total energy variance in $\rho_R^H(t)$ is

$$\langle H_R^2 \rangle - \langle H_R \rangle^2 \sim \sum_{i=1}^{i_{\max}(t)} \frac{1}{G_N M(t_i)} \frac{\alpha}{(G_N M(t_i))^2} \Delta_i \approx \frac{1}{G_N^3 M_0^3} \int_0^t dt' \frac{\alpha}{1 - \frac{t'}{M_0^3 G_N^2}}$$

$$= \frac{1}{G_N} \int_0^{\frac{t}{t_{\text{total}}}} dx \frac{\alpha}{1-x} \sim \frac{1}{G_N} \log \frac{1}{1 - \frac{t}{t_{\text{total}}}}, \tag{31}$$

where we have introduced the notation $t_{\text{total}} = G_N^2 M_0^3$ for the total time until the end of the evaporation process. Hence, the energy variance in $\rho_R^H(t)$ is $\mathcal{O}(1/G_N)$, with a dimensionless factor that keeps track of the stage in the evaporation process and grows with time.

---

[8]We will not specify the choice of $d$. Up to $\mathcal{O}(1)$ dimensionless constants, the final result for the energy variance and for $F_A(t)$ will turn out to be independent of whether we assume $d = 2$ (by taking into account the fact that emission into spherically symmetric modes is favored by greybody factors), or assume $d = 4$ (ignoring the greybody factors).

Now using this expression for the energy variance in (23), we see that before the Page time, when $S_{B,\text{eq}} - S_{R,\text{eq}}$ is large, we have a very small value of the subsystem QFI of the radiation, $F_R \sim 1/G_N \mathcal{O}(e^{-1/G_N})$, which gives a time uncertainty of $\mathcal{O}(e^{1/G_N})$ (recall Eq. (3) for the relation between the Fisher information and the time uncertainty). Indeed, taking into account the change in the number of particles in the radiation gives us a better way of estimating the time in this case, with uncertainty $\sim t_{\text{evap}} = G_N M$. On the other hand, after the Page time, the QFI of $\rho_R(t)$ is given by

$$F_R(t) \sim (\langle H_R^2 \rangle - \langle H_R \rangle^2)_{\rho_R^H(t)} \sim \frac{1}{G_N} \log \frac{1}{1 - \frac{t}{t_{\text{total}}}}, \qquad S_{R,\text{eq}} > S_{B,\text{eq}}. \tag{32}$$

The corresponding uncertainty $|\delta t|$ in time using the optimal quantum measurement on $\rho_R(t)$ is $\mathcal{O}(\sqrt{G_N})$, which is much smaller than the naive best-case uncertainty of $t_{\text{evap}}$ based on Hawking's calculation. This is true as long as the black hole mass is much larger than the Planck mass ($M \gg 1/\sqrt{G_N}$). Once the black hole mass becomes comparable to the Planck mass, the black hole evolves very rapidly in Hawking's calculation and the semiclassical analysis is even naively expected to break down.

It would be interesting in future work to understand what new prescriptions may be needed to resolve this version of the information paradox and reproduce the behavior (32). See the discussion section Sec. 7 for more comments.

# 4 Classical fisher information for simple measurements

Based on our study of the quantum Fisher information in chaotic spin chain systems in the previous section, we found that with *optimal* measurements on any subsystem larger than half, the uncertainty of the time estimation task in $|\psi(t)\rangle$ is suppressed as $1/n$, where $n$ is the full system size. We conjectured that this behavior is universal in unitary chaotic quantum many-body systems. This result for $|\psi(t)\rangle$ is very different from the *infinite* uncertainty for time estimation in an equilibrium state $\rho^{(\text{eq})}$, which by definition does not evolve with time. It is reasonable to expect, however, that the measurements needed to accurately estimate the time using more than half the system in $|\psi(t)\rangle$ have a high computational complexity. This expectation would be particularly important in the context of the evaporating black hole setup of Sec. 3.4, as it would imply that computationally bounded observers would see results consistent with Hawking's state. Complexity-theoretic criteria for the validity of semiclassical gravity have previously been proposed in the context of other information-theoretic tasks, see for instance [21–26].

To address the question about the complexity of time estimation, recall from Sec. 2 that the optimal measurement basis (in which the uncertainty is given by the quantum Fisher information) is the eigenbasis of $\mathcal{R}_\rho^{-1}(\partial \rho / \partial t)$. For the full system, recall that there is a simple general expression given by (9). The resulting measurement basis is at least as complex as the state $|\psi(t)\rangle$ itself.

For a subsystem, we do not have a simple expression for the optimal measurement basis, but can find it numerically in examples like the chaotic spin chain by diagonalizing $\mathcal{R}_{\rho_A}^{-1}([\partial \rho_A / \partial t])$. While directly evaluating the computational complexity is beyond the scope of current techniques, we find that the entanglement entropy of typical states in the optimal measurement basis at late times is volume-law and close to the Page value. An example is shown in Fig 8. This tells us that the complexity of the measurement basis is at least proportional to the subsystem volume [49].

A natural alternative setup is to restrict to *simple measurements* on a subsystem, and consider the classical Fisher information (CFI) associated with them (3). On making this restric-

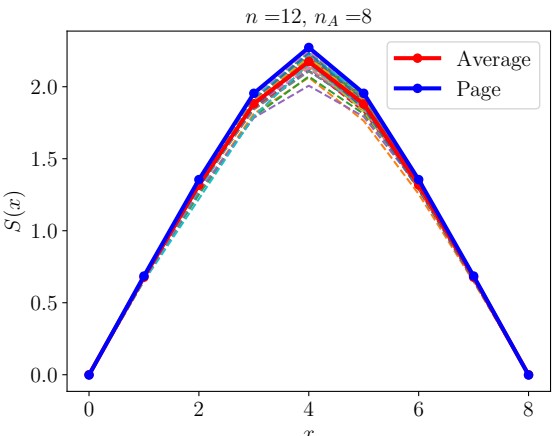

Figure 8: We evolve a single initial random product state in the chaotic spin chain (16) for $L = 12$ to $t = 10$, by which time we expect the QFI reaches its large saturation value (see Fig. 4). We pick $n_A = 8$, find the eigenstates of the operator $\mathcal{R}_{\rho_A}^{-1}(\partial \rho_A / \partial t)$, and show their entanglement entropies in arbitrarily chosen intervals of size $x$ as a function of $x$. The entropies of individual eigenstates are shown with dashed lines, and their average is shown with a thick red curve. The blue curve is the Page value [11] for the corresponding system sizes.

tion, do we get a small saturation value of the CFI even on considering a subsystem bigger than half of the system? The simplest measurements we can consider are in the computational basis (say, the eigenbasis of the $Z$ operators at each site). Both from the chaotic spin chain and from the random pure state model (21), we find that the late-time saturation value of this CFI has the following behavior:

$$f_A^{\text{comp}}(t)_{\text{late times}} = \kappa \frac{n}{d_{\bar{A}}}, \tag{33}$$

where $\kappa$ is a constant of units energy density squared which depends on microscopic couplings in the Hamiltonian. See Fig. 9 for the numerical spin chain result, and Appendix E for the random pure state calculation. Hence, as long as $n_A < n - \mathcal{O}(\log n)$, even for $n_A > n/2$, the late-time value of the $f_A^{\text{comp}}$ is exponentially suppressed in $n$. The late-time value is therefore effectively indistinguishable from the zero result we would obtain from the equilibrium density matrix $\rho^{(\text{eq})}$, or equivalently from Hawking's state in the black hole context. Moreover, the numerical result of Fig. 9 (top) shows that the decay with time is monotonic in this regime, which is again intuitive from the perspective of thermalization.

What about the regime where the subsystem is so large that $n_{\bar{A}} \leq \mathcal{O}(\log n)$? In the evaporating black hole setup, this corresponds to very late times in the evaporation process where the black hole entropy is $\mathcal{O}(\log G_N)$. In this regime, the CFI is no longer exponentially suppressed, and in particular, if we take $A$ to be the full system, the CFI is extensive in $n$ (see Fig. 9 (bottom)). This tells us that it is possible to estimate time with high accuracy ($\mathcal{O}(1/n)$ variance) using measurements of $\mathcal{O}(1)$ complexity if we have access to the full system. The number of measurement outcomes needed is also likely to be $\mathcal{O}(1)$ or at worst polynomial in $n$. See the next section for some evidence for this using explicit experiments. In fact, as we will discuss in the next section, we can further use this simple procedure to distinguish the full state $|\psi(t)\rangle$ of the system from a thermal density matrix such as $\rho^{(\text{eq})}$.[9]

---

[9]There are other known ways of distinguishing $\text{Tr}_{\bar{A}}[|\psi(t)\rangle\langle\psi(t)|]$ from $\text{Tr}_{\bar{A}}\rho^{(\text{eq})}$ with $\mathcal{O}(\text{poly}(n))$ sample and circuit complexity when $n_{\bar{A}} \leq \mathcal{O}(\log(n))$, such as the SWAP test [9,26,50] which makes use of the large difference in second Renyi entropy between these states. See another example of a test that works in this regime in Section 5 of [23].

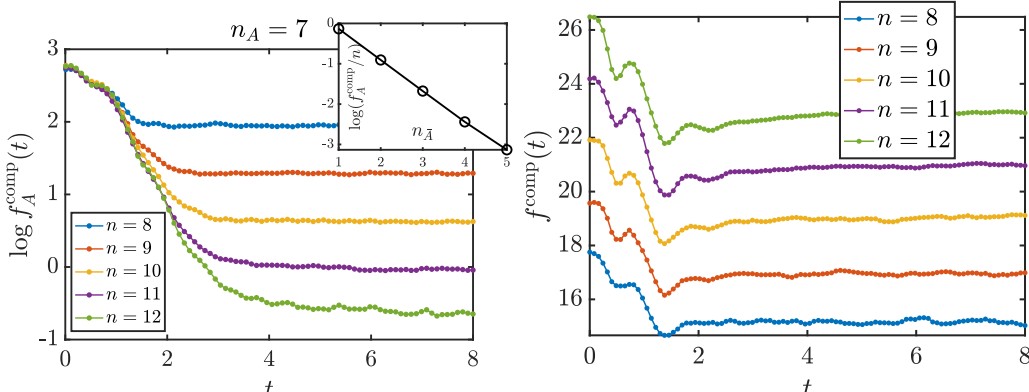

Figure 9: **Left:** Subsystem computational basis CFI $f_A^{\text{comp}}$ for $n_A = 7$ and different full system sizes, starting from intial random product states and averaged over 400 samples. We see that there is an exponential decay of the saturation value with $n_{\bar{A}}$ for fixed $n_A$. This should be contrasted with the late-time result in Fig. 3 (middle) for $F_A(t)$, the Fisher information corresponding to optimal measurements, which is almost independent of $n_{\bar{A}}$. **Left, inset:** We plot the late-time saturation value of $f_A^{\text{comp}}(t)$ as a function of $n_{\bar{A}}$, and the scaling matches (33). **Right:** Full system computational basis CFI $f^{\text{comp}}$ for different full system sizes. Both the initial and the final value are proportional to $n$.

Note that we have assumed here that evaluating an estimator function $T_{\text{est}}(\xi_1, \ldots, \xi_N)$ on the measurement outcomes which saturates the Cramer-Rao bound (3) is a simple process. In particular, suppose we use the maximum likelihood estimate (MLE) discussed around (4), and in more detail in the next section. Then we can evaluate $T_{\text{est}}$ efficiently if the form of $p_\xi(t) = \langle \xi | e^{-iHt} \sigma_0 e^{iHt} | \xi \rangle$ as a function of time for each $\xi = 1, \ldots, 2^n$ in the computational basis is given to us as input. We leave discussions of whether this additional resource is reasonable, and how complex it is to actually evaluate $p_\xi(t)$, to future work. For now, we note that even with this additional resource, if we are in the regime where $n_{\bar{A}} \geq \mathcal{O}(\log n)$, operations of $\mathcal{O}(1)$ circuit and sample complexity that we have considered here cannot be used to estimate $t$ or distinguish $|\psi(t)\rangle$ from $\rho^{(\text{eq})}$ at late times.

Regardless of the above somewhat intricate complexity-theoretic considerations, the large value of the computational basis CFI in the regime $n_{\bar{A}} \leq \mathcal{O}(\log n)$ indicates that the state $\text{Tr}_{\bar{A}}[|\psi(t)\rangle\langle\psi(t)|]$ is not thermalizing to a steady state in this regime, even from the perspective of simple measurements. It seems difficult to reconcile this statement with the fact that we expect (for instance based on the model of a Haar-random state) that the probability of any given measurement outcome for a simple measurement is exponentially close in $n$ between $|\psi(t)\rangle$ and $\rho^{(\text{eq})}$, even on the full system (see for instance [22, 23]). To better understand how these statements can be consistent, we note that there is an $\mathcal{O}(1)$ total variation distance (trace distance) between the classical probability distribution in the computational basis associated with the late-time $|\psi(t)\rangle$ (modelled as a Haar-random state) and the uniform distribution [51].[10] More explicitly, for the full system, we find that while the individual probability differences appearing in the trace distance are suppressed as $\mathcal{O}(e^{-n})$, the sum over exponentially many terms makes the trace distance $\mathcal{O}(1)$. Indeed, this result was previously used in [32] to deduce that the ability to estimate time using simple measurements on the full system is surprisingly good, using a different measure involving the classical mutual information. For a subsystem $A$, we find in Appendix F.2 that the total variation distance decays exponentially with $n_{\bar{A}}$. This confirms from a different perspective that the state $\text{Tr}_{\bar{A}}[|\psi(t)\rangle\langle\psi(t)|]$ equilibrates from the

---

[10]We present a self-contained derivation of this result using a resolvent technique in Appendix F.1.

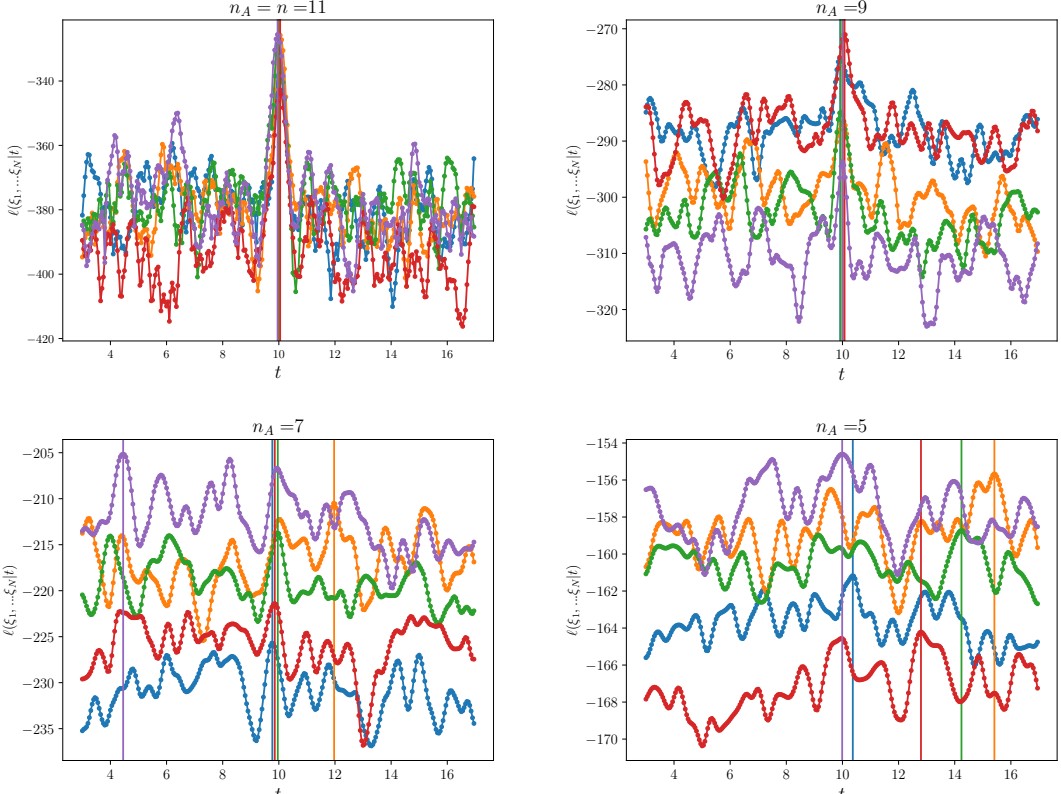

Figure 10: We time-evolve a single instance of an initial random product state by the chaotic spin chain Hamiltonian up to time $t_0 = 10$. The full system size is taken to be $n = 11$, and we consider different values of $n_A = 11, 9, 7, 5$. In each case, we draw $N = 50$ samples in the computational basis and plot the log-likelihood function (4) based on these (and repeat the process 5 times). We see that the estimate $T_{\text{ML}}$, indicated with the vertical lines corresponding to the global maxima, is much more reliable in cases with a larger value of the classical Fisher information $f_A^{\text{comp}}$. By considering similar data for other full system sizes $n = 8, 9, 10$, the $N$ needed for the estimate to become reliable does not appear to grow significantly with $n$: in particular, $N = 50$ used here is a small fraction of the 2048 states in the full Hilbert space in this case.

perspective of simple measurements in the regime $n_A \leq n - \mathcal{O}(\log n)$, but not necessarily for larger $n_A$.

# 5 An experiment on estimating time

The quantum and classical Fisher informations that we have calculated in various models in the previous sections indicate certain setups where we should in principle be able to accurately estimate $t_0$ using the time-evolved state $\rho(t_0)$. How would we carry out this process of time estimation in practice, using some set of measurement outcomes from the state? In this section, we will discuss an explicit estimator function called the maximum likelihood estimator (MLE), which is widely used in statistical inference. We will explicitly demonstrate its application for the task of time estimation in a quantum many-body system, by performing a numerical "experiment" in the spin chain model of Sec. 3.1.

Let us first recall the definition of the maximum likelihood estimate from Sec. 2. In our

setup, the family of states $\rho(t)$ is given by either $e^{-iHt} |\psi_0\rangle\langle\psi_0| e^{iHt}$ or $\mathrm{Tr}_{\bar{A}}[e^{-iHt} |\psi_0\rangle\langle\psi_0| e^{iHt}]$, and hence the measurement outcomes correspond to certain basis states $|\xi\rangle$ on either the full system or subsystem $A$. Given outcomes $\xi_1, \ldots, \xi_N$ from independent measurements on $N$ copies of the state $\rho(t_0)$, the maximum likelihood estimate $T_{\mathrm{ML}}(\xi_1,\ldots,\xi_N)$ is the value of $t$ that corresponds to the global maximum of the log-likelihood function $\ell(\xi_1,\ldots,\xi_N|t)$ defined in (4) over $t$. If the form of $p_\xi(t)$ as a function of $t$ for any outcome $|\xi\rangle$ is given to us as input, or can be computed numerically as we do in our "experiment" below, $T_{\mathrm{ML}}(\xi_1,\ldots,\xi_N)$ can be evaluated in a straightforward way. On general grounds [33], for large $N$, the mean of $T_{\mathrm{ML}}(\xi_1,\ldots,\xi_N)$ should approach the true underlying value $t_0$ of time, and its uncertainty $(\delta T_{\mathrm{ML}})^2$ defined in (2) should saturate the Cramer-Rao bound (3).

In a quantum many-body system of volume $n$, a complete measurement of any extensive subsystem has $\mathcal{O}(e^n)$ possible outcomes. A potential concern may be that the sample size $N$ needed for the Cramer-Rao bound to be saturated might itself grow rapidly with $n$. We will provide evidence using the spin chain experiment below that this is not the case, and individual realizations of $T_{\mathrm{ML}}$ for $\mathcal{O}(1)$ $N$ are very close to $t_0$ in cases where the Fisher information is large.

Let us carry out some explicit (numerical) experiments to estimate time. We take the initial state $|\psi_0\rangle$ to be some arbitrary single choice of a random product state on system size $n = 11$. We numerically sample from the probability distribution of either the full time-evolved state $e^{-iHt_0} |\psi_0\rangle$, or its reduced density matrix on a subsystem $A$, in the computational basis using python. We use the Hamiltonian of the chaotic spin chain (16). We obtain $N$ samples from this distribution, and plug these into (4) to obtain the log-likelihood function of $t$. In practice, we find this function by computing $p_{\xi_i}(t)$ for any given $\xi_i$ as a function of $t$ by numerically computing it for some finite set of times and interpolating. In Fig. 10, we show results from drawing $N = 50$ samples from the state at $t_0 = 10$ for various subsystem sizes, and repeating the experiment 5 times.

Recall from (33) that the late-time computational basis CFI $f_A^{\mathrm{comp}}$ is large for the full system, and rapidly decreases for subsystems. Consistent with this, in the case of the full system (top left of Fig. 10), we find a very sharp maximum in the log-likelihood function close to $t = t_0$ for each repetition of the experiment. As we decrease the subsystem size, the local maximum close to $t_0$ becomes less sharp and the variance of its location increases. Note that in all cases, the log-likelihood function has a large number of other local maxima besides the global maximum in the range of times we consider. An interesting feature which we could not have predicted *a priori* is that in cases with large CFI, the global maximum always seems to be very close to the actual value of $t_0$. In cases where $f_A^{\mathrm{comp}}$ is small, picking the global maximum yields unreliable results.

We can now see how such an experiment can be used to distinguish the time-evolved state from the equilibrium density matrix. Suppose we are given an $\mathcal{O}(1)$ number of copies of a state which is either $\rho_A(t) = \mathrm{Tr}_{\bar{A}}[e^{-iHt} |\psi_0\rangle\langle\psi_0| e^{iHt}]$ for some known $|\psi_0\rangle$ and $H$, or the equilibrium density matrix $\rho_A^{(\mathrm{eq})} = \mathrm{Tr}_{\bar{A}}[\rho^{(\mathrm{eq})}]$. We do not know which of the two states we have, but are told from the Born rule what the functions $p_\xi(t)$ for each $\xi$ would be as a function of $t$, if the state were $\rho_A(t)$. Suppose the computational basis CFI of $\rho_A(t)$ is large. Then one procedure that allows us to decide which state we have is to measure in the computational basis, and apply the MLE according to the state $\rho_A(t)$. We can repeat this experiment some $\mathcal{O}(1)$ number of times, as we do above in Fig. 10. If the variance in our resulting estimates of $t$ is small and decreasing as $1/N$ with the sample size $N$, this indicates that the state we are measuring is indeed $\rho_A(t)$. If not, the state is $\rho_A^{(\mathrm{eq})}$. This provides an efficient way of distinguishing $\rho_A(t)$ from $\rho_A^{(\mathrm{eq})}$. In cases where the CFI of $\rho(t)$ is exponentially small in $n$, the difference between $\rho_A^{(\mathrm{eq})}$ and $\rho_A(t)$ would likely become impossible to detect due to constraints of any reasonable experimental apparatus.

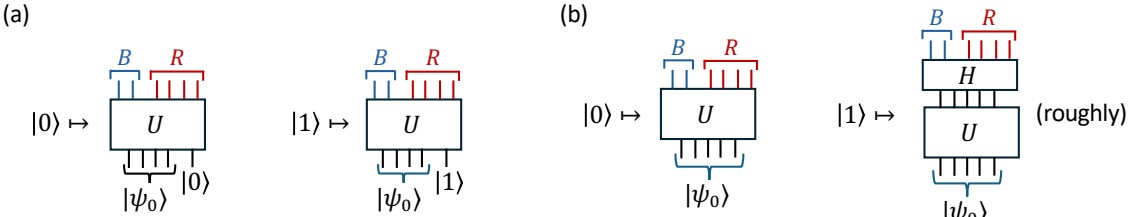

Figure 11: We compare the structure of the isometric encodings in the quantum error correcting code in the Hayden-Preskill protocol (a) and the one that appears naturally in the time estimation task (b). In both cases, $U$ is time-evolution operator up to some late-time $t$ in a chaotic system, which can for instance be modelled as a Haar-random unitary. (a) is a much more effective quantum error-correcting code than (b) as it makes use of the full scrambling power of the time-evolution operator to non-locally encode the information of the logical qubit.

# 6 Quantum error-correction perspective and comparison to Hayden-Preskill protocol

In this section, we discuss a further physical interpretation of the result (13) from Sec. 3 about the transition in the saturation value of the quantum Fisher information $F_A(t)$ as a function of subsystem size. We will interpret this transition from the perspective of quantum error-correction. The sudden improvement in our ability to estimate time using more than half of the system is reminiscent of the sudden improvement in the ability to recover information in the Hayden-Preskill protocol [12]. To compare the two tasks more directly, in this section we will analyse the question about inferring the value of time in the language of quantum error-correction.

The quantum error-correction task we consider was introduced in [30].[11] Let us motivate it as follows. The information of interest, which we want to encode and recover, is whether the time is $t$ or $t + dt$. Say that the logical $|0\rangle$ state corresponds to time $t$, and the logical $|1\rangle$ state to $t + dt$. Intuitively, this logical information is encoded in the quantum many-body system via the following map:

$$|0\rangle \to |\psi(t)\rangle \,, \qquad |1\rangle \to |\psi(t + dt)\rangle \,. \tag{34}$$

This map is not quite suitable for setting up a quantum error-correcting code, as it is not an isometry due to the non-orthogonality of $|\psi(t)\rangle$ and $|\psi(t + dt)\rangle$. Note that the normalized state along the component of $|\psi(t + dt)\rangle$ that is orthogonal to $|\psi(t)\rangle$ is $|\xi(t)\rangle$ defined in (9). Hence, a natural choice of an isometry to replace the map (34) is:

$$|0\rangle \mapsto |\psi(t)\rangle \,, \qquad |1\rangle \mapsto |\xi(t)\rangle \,. \tag{35}$$

This is an encoding of one logical qubit into the $n$ physical qubits of our quantum many-body system. Now suppose we want to correct for erasure errors on some subsystem $B$ (which represents the black hole in the black hole evaporation setup). Can we recover the logical information from the remaining subsystem $R$?

Faist et al [30] found a relation between the QFI and a weaker variant of the Knill-Laflamme conditions for the above error-correcting code as follows. In our notation, they showed that if the QFI of the subsystem $R$, $F_R(t)$, is equal to the QFI of the full system, then their Knill-Laflamme condition is exactly satisfied. Hence, if there is no sensitivity loss in metrology on

---

[11]Also see [52] for a related quantum error-correction task and its holographic description.

tracing out subsystems, the code is perfectly error-correctable (in the weaker sense of [30]). In the present physical setup, we find that the subsystem QFI $F_R(t)$ is always smaller than the full system QFI $F(t)$. This motivates us to ask about the approximate error-correcting properties of the code (35). Specifically, we would like to see if the error-correction works better in the regime where the subsystem QFI is extensive (even though it is not as large as the full system QFI), and poorly in the regime where the subsystem QFI is exponentially small.

A good test for error correctability of a logical qubit is to examine the distinguishability of two orthogonal codewords under noise. In particular, let us consider the late-time regime, in which $|\psi(t)\rangle$ can be modelled by $|\psi_{\text{Haar}}\rangle$ in (21), and $|\xi\rangle$ is still defined according to (9) for the particular choice our many-body Hamiltonian $H$. We would like to see how well $\rho_R \equiv \text{Tr}_B[|\psi\rangle\langle\psi|]$ and $\sigma_R \equiv \text{Tr}_B[|\xi\rangle\langle\xi|]$ can be distinguished.

We use Holevo's just-as-good fidelity $F_H(\rho, \sigma) \equiv \text{Tr}[\sqrt{\rho}\sqrt{\sigma}]$ as the distinguishability measure.[12] We find that the noisy codewords are indistinguishable for less than half of the system, and the fidelity improves linearly for more than half of the system (see Appendix G for details):

$$F_H(\rho_R, \sigma_R) \approx \begin{cases} 1, & d_R < d_B, \\ \frac{n_B}{n}, & d_R > d_B. \end{cases} \tag{36}$$

This result qualitatively matches the behavior (22) that we found for the QFI.

In the Hayden-Preskill protocol, the isometric encoding of the quantum error-correcting code is as shown in Fig. 11(a). In this case, the output states corresponding to any two orthogonal input states become perfectly distinguishable on any subsystem larger than half of the system, i.e., the information becomes perfectly recoverable just after the Page time. In (36), the states $\rho_R$ and $\sigma_R$ do not become perfectly distinguishable just after the Page time, but there is a sudden improvement in their distinguishability after the Page time: the fidelity drops from 1 to 1/2. Comparing the structure of the isometric encodings in the two codes in Fig. 11, we see that the two codes make use of the scrambling time-evolution operator in a different way, and in particular, the code in the Hayden-Preskill case uses the full power of the scrambling evolution to non-locally encode the logical information, while the code (35) does not.

# 7 Conclusions and discussion

In this paper, we used an operational question about time-estimation to probe new features of the dynamics of quantum many-body systems and black holes. We found an interesting interplay between thermalization and unitarity in the quantum Fisher information which determines the uncertainty of the optimal time estimate. For a small subsystem, we found a sharp distinction between free-fermion integrable and chaotic systems in terms of this quantity. Further, we understood its non-monotonic evolution for a subsystem larger than half of the system in terms of the rapid rotation of the support of the reduced density matrix in the full Hilbert space. We found a version of the black hole information loss paradox in terms of the failure of Hawking's naive calculation in semiclassical gravity to agree with general predictions for this quantity based on unitarity. We further studied how the behaviour of the classical Fisher information associated with simple measurements is consistent with expectations from coarse-grained properties of thermalization in chaotic many-body systems. Moreover, the Fisher information with this restriction on complexity also matches expectations based on semiclassical gravity in the black hole context.

---

[12]This measure is easier to evaluate than the fidelity, and it is "just as good" because it also satisfies the Fuchs–van de Graaf inequalities, $1 - F_H(\rho, \sigma) \leq \frac{1}{2}\|\rho - \sigma\|_1 \leq \sqrt{1 - F_H(\rho, \sigma)^2}$.

There are a number of interesting directions for future work related to general quantum many-body systems, black holes, and complexity theory, which we summarize in the three sections below.

## 7.1 Questions about quantum many-body systems

An interesting feature of the QFI for time estimation compared to other information-theoretic quantities that have been studied previously is that it involves the relation between the time-evolved state and the Hamiltonian in a crucial way. While we have focused on the class of effectively "infinite-temperature" initial pure states and emphasized what the QFI is telling us about their entanglement properties such as their Schmidt vectors, it is important to note that the QFI is not simply a measure of entanglement. For example, $F_A(t)$ is zero in any subsystem for any energy eigenstate of the system, although typical eigenstates are highly entangled. An important question for future work is to understand the behaviour of this quantity for initial states whose dynamics are more strongly constrained by energy conservation, such as initial states in a microcanonical window, and to understand the interplay between information and energy dynamics in this setting. The equilibrium approximation of [53] should provide a useful tool for addressing this question.

It would be interesting to see if some version of the QFI can be calculated in random models of local unitary chaotic systems such as random unitary circuits [54,55], which are analytically tractable for computing certain quantities such as the Renyi entropies. It would also be interesting to see if the QFI can be related to other quantities that capture universal aspects of chaos, such as OTOCs. A particularly natural set of quantities that may be related to the QFI would be the time-evolved circuit complexity [6] and Krylov complexity of the state [56], which naturally involve distance metrics between nearby states. It would be important to come up with a definition of mixed-state complexity that is appropriate for comparison to the subsystem QFI [57–59].

We found that the subsystem QFI is a useful way of detecting the difference between chaotic and free-fermion integrable systems, but behaves in a qualitatively similar manner between chaotic and interacting integrable systems. It would be interesting to understand whether the behavior of this quantity in interacting integrable systems can be understood in terms of quasiparticle interactions, in a similar way to the discussion for OTOCs in [60]. It would also be interesting to understand the behavior in many-body localized (MBL) systems [61], which exhibit integrability through a different mechanism. Other interesting cases would be ones that are intermediate between integrable and chaotic, such as systems with Hilbert space fragmentation and quantum many-body scars [62].

## 7.2 Questions about black holes

One main question we pose about black holes is how the prediction (32) that the subsystem QFI of the radiation is $\mathcal{O}(1)$ after the Page time can be reproduced with a gravity calculation. Recent results on obtaining the Page curve from gravity calculations [17–20] made use of the AdS/CFT correspondence, and in particular of prescriptions for calculating the entanglement entropy of the boundary theory using bulk quantities [63–66].

Finding a bulk dual of the quantum Fisher information in order to set up a similar calculation would require some further work. From (7), the QFI is related to the Bures distance, or equivalently the fidelity between the states at $t$ and $t + dt$. The fidelity is the exponent of the Renyi-$\frac{1}{2}$ relative entropy, which may not have a simple semiclassical gravity dual. von Neumann entropic quantities such as the relative entropy are better understood as having semiclassical gravity duals [67,68]. Hence, for the purpose of understanding QFI in the bulk, it may be more appropriate to study a variant of the QFI, known as the Boguliubov-Kubo-Mori

QFI [69,70], that is based on the gradient of the relative entropy. This quantity may provide a better starting point for exploring the physical phenomena we have identified through the QFI in gravity. From preliminary numerical studies, we have found that this quantity behaves in a qualitatively similar way to the QFI in quantum many-body systems.

Prior works [71–74] have explored the relevance of the Fisher information metric in gravity, but the dynamics of QFI associated with time estimation have not yet been analyzed. Assuming that one could find a bulk dual of the QFI or some other related quantity, another interesting setup to apply it to in gravity would be a quantum quench in the boundary CFT, which corresponds to black hole formation in the bulk [75–77]. In particular, it would be very interesting to see if such a calculation could reproduce the increasing behaviour of the larger subsystem QFI at intermediate times, see Fig. 1.

### 7.3 Questions about complexity

In the regime where $n/2 < n_A < n - \mathcal{O}(\log(n))$, it is often believed that it is exponentially hard to distinguish $\mathrm{Tr}_{\bar{A}}[|\psi(t)\rangle\langle\psi(t)|]$ from $\mathrm{Tr}_{\bar{A}}[\rho^{(\mathrm{eq})}]$. This exponential hardness has been particularly important in the evaporating black hole example, as it implies that observers with bounded computational complexity will not be able to detect that the fundamental description of quantum gravity is distinct from semiclassical effective field theory [21–26]. Our results imply that observers who have the ability to measure the QFI of $\rho_R$ after the Page time, and hence see that it has an extensive value instead of the zero value for $\rho_R^{(\mathrm{eq})}$, can tell the difference between semiclassical gravity and the fundamental description.

Is measuring the subsystem QFI in the regime $n/2 < n_A < n - \mathcal{O}(\log(n))$ provably an exponentially hard task? If so, is it hard because making the measurement in the optimal basis is itself exponentially complex, or because implementing the maximum likelihood estimate on the measurement outcomes is exponentially complex? Our results on entanglement entropy of the measurement basis (Fig. 8) lower-bound its computational complexity as $C \geq \mathcal{O}(n)$, but determining whether or not the complexity is exponential in $n$ would be an important question for future work.

We stress that the statement we want to make here is about the difficulty of distinguishing $\rho^{(\mathrm{eq})}$ from a known one-parameter family of states $\rho_A(t)$. This is likely a stronger statement than the one commonly studied in the complexity theory literature, where the state $\rho_A$ to be distinguished from $\rho^{(\mathrm{eq})}$ is randomly sampled from an ensemble [9,50,78]. While some hardness results are proved and argued in the case where $\rho_A$ is drawn from an ensemble [9,79], there is no known rigorous hardness result for our setting. Together with the evidence and arguments from [23,25,26,78], our results motivate a further investigation of this complexity-theoretic question that is vital to the validity of semiclassical physics.

In this paper, we have worked out the QFI for the optimal measurement basis and the CFI for the simple computational basis, and found that they show sharply different behaviors. It would also be interesting to calculate the CFI for other computationally efficient measurement bases, and see how much the CFI can improve by using a circuit of polynomial depth before measuring.

It would also be interesting to perform the experiment of Sec. 5 to estimate the time in an actual lab setting, making use of recent progress in simulating quantum many-body dynamics with a large number of degrees of freedom (see for instance [80]). This would allow us to check if the sample complexity for the time estimation task is indeed $\mathcal{O}(1)$ in a quantum many-body system. It is likely the hardest step in these cases may be to evaluate the classical probabilities $p_\xi(t)$ of Sec. 5 as a function of $t$ in order to perform the MLE. An important ingredient of this effort would then be to find an efficient way to evaluate these probabilities, or to see if it is provably difficult to do so.

## Acknowledgments

We thank Adam Bouland, Kedar Damle, Netta Engelhardt, Jeongwan Haah, Patrick Hayden, Michael Knap, Johannes Knolle, Nima Laskhari, Samir Mathur, Onkar Parrikar, Frank Pollmann, Rajdeep Sensharma, Al Shapere, Piyush Srivastava, Douglas Stanford, and Hong-Yi Wang for helpful discussions, and Soonwon Choi, Luca Delacretaz, and Henry Lin for comments on the draft.

**Funding information** Authors are required to provide funding information, including relevant agencies and grant JW would like to thank the support from AFOSR (award FA9550-19-1-0369). SV is supported by Google. HT is supported by the Simons Foundation and the Shoucheng Zhang Fellowship.

## A  Subsystem QFI in terms of Schmidt decomposition

In this appendix, we will further analyze the general formula for the subsystem quantum Fisher information $F_A(t)$ from (10) for the case where the global state $|\psi(t)\rangle$ is pure. Let us rewrite (10) here for the pure state case for clarity:

$$F_A(t) = \sum_{\substack{i,j \text{ s.t.} \\ p_i + p_j \neq 0}} \frac{2}{p_i + p_j} |\langle i| \mathrm{Tr}_{\bar{A}}[H, |\psi(t)\rangle\langle\psi(t)|] |j\rangle|^2, \tag{A.1}$$

where $p_i, |i\rangle$ are eigenvalues and eigenstates of $\mathrm{Tr}_{\bar{A}}[|\psi(t)\rangle\langle\psi(t)|]$. Recall that we have suppressed the $t$-dependence of $p_i, |i\rangle$ in the notation. We will further drop the $t$ label in $|\psi(t)\rangle$ for most of the remaining discussion to simplify notation. It is useful to divide (A.1) into the following terms:

$$F_A = 2F_{A,+} - F_{A,-} - (F_{A,-})^*, \tag{A.2}$$

where

$$F_{A,+} = \sum_{\substack{1 \leq j,k \leq \min(d_A, d_{\bar{A}}) \\ p_j + p_k \neq 0}} \frac{2}{p_j + p_k} \langle j| \mathrm{Tr}_{\bar{A}}[H |\psi\rangle\langle\psi|]|k\rangle \langle k| \mathrm{Tr}_{\bar{A}}[|\psi\rangle\langle\psi| H]|j\rangle, \tag{A.3}$$

$$F_{A,-} = \sum_{\substack{1 \leq j,k \leq \min(d_A, d_{\bar{A}}) \\ p_j + p_k \neq 0}} \frac{2}{p_j + p_k} \langle j| \mathrm{Tr}_{\bar{A}}[H |\psi\rangle\langle\psi|]|k\rangle \langle k| \mathrm{Tr}_{\bar{A}}[H |\psi\rangle\langle\psi|]|j\rangle. \tag{A.4}$$

To avoid repeatedly using $\min(d_A, d_{\bar{A}})$, let us refer to $S$ as the smaller subsystem and $B$ as its complement in the remaining discussion. The Schmidt decomposition of $|\psi\rangle$ can be expressed as:

$$|\psi\rangle = \sum_{a=1}^{d_S} \sqrt{p_a} |\psi_a\rangle_S |\phi_a\rangle_B. \tag{A.5}$$

We can take the Schmidt vectors $|\phi_a\rangle$, $a = 1, \ldots, d_S$ to be the first $d_S$ basis vectors of an orthonormal basis for subsystem $B$, and the remaining basis vectors can be labelled $|\phi_a\rangle$, $a = d_S + 1, \ldots, d_B$. These remaining $|\phi_a\rangle$'s are the zero eigenvectors of $\rho_B$, which should also be included in the sums (A.3) and (A.4) when we take $A = B$. We can also extend the set of $p_i$ to include $i$ from $i = d_S + 1$ to $i = d_B$, setting all $p_i$ in this range to zero. We will not put explicit $S$ and $B$ subscripts on the Schmidt vectors in the rest of the discussion: the $|\psi_a\rangle$'s always live in $S$ and $|\phi_a\rangle$'s in $B$.

Now consider the expression for $F_{S,+}$, taking $A = S$ in (A.3). Using the Schmidt decomposition (A.5), we can write

$$\langle\psi_j|\mathrm{Tr}_B[H|\psi\rangle\langle\psi|]|\psi_k\rangle = \sqrt{p_k}\,\langle\psi_j|\langle\phi_k|H|\psi\rangle\,. \tag{A.6}$$

Putting this into the expression for $F_{S,+}$, and assuming that all $p_i$ for $i$ from 1 to $d_S$ are non-zero, we find

$$F_{S,+} = \sum_{j=1}^{d_S}\sum_{k=1}^{d_S}\frac{2p_k}{p_j+p_k}\,\langle\psi_j|\langle\phi_k|H|\psi\rangle\langle\psi|H|\psi_j\rangle|\phi_k\rangle \tag{A.7}$$

$$= \sum_{j=1}^{d_S}\sum_{k=1}^{d_B}\frac{2p_k}{p_j+p_k}\,\langle\psi_j|\langle\phi_k|H|\psi\rangle\langle\psi|H|\psi_j\rangle|\phi_k\rangle\,, \tag{A.8}$$

where we have used that $p_k$ from $k = d_S + 1$ to $k = d_B$ are zero. We can similarly express $F_{B,+}$ in terms of the Schmidt coefficients and vectors:

$$F_{B,+} = \sum_{j=1}^{d_S}\sum_{k=1}^{d_B}\frac{2p_j}{p_j+p_k}\,\langle\psi_j|\langle\phi_k|H|\psi\rangle\langle\psi|H|\psi_j\rangle|\phi_k\rangle\,. \tag{A.9}$$

Putting (A.8) and (A.9) together, we therefore have

$$F_{S,+} + F_{B,+} = 2\sum_{j=1}^{d_S}\sum_{k=1}^{d_B}\langle\psi|H|\psi_j\rangle|\phi_k\rangle\langle\psi_j|\langle\phi_k|H|\psi\rangle = 2\langle\psi|H^2|\psi\rangle\,. \tag{A.10}$$

Note that for $|\psi\rangle = |\psi(t)\rangle = e^{-iHt}|\psi\rangle$, (A.10) is independent of $t$. Hence,

$$F_{S,+}(t) + F_{B,+}(t) = 2\langle\psi_0|H^2|\psi_0\rangle\,. \tag{A.11}$$

Next, consider $F_{A,-}$. Again by using the Schmidt decomposition of $|\psi\rangle$, one can check that

$$F_{S,-}(t) = F_{B,-}(t) = \sum_{j=1}^{d_S}\sum_{k=1}^{d_S}\frac{2\sqrt{p_j}\sqrt{p_k}}{p_j+p_k}\,\langle\psi|H|\psi_j\rangle|\phi_k\rangle\langle\psi|H|\psi_k\rangle|\phi_j\rangle\,. \tag{A.12}$$

We further notice that $F_S = F_{B,\mathrm{ent}}$ when $S$ and $B$ are maximally entangled in $|\psi(t)\rangle$. ($F_{B,\mathrm{ent}}$ is defined below (14).) To show this, we first note that:

$$\begin{aligned}
F_{B,\mathrm{ent}}(t) &= \sum_{\substack{i,j \text{ s.t.}\\ p_i\neq 0, p_j\neq 0}}\frac{2}{p_i+p_j}|\langle i|\mathrm{Tr}_S[H,|\psi(t)\rangle\langle\psi(t)|]|j\rangle|^2\\
&= \sum_{j=1}^{d_S}\sum_{k=1}^{d_S}\frac{2p_j}{p_j+p_k}\,\langle\psi_j|\langle\phi_k|H|\psi\rangle\langle\psi|H|\psi_j\rangle|\phi_k\rangle - F_{B,-} - F_{B,-}^*\,.
\end{aligned} \tag{A.13}$$

The first term of above equation only sums over non-zero eigenvalues, so it is different from $F_{B,+}$, but similar to $F_{S,+}$ (see (A.7) for comparison). In particular, when the spectrum is flat ($p_k = p_j$), the first term equals $F_{S,+}$. Then using the fact that $F_{B,-} = F_{S,-}$, (A.13) becomes equal to $F_S$.

(A.7), (A.11), and (A.12) will be useful for our later calculations. The structure of (A.11) and (A.12) is interesting. While the term $F_{A,-}$ is equal for subsystems $A = S$ and $A = B$ (similar to quantities like entanglement entropy) at all times, $F_{A,+}$ suggests that there is some tradeoff between $S$ and $B$ in the time-evolution.

Consistent with the idea of such a tradeoff, we note as an interesting aside that (A.11) and (A.12) together are equivalent to a certain "time-energy uncertainty relation" found in [30]. This is an uncertainty relation that quantifies a trade-off between the fluctuations in time measurements and the fluctuations in energy measurements. We briefly review it in the remaining part of this section.

Any uncertainty relation involves a pair of conjugate variables. In [30], the authors identified a variable $\eta$ conjugate to $t$ that is generated by the optimal local time measurement operator $T$. As explained in [30], the theory of quantum fisher information gives us a formula for the optimal operator $T$ that measures the time close to $t$ and minimizes the variance of the time estimation on the full system,

$$T = t + \frac{1}{F(t)} \sum_{i,j} \frac{2}{p_i + p_j} \langle i | [H, |\psi\rangle\langle\psi|] | j \rangle \, |i\rangle\langle j| \, . \tag{A.14}$$

Consider now a conjugate variable $\eta$ that is defined via an evolution generated by $T$,

$$\partial_\eta |\psi(\eta)\rangle\langle\psi| = i[T, |\psi\rangle\langle\psi|] \, . \tag{A.15}$$

We consider infinitesimal flows in the $\eta$ direction, so the state $|\psi\rangle\langle\psi|$ appearing on the RHS of (A.15) is the same as the one that appears in (A.14). It is shown that the optimal operator to measure $\eta$, analogous to the operator (A.14) for measuring $t$, simply equals the Hamiltonian. Hence, the parameter $\eta$ can be seen as the total energy of the system.

Now we consider the problem of estimating the total energy of the system by performing the measurement on a subsystem. Let $\rho_{A\bar{A}}(t, \eta)$ be the pure state on $A\bar{A}$ (we return to the notation $A\bar{A}$ to indicate that $A$ can be either the smaller or the larger subsystem). We are interested in the variances of estimations of $t$ and $\eta$ on subsystems $A$ and $\bar{A}$ respectively. We denote corresponding the QFI of the full system by $F(t)$ and $F(\eta)$ respectively. [30] first show that on the full system, the QFI $F(\eta)$ of the state under the flow by (A.15) is determined by the QFI $F(t)$ under the flow of $t$ as follows:

$$F(\eta) = 4/F(t) = 1/\left( \langle H^2 \rangle_{\psi_0} - \langle H \rangle_{\psi_0}^2 \right) \, . \tag{A.16}$$

Next, just as we can consider the flow by $t$ for the reduced density matrices of a subsystem, we can do the same for the flow by $\eta$ and consider the subsystem QFI $F_{\bar{A}}(\eta)$. [30] then find the following time-energy uncertainty relation for subsystems:

$$\frac{F_A(t)}{F(t)} + \frac{F_{\bar{A}}(\eta)}{F(\eta)} = 1 \, , \tag{A.17}$$

where using (A.15), $F_{\bar{A}}(\eta)$ is given by the following formula:

$$F_{\bar{A}}(\eta) = \frac{1}{4\left( \langle H^2 \rangle_{\psi_0} - \langle H \rangle_{\psi_0}^2 \right)} \sum_{\substack{i,j \text{ s.t.} \\ p_i+p_j\neq 0}} \frac{2}{p_i + p_j} |\langle i| \mathrm{Tr}_S \left\{ H - \langle H \rangle_{\psi_0}, |\psi\rangle\langle\psi| \right\} |j\rangle|^2$$

$$= \frac{4}{F(t)^2} \left( 2F_{\bar{A},+}(t) + F_{\bar{A},-}(t) + F_{\bar{A},-}(t)^* \right) \, , \tag{A.18}$$

where the second line follows from the same decomposition of (A.2) after changing the commutator to the anticommutator.

Note that the above equation involves an anticommutator inside the trace. The result (A.17) says that the better one can estimate the evolution time $t$ of the full system from a subsystem, the worse one can estimate the energy $\eta$ of the full system from the complementary subsystem, and vice versa.

Let us now show the uncertainty relation (A.17) is equivalent to our (A.11) and (A.12). For simplicity, we assume w.l.o.g. that $\langle H \rangle_{\psi_0} = 0$. Then (A.11) is equivalent to

$$2F_{A,+}(t) + 2F_{\bar{A},+}(t) = F(t), \tag{A.19}$$

and (A.18) is equivalent to

$$F_{\bar{A}}(\eta) = \frac{1}{4 \langle H^2 \rangle_{\psi_0}} \sum_{\substack{i,j \text{ s.t.} \\ p_i + p_j \neq 0}} \frac{2}{p_i + p_j} | \langle i | \text{Tr}_S \{ H, |\psi\rangle\langle\psi| \} | j \rangle |^2$$

$$= \frac{4}{F(t)^2} \left( 2F_{\bar{A},+}(t) + F_{\bar{A},-}(t) + F_{\bar{A},-}(t)^* \right). \tag{A.20}$$

Adding and subtracting $F_{A,-}(t)$ gives

$$2F_{A,+}(t) - F_{A,-}(t) - F_{A,-}(t)^* + 2F_{\bar{A},+}(t) + F_{A,-}(t) + F_{A,-}(t)^* = F(t). \tag{A.21}$$

Using (A.12),

$$2F_{A,+}(t) - F_{A,-}(t) - F_{A,-}(t)^* + 2F_{\bar{A},+}(t) + F_{\bar{A},-}(t) + F_{\bar{A},-}(t)^* = F(t). \tag{A.22}$$

Dividing both sides by $F(t)$, we have

$$\frac{F_A(t)}{F(t)} + \frac{2F_{\bar{A},+}(t) + F_{\bar{A},-}(t) + F_{\bar{A},-}(t)^*}{F(t)} = \frac{F_A(t)}{F(t)} + \frac{2F_{\bar{A},+}(t) + F_{\bar{A},-}(t) + F_{\bar{A},-}(t)^*}{F(t)^2 F(\eta)/4}$$

$$= \frac{F_A(t)}{F(t)} + \frac{F_{\bar{A}}(\eta)}{F(\eta)} = 1, \tag{A.23}$$

where the first equality uses $F(t)F(\eta) = 4$ and the second equality uses (A.20). This is the uncertainty relation (A.17).

# B Subsystem QFI for random pure states

In this Appendix, we will give a derivation for (22). We will always use $S$ to denote the subsystem smaller than half of the full system, and $B$ to denote its complement. Recall the Schmidt decomposition of a random pure state from (21), which we repeat here for convenience:

$$|\psi\rangle = \sum_{i=1}^{d_S} \sqrt{p_i} (V |i\rangle)_S (U |\tilde{i}\rangle)_B. \tag{B.1}$$

We take the $d_S$ real numbers $p_i$, the $d_S \times d_S$ unitary $V$, and the the $d_B \times d_B$ unitary $U$ to be random and uncorrelated with each other. $U$ and $V$ are both Haar-random unitaries in their respective Hilbert spaces. $\{|i\rangle\}$ and $\{|\tilde{i}\rangle\}$ are arbitrary fixed sets of $d_S$ orthonormal states in $S$ and $B$ respectively. $p_i$ have the statistics of the eigenvalues of normalized Wishart matrices $\frac{YY^\dagger}{\text{Tr}[YY^\dagger]}$, where Y is a $d_S \times d_B$ matrix of independent complex Gaussian random variables drawn from the distribution $p(Y) = \mathcal{N}^{-1} e^{-d_B \text{Tr}[YY^\dagger]}$. In this appendix, we will use overlines to indicate all averages.

Let us first evaluate $F_{S,+}$ using (A.7). For any given realization of (B.1), we have

$$F_{S,+} = \sum_{j,k,a,b=1}^{d_S} \frac{2p_k}{p_j + p_k} \sqrt{p_a} \sqrt{p_b} \langle \psi_j | \langle \phi_k | H |\psi_a\rangle |\phi_a\rangle \langle \psi_b | \langle \phi_b | H |\psi_j\rangle |\phi_k\rangle$$

$$= \sum_{j,k,a,b=1}^{d_S} \frac{2p_k \sqrt{p_a p_b}}{p_j + p_k} \sum_{\substack{\alpha_1, \alpha_2, \\ \alpha_3, \alpha_4 = 1}}^{d_S} \sum_{\substack{\beta_1, \beta_2, \\ \beta_3, \beta_4 = 1}}^{d_B} \langle \alpha_1 | \langle \beta_1 | H |\alpha_2\rangle |\beta_2\rangle \langle \alpha_3 | \langle \beta_3 | H |\alpha_4\rangle |\beta_4\rangle \tag{B.2}$$

$$\times V_{j\alpha_1} V_{a\alpha_2}^* V_{b\alpha_3} V_{j\alpha_4}^* U_{k\beta_1} U_{a\beta_2}^* U_{b\beta_3} U_{k\beta_4}^*.$$

We have the following rules for the Haar averages:

$$\overline{V_{j\alpha_1}V_{a\alpha_2}^*V_{b\alpha_3}V_{j\alpha_4}^*} = \frac{1}{d_S^2-1}\left(\delta_{aj}\delta_{jb}\delta_{\alpha_1\alpha_2}\delta_{\alpha_3\alpha_4} + \delta_{jj}\delta_{ab}\delta_{\alpha_1\alpha_4}\delta_{\alpha_3\alpha_2}\right.$$
$$\left. -\frac{1}{d_S}\left(\delta_{aj}\delta_{jb}\delta_{\alpha_1\alpha_4}\delta_{\alpha_2\alpha_3} + \delta_{jj}\delta_{ab}\delta_{\alpha_1\alpha_2}\delta_{\alpha_3\alpha_4}\right)\right), \tag{B.3}$$

$$\overline{U_{k\beta_1}U_{a\beta_2}^*U_{b\beta_3}U_{k\beta_4}^*} = \frac{1}{d_B^2-1}\left(\delta_{ka}\delta_{kb}\delta_{\beta_1\beta_2}\delta_{\beta_3\beta_4} + \delta_{kk}\delta_{ab}\delta_{\beta_1\beta_4}\delta_{\beta_3\beta_2}\right.$$
$$\left. -\frac{1}{d_B}\left(\delta_{ka}\delta_{kb}\delta_{\beta_1\beta_4}\delta_{\beta_3\beta_2} + \delta_{kk}\delta_{ab}\delta_{\beta_1\beta_2}\delta_{\beta_3\beta_4}\right)\right). \tag{B.4}$$

For $F_{S,-}$, we have

$$F_{S,-} = \sum_{j,k,a,b=1}^{d_S} \frac{2\sqrt{p_k}\sqrt{p_j}}{p_j+p_k}\sqrt{p_a}\sqrt{p_b}\,\langle\psi_a|\langle\phi_a|H|\psi_k\rangle|\phi_j\rangle\langle\psi_b|\langle\phi_b|H|\psi_j\rangle|\phi_k\rangle$$
$$= \sum_{j,k,a,b=1}^{d_S} \frac{2\sqrt{p_kp_jp_ap_b}}{p_j+p_k}\sum_{\substack{\alpha_1,\alpha_2,\\\alpha_3,\alpha_4=1}}^{d_S}\sum_{\substack{\beta_1,\beta_2,\\\beta_3,\beta_4=1}}^{d_B}\langle\alpha_2|\langle\beta_2|H|\alpha_1\rangle|\beta_1\rangle\langle\alpha_3|\langle\beta_3|H|\alpha_4\rangle|\beta_4\rangle \tag{B.5}$$
$$\times V_{a\alpha_2}V_{k\alpha_1}^*V_{b\alpha_3}V_{j\alpha_4}^*U_{a\beta_2}U_{j\beta_1}^*U_{b\beta_3}U_{k\beta_4}^*.$$

Using the averages over $U$ and $V$, we find

$$\overline{F_{S,+}} = \frac{\mathrm{Tr}[H^2]}{(d_S^2-1)(d_B^2-1)}\left[(d_S^{-1}-d_B^{-1})\frac{I}{4} + (d_S^2-1)(1-d_S^{-1}d_B^{-1})\right]$$
$$+ \frac{\mathrm{Tr}[H]^2}{(d_S^2-1)(d_B^2-1)}\left[(-d_S^{-1}+d_B^{-1})\frac{I}{4}\right]$$
$$+ \frac{\mathrm{Tr}_B[(\mathrm{Tr}_S H)^2]}{(d_S^2-1)(d_B^2-1)}\left[(-1+d_S^{-1}d_B^{-1})\frac{I}{4}\right] \tag{B.6}$$
$$+ \frac{\mathrm{Tr}_S[(\mathrm{Tr}_B H)^2]}{(d_S^2-1)(d_B^2-1)}\left[(1-d_S^{-1}d_B^{-1})\frac{I}{4} + (d_S^2-1)(d_S^{-1}-d_B^{-1})\right],$$

and

$$\overline{F_{S,-}} = \frac{\mathrm{Tr}[H^2]}{(d_S^2-1)(d_B^2-1)}\left[(d_S^{-1}+d_B^{-1})\frac{I}{4} + (d_S^2-1)(-d_S^{-1}d_B^{-1})\right]$$
$$+ \frac{\mathrm{Tr}[H]^2}{(d_S^2-1)(d_B^2-1)}\left[(d_S^{-1}+d_B^{-1})\frac{I}{4} + (d_S^2-1)(-d_S^{-1}d_B^{-1})\right]$$
$$+ \frac{\mathrm{Tr}_B[(\mathrm{Tr}_S H)^2]}{(d_S^2-1)(d_B^2-1)}\left[(-1-d_S^{-1}d_B^{-1})\frac{I}{4} + (d_S^2-1)(d_S^{-1})\right] \tag{B.7}$$
$$+ \frac{\mathrm{Tr}_S[(\mathrm{Tr}_B H)^2]}{(d_S^2-1)(d_B^2-1)}\left[(-1-d_S^{-1}d_B^{-1})\frac{I}{4} + (d_S^2-1)(d_S^{-1})\right],$$

where

$$I = \overline{\sum_{j,k=1}^{d_S}\frac{2(p_j-p_k)^2}{p_j+p_k}}. \tag{B.8}$$

Altogether, using $\overline{F_S} = 2\overline{F_{S,+}} - 2\overline{F_{S,-}}$, we obtain:

$$\overline{F_S} = \frac{2}{(d_B^2-1)}\left(\text{Tr}[H^2] + \frac{1}{d_S d_B}\text{Tr}[H]^2 - \frac{1}{d_S}\text{Tr}_B[(\text{Tr}_S H)^2] - \frac{1}{d_B}\text{Tr}_S[(\text{Tr}_B H)^2]\right) \tag{B.9}$$

$$+ \frac{I}{(d_S^2-1)(d_B^2-1)}\left(-\frac{1}{d_B}\text{Tr}[H^2] - \frac{1}{d_S}\text{Tr}[H]^2 + \frac{1}{d_S d_B}\text{Tr}_B[(\text{Tr}_S H)^2] + \text{Tr}_S[(\text{Tr}_B H)^2]\right) \tag{B.10}$$

$$\equiv F_S^{\text{flat}} + F_S^{\text{non-flat}}, \tag{B.11}$$

where $F_S^{\text{flat}}$ and $F_S^{\text{non-flat}}$ are defined in (B.9) and (B.10) respectively. The motivation for these definitions is that if we assume that the Schmidt spectrum is flat ($p_j = d_S^{-1}$ for each $j$), we have $F_S^{\text{non-flat}} = 0$ and $\overline{F_S} = F_S^{\text{flat}}$ as a consequence of $I = 0$.

Now, we want to estimate the magnitude of $F_S$ in the limit $d_B \gg d_S \gg 1$. We write the full Hamiltonian as

$$H = H_S + H_B + H_{\text{int}}, \tag{B.12}$$

where $H_S$ and $H_B$ include all terms contained within $S$ and $B$ respectively.

We first estimate the magnitude of $F_S^{\text{flat}}$. We notice that if $H_{\text{int}} = 0$, we have precisely $F_S^{\text{flat}} = 0$. Collecting the terms that involve $H_{\text{int}}$, we find:

$$F_S^{\text{flat}} = \frac{1}{d_B^2-1}\left(\text{Tr}[H_{\text{int}}^2] + \frac{1}{d_S d_B}(\text{Tr}[H_{\text{int}}])^2 - \frac{1}{d_S}\text{Tr}_B(\text{Tr}_S H_{\text{int}})^2 - \frac{1}{d_B}\text{Tr}_S(\text{Tr}_B H_{\text{int}})^2\right)$$

$$\sim \frac{d_S^2}{d} \times \mathcal{O}(|\partial S|), \tag{B.13}$$

where $|\partial S|$ is the number of degrees of freedom involved in $H_{\text{int}}$, and is proportional to the area of the boundary of $S$ for a local Hamiltonian. Next, we estimate the magnitude of $F_S^{\text{non-flat}}$, by first evaluating integral $I$. Define $\mu(x)$ as the normalized density distribution of the Schmidt spectrum:

$$\mu(x) = \frac{1}{d_S}\sum_{i=1}^{d_S}\delta(x - p_i). \tag{B.14}$$

Then $I$ can be expressed as follows:

$$I = d_S^2 \int \mathrm{d}x \mathrm{d}y \langle\mu(x)\mu(y)\rangle \frac{2(x-y)^2}{x+y}. \tag{B.15}$$

Here we can decompose the correlators $\langle\mu(x)\mu(y)\rangle$ into disconnected and connected parts. The disconnected part is built from $\langle\mu(x)\rangle$, which is the Marchenko-Pastur distribution and is approximately $\delta(x - 1/d_S)$ up to a small width of $\mathcal{O}(1/\sqrt{d})$. Since we are in the limit $d_B \gg d_S \gg 1$, we can safely ignore the level repulsion of the eigenvalues (which gives the connected part of the correlator), and approximate $\langle\mu(x)\mu(y)\rangle \approx \langle\mu(x)\rangle\langle\mu(y)\rangle$. The average density of the MP distribution is given by:

$$\langle\mu(x)\rangle = \frac{d_B}{2\pi}x^{-1}\sqrt{(x-x_-)(x_+-x)}, \qquad x_- < x < x_+, \tag{B.16}$$

with the edge of the spectrum given by:

$$x_\pm = d_S^{-1}\left(1 \pm d_S^{1/2}d_B^{-1/2}\right)^2 \approx d_S^{-1} \pm 2d^{-1/2}. \tag{B.17}$$

In the limit $d_B \gg d_S \gg 1$, the MP distribution is a semi-circle centered at $x_c \equiv d_S^{-1}$ with width $\Delta \equiv 2d^{-1/2}$. Since the width is small, we can make the approximation $\frac{2(x-y)^2}{xy(x+y)} \approx \frac{2(x-y)^2}{2x_c^3}$. We

then obtain:

$$\begin{aligned}
I &\approx d_S^2 \left(\frac{d_B}{2\pi}\right)^2 \frac{1}{2(d_S^{-1})^3} \times \int_{-\Delta}^{\Delta} dx dy\, 2(x-y)^2 \sqrt{(\Delta^2 - x^2)(\Delta^2 - y^2)} \\
&= d_S^2 \left(\frac{d_B}{2\pi}\right)^2 \frac{1}{2(d_S^{-1})^3} \times \frac{\pi^2}{4}\Delta^6 \\
&= 2d_S^2/d_B\,.
\end{aligned} \tag{B.18}$$

Now we are ready to estimate $F_S^{\text{non-flat}}$. First notice that in contrast to $F_S^{\text{flat}}$, $F_S^{\text{non-flat}}$ does not equal zero when $H_{\text{int}} = 0$. So to calculate the leading order contribution of $F_S^{\text{non-flat}}$, we can safely take $H_{\text{int}} = 0$. In this case, we obtain:

$$F_S^{\text{non-flat}} = \frac{\left(\text{Tr}_S[H_S^2] - \frac{1}{d_S}(\text{Tr}_S H_S)^2\right)}{d_S^2 - 1} I \approx 2d_B^{-1}\left(\text{Tr}_S[H_S^2] - \frac{1}{d_S}(\text{Tr}_S H_S)^2\right) \sim \frac{d_S^2}{d} \times \mathcal{O}(n_S). \tag{B.19}$$

Comparing (B.13) and (B.19), we find that they have the same $\sim \frac{d_S^2}{d}$ leading scaling, but with coefficients of the exponential part that are proportional to the area and volume of $S$ respectively. Since we have a local Hamiltonian $H$, we should have $|\partial S| \ll n_S$. So the leading contribution to $F_S$ comes from $F_S^{\text{non-flat}}$:

$$F_S \approx F_S^{\text{non-flat}} \approx 2d_B^{-1}\left(\text{Tr}_S[H_S^2] - \frac{1}{d_S}(\text{Tr}_S H_S)^2\right) \sim \frac{d_S^2}{d} \times \mathcal{O}(n_S). \tag{B.20}$$

Next, let us evaluate $\overline{F_B}$ for the larger subsystem $B$. We use the relations (A.10) and (A.12) to get

$$\overline{F_{B,+}} = 2\overline{\langle\psi|H^2|\psi\rangle} - \overline{F_{S,+}} = \frac{2}{d}\text{Tr}[H^2] - \overline{F_{S,+}}, \qquad \overline{F_{B,-}} = \overline{F_{S,-}}. \tag{B.21}$$

So, we obtain:

$$\overline{F_B} = \frac{4}{d}\text{Tr}[H^2] - 2\overline{F_{S,+}} - 2\overline{F_{S,-}} \equiv F_B^{\text{flat}} + F_B^{\text{non-flat}}, \tag{B.22}$$

where like in (B.11), $F_B^{\text{non-flat}}$ is the part of $F_B$ proportional to $I$ and $F_B^{\text{flat}}$ is the remaining contribution.

First, we estimate the magnitude of $F_S^{\text{flat}}$. Up to exponentially small contributions, we find

$$F_B^{\text{flat}} = 4\left(\frac{\text{Tr}_B(H_B^2)}{d_B} - \frac{(\text{Tr}_B H_B)^2}{d_B^2}\right) \sim n_B\,. \tag{B.23}$$

Next, we estimate the magnitude of $F_B^{\text{non-flat}}$. $F_B^{\text{non-flat}}$ is given by:

$$F_B^{\text{non-flat}} = \frac{I}{(d_S^2 - 1)(d_B^2 - 1)}\left(-\frac{1}{d_S}\text{Tr}[H^2] - \frac{1}{d_B}\text{Tr}[H]^2 + \text{Tr}_B[(\text{Tr}_S H)^2] + \frac{1}{d_S d_B}\text{Tr}_S[(\text{Tr}_B H)^2]\right). \tag{B.24}$$

In the limit $d_B \gg d_S \gg 1$ and for a local Hamiltonian, we can drop the $H_{\text{int}}$ term, and obtain:

$$F_B^{\text{non-flat}} \approx \frac{\text{Tr}_B[H_B^2]}{d_B^2 - 1} I \sim \frac{d_S^2}{d_B^2} \times \mathcal{O}(n_B). \tag{B.25}$$

Comparing with (B.23), we find that $F_B^{\text{non-flat}}$ is exponentially suppressed. As a result, we obtain the magnitude of $\overline{F_B}$ as:

$$\overline{F_B} \approx F_B^{\text{flat}} \approx 4\left(\frac{\text{Tr}_B(H_B^2)}{d_B} - \frac{(\text{Tr}_B H_B)^2}{d_B^2}\right) \sim n_B\,. \tag{B.26}$$

(B.20) and (B.26) together give our result in (22). We note that up to the volume-law vs. area law behaviour of the coefficient of the exponentially small value of $F_S$, the same result could have been obtained using the following models with a flat entanglement spectrum (as long as $H_{\text{int}} \neq 0$):

$$|\psi_{\text{flat}}\rangle = \sum_{i=1}^{d_S} d_S^{-1/2} V |i\rangle_S \otimes U |\tilde{i}\rangle_B\,, \tag{B.27}$$

or more simply,

$$|\psi'_{\text{flat}}\rangle = \sum_{i=1}^{d_S} d_S^{-1/2} |i\rangle_S \otimes U |\tilde{i}\rangle_B\,, \tag{B.28}$$

where $\{|i\rangle_S\}$ is an arbitrary basis for $S$, $\{|\tilde{i}\rangle_B\}$ is an arbitrary set of $d_S$ orthonormal states in $B$, and $U, V$ are Haar-random unitaries. We will sometimes use these models to simplify later calculations.

## C  A Brownian GUE toy model for the late-time state

This Appendix is motivated by our observation in Fig. 5 that the key reason for the growth of the QFI $F_B(t)$ at late times is the rotation of the support of $\rho_B(t)$ in $\mathcal{H}_B$. (Like in previous appendices, we will use $S$ to denote the smaller subsystem and $B$ to denote the larger subsystem.) Why does the speed of rotation, $F_{B,\text{rot}}$ from (15), grow with time? Intuitively, this result seems to be telling us that as the eigenbasis of $\rho_B$ becomes more and more complicated and approaches a random subspace of $\mathcal{H}_B$, it is also able to change faster as a function of time on being slightly perturbed by the Hamiltonian. To check this intuition, we consider a model for the time-evolved state where by construction, the eigenbasis of $\rho_B$ is becoming increasingly complicated with time:

$$|\psi(t)\rangle = \sum_{i=1}^{d_S} d_S^{-1/2} V |i\rangle_S \otimes U_{\text{aux}}(t) |\tilde{\psi}_i\rangle_B\,, \tag{C.1}$$

where $V$ is a Haar-random unitary, and $U_{\text{aux}}(t)$ is the time-evolution operator associated with a time-dependent Hamiltonian with Brownian type disorder:

$$U_{\text{aux}}(t) = \mathcal{P} \exp\left[ i \int_0^t dt' H_{\text{aux}}(t') \right]\,, \tag{C.2}$$

$$\overline{H_{\text{aux},ij}(t) H_{\text{aux},kl}(t')} = d_B^{-1} \delta_{il} \delta_{jk} \delta(t - t')\,, \qquad i, j, k, l = 1, \ldots, d_B\,. \tag{C.3}$$

Here, $H_{\text{aux}}(t)$ is a $d_B \times d_B$ matrix. This model, called Brownian Gaussian Unitary Ensemble (BGUE), was developed and extensively studied in [39, 40]. Since the dynamics of the eigenvalues of $\rho_{S,B}$ do not play an important role in the phenomenon we are interested in, we set them all equal to $1/d_S$, corresponding to the maximally mixed state on $S$.

Our goal is to understand explicitly whether, as the eigenbasis of $\rho_B$ becomes more and more complicated in this model, it also evolves more rapidly on perturbation by a fixed local chaotic Hamiltonian $H$ like that of the spin chain (16). We therefore evaluate the subsystem QFI of this time-evolved state (C.1) on further infinitesimal evolution by the original Hamiltonian $H$. Hence, in the formula (10), we replace $e^{-iHt} \sigma_0 e^{iHt}$ in (10) with $|\psi(t)\rangle \langle \psi(t)|$, and replace $p_i, |i\rangle$ with the eigenvalues and eigenvalues of $\text{Tr}_S[|\psi(t)\rangle \langle \psi(t)|]$, but use the Hamiltonian $H$ from (16).

The advantage of using the model (C.1) is that it is analytically tractable. The averaged $F_A(t)$ of (C.1) only involves the first and second moment of the $U_{\text{aux}}(t)$ ensemble:

$$\overline{U_{\text{aux}}(t) \otimes U_{\text{aux}}^*(t)}\,, \qquad \overline{U_{\text{aux}}(t) \otimes U_{\text{aux}}(t) \otimes U_{\text{aux}}^*(t) \otimes U_{\text{aux}}^*(t)}\,, \tag{C.4}$$

which is exactly solvable with analytical results [40]. Note that the ensemble of BGUE saturates to the Haar random ensemble at $t \to +\infty$ [40], giving us the results from $|\psi_{\text{flat}}\rangle$ defined in (B.27) in the $t \to \infty$ limit. This model is too simple to incorporate any effects of locality, but will allow us to see that even without locality, it takes some $O(1)$ amount of time for the speed of rotation of $\rho_B(t)$ to reach its saturation value.

Having understood the motivations for the BGUE toy model, let us specify what to calculate. We eventually want to obtain $\overline{F_S(t)}, \overline{F_B(t)}, \overline{F_{B,\text{ent}}(t)}, \overline{F_{B,\text{rot}}(t)}$. Using conclusions from Appendix A, we have:

$$\overline{F_S(t)} = 2\overline{F_{S,+}(t)} - 2\overline{F_{S,-}(t)}, \qquad \overline{F_B(t)} = 2\overline{F_{B,+}(t)} - 2\overline{F_{B,-}(t)},$$
$$\text{with:} \quad 2\overline{F_{S,+}(t)} + 2\overline{F_{B,+}(t)} = 4\overline{\langle\psi(t)|H^2|\psi(t)\rangle}, \qquad \overline{F_{S,-}(t)} = \overline{F_{B,-}(t)}, \tag{C.5}$$

where we used the fact that $\overline{F_{S,-}(t)}, \overline{F_{B,-}(t)}$ are real numbers. Specifying the Schmit spectrum to be maximally mixed, we further obtain additional constraints:

$$2\overline{F_{B,+}(t)} = \overline{F_{B,\text{rot}}(t)} + 2\overline{F_{S,+}(t)}. \tag{C.6}$$

Therefore, we only need to calculate three independent variables

$$\overline{F_{S,+}(t)}, \qquad \overline{F_{S,-}(t)}, \qquad \overline{\langle\psi(t)|H^2|\psi(t)\rangle},$$

and then we can obtain everything else:

$$\overline{F_{B,\text{ent}}(t)} = \overline{F_S(t)} = 2\overline{F_{S,+}(t)} - 2\overline{F_{S,-}(t)}, \qquad \overline{F_{B,\text{rot}}} = 4\overline{\langle\psi(t)|H^2|\psi(t)\rangle} - 4\overline{F_{S,+}(t)}.$$

The derivation of $\overline{F_{S,+}(t)}, \overline{F_{S,-}(t)}, \overline{\langle\psi(t)|H^2|\psi(t)\rangle}$ essentially uses the technique developed in [40], with analytical result of first and second moment of BGUE ensemble as a function of time. Since the technical details are a bit involved, we only report the results.

We first define some intermediate variables for notational simplicity. First, denote:

$$P_B \equiv P_{\text{ent},0} = \sum_{i=1}^{d_S} (|\tilde{\psi}_i\rangle\langle\tilde{\psi}_i|)_B, \tag{C.7}$$

where $|\tilde{\psi}_i\rangle_B$ are defined in (C.1). Therefore, $P_B$ is the projector of subspace in $\mathcal{H}_B$ that is entangled with $S$ at $t = 0$. Next, we define a set of intermediate variables $\{g_a\}_{a=1}^{12}$:

$$
\begin{aligned}
&g_1 = \text{Tr}[H^2], &&g_2 = \text{Tr}[H^2 P_B], &&g_3 = \text{Tr}_S[(\text{Tr}_B[P_B H])^2], \\
&g_4 = \text{Tr}[(P_B H)^2], &&g_5 = \text{Tr}_S[\text{Tr}_B[P_B H]\text{Tr}_B[H]], &&g_6 = \text{Tr}_S[(\text{Tr}_B[H])^2], \\
&g_7 = \text{Tr}_B[(\text{Tr}_S[H])^2], &&g_8 = \text{Tr}_B[P_B(\text{Tr}_S[H])^2], &&g_9 = (\text{Tr}[P_B H])^2, \\
&g_{10} = \text{Tr}_B[(P_B\text{Tr}_S[H])^2], &&g_{11} = \text{Tr}[P_B H]\text{Tr}[H], &&g_{12} = (\text{Tr}H)^2,
\end{aligned} \tag{C.8}
$$

where $H$ is the original chaotic Ising model Hamiltonian.

We further define another set of intermediate variables $\{f_b(t)\}_{b=1}^{8}$ [40]:

$$
\begin{pmatrix} f_1(t) \\ f_2(t) \\ f_3(t) \\ f_4(t) \\ f_5(t) \\ f_6(t) \\ f_7(t) \\ f_8(t) \end{pmatrix} =
\begin{pmatrix}
0 & 0 & \frac{1}{4} & \frac{1}{2} & \frac{1}{4} \\
0 & \frac{d_B^2-2}{d_B(d_B^2-4)} & \frac{-1}{4(d_B-2)} & \frac{-1}{2d_B} & \frac{-1}{4(d_B+2)} \\
0 & 0 & -\frac{1}{4} & 0 & \frac{1}{4} \\
0 & \frac{-1}{d_B^2-4} & \frac{1}{4(d_B-2)} & 0 & \frac{-1}{4(d_B+2)} \\
\frac{1}{d_B^2-1} & \frac{-2}{d_B^2-4} & \frac{1}{2(d_B-1)(d_B-2)} & 0 & \frac{1}{2(d_B+1)(d_B+2)} \\
0 & 0 & \frac{1}{4} & -\frac{1}{2} & \frac{1}{4} \\
\frac{-1}{d_B^3-d_B} & \frac{4}{d_B(d_B^2-4)} & \frac{-1}{2(d_B-1)(d_B-2)} & 0 & \frac{1}{2(d_B+1)(d_B+2)} \\
0 & \frac{2}{d_B(d_B^2-4)} & \frac{-1}{4(d_B-2)} & \frac{1}{2d_B} & \frac{-1}{4(d_B+2)}
\end{pmatrix}
\times
\begin{pmatrix} 1 \\ e^{-t} \\ e^{-(2-2d_B^{-1})t} \\ e^{-2t} \\ e^{-(2+2d_B^{-1})t} \end{pmatrix}. \tag{C.9}
$$

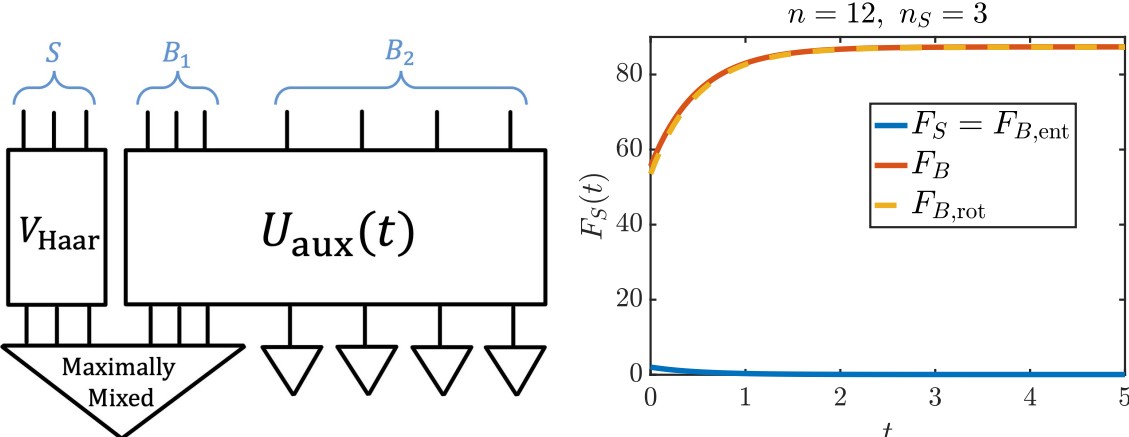

Figure 12: **Left:** An illustration of the time-dependent toy model defined in (C.1) and (C.13). **Right:** Dynamics of QFI in BGUE toy model.

We finally obtain:

$$
\begin{aligned}
\overline{F_{S,+}(t)} &= (d_S^{-2} g_4) f_1(t) + (2 d_S^{-1} g_2 + 2 d_S^{-2} g_5) f_2(t) + (2 d_S - 2 g_3) f_3(t) \\
&\quad + (4 d_S^{-2} g_2 + 4 d_S^{-1} g_5) f_4(t) + (g_1 + d_S^{-1} g_6) f_5(t) \\
&\quad + (d_S^{-2} g_4) f_6(t) + (d_S^{-1} g_1 + g_6) f_7(t) + (2 d_S^{-1} g_2 + 2 d_S^{-2} g_5) f_8(t).
\end{aligned}
\tag{C.10}
$$

For $\overline{F_{S,-}(t)}$, we further define $\alpha = d_S^{-1} \frac{1}{d_S^2 - 1}, \beta = d_S^{-1} \frac{-1}{d_S^3 - d_S}$, then we find:

$$
\begin{aligned}
\overline{F_{S,-}(t)} &= (\alpha g_3 + \beta g_4 + \beta g_9 + \alpha g_{10}) f_1(t) + (2 d_S^{-2} g_5 + 2 d_S^{-2} g_8) f_2(t) \\
&\quad + (2\beta g_3 + 2\alpha g_4 + 2\alpha g_9 + 2\beta g_{10}) f_3(t) + (4 d_S^{-2} g_2 + 4 d_S^{-2} g_{11}) f_4(t) \\
&\quad + (d_S^{-1} g_6 + d_S^{-1} g_7) f_5(t) + (\alpha g_3 + \beta g_4 + \beta g_9 + \alpha g_{10}) f_6(t) \\
&\quad + (d_S^{-1} g_1 + d_S^{-1} g_{12}) f_7(t) + (2 d_S^{-2} g_5 + 2 d_S^{-2} g_8) f_8(t).
\end{aligned}
\tag{C.11}
$$

For $\overline{\langle \psi(t) | H^2 | \psi(t) \rangle}$, we obtain:

$$
\overline{\langle \psi(t) | H^2 | \psi(t) \rangle} = d_S^{-2} \text{Tr}[H^2 P_B] e^{-t} + d_S^{-1} d_B^{-1} \text{Tr}[H^2](1 - e^{-t}).
\tag{C.12}
$$

Lastly, we need to specify $P_B$ in (C.7). To start with a simple choice of initial state which becomes increasingly complicated at later times, we assume at $t = 0$, $S$ is only entangled with the subsystem $B_1 \in B$ which is adjacent to $S$ (see Fig. 12 for illustration), with $n_{B_1} = n_S$. Therefore, we take:

$$
P_B = \mathbf{1}_{B_1} \otimes (|\phi_0\rangle\langle\phi_0|)_{B_2},
\tag{C.13}
$$

where $|\phi_0\rangle_{B_2}$ is a random product state on $B_2$. An illustration of whole setting of toy model is given in Fig.12.

Now, plugging in everything, we plot the dynamics of QFI of this toy model in Fig. 12. This should be compared with Fig. 5 (the $t = 0$ point in the former should be identified with $t \sim 2$ in the latter). We observe that our BGUE toy model successfully reproduces the following features of the spin chain result in Fig. 5: (i) $F_S(t) \approx F_{B,\text{ent}}(t)$, and both quantities are saturated to a very small number (in this case, the two are precisely equal to the flat spectrum); and (ii) $F_B(t) \approx F_{B,\text{rot}}(t)$ at late times, and they are growing monotonically before saturation.

# D Behavior of the subsystem QFI in an interacting integrable system

In the main text, we discussed a sharply different behavior of the subsystem QFI, $F_A(t)$, between chaotic and free-fermion integrable systems in Fig. 6 in the case where $A$ is smaller than half of the system. In this appendix, we further numerically probe the behavior of the subsystem QFI in the Heisenberg XXZ spin chain (with periodic boundary conditions),

$$H_{\text{XXZ}} = \sum_{i=1}^{L} X_i X_{i+1} + Y_i Y_{i+1} + \Delta Z_i Z_{i+1} \, , \tag{D.1}$$

which is an example of an interacting integrable system. We find the behavior shown in Fig. 13 (for $\Delta = 0.5$), which is qualitatively similar to that in the chaotic case: in particular, the decay of the $F_A(t)$ for a small subsystem is monotonic in time (up to small fluctuations), and the saturation value decays exponentially in $n$ for fixed $n_A$. The late-time fluctuations appear to be larger than in the chaotic case. We leave a systematic comparison of the fluctuations of $F_A(t)$ between chaotic and interacting integrable systems to future work. Previously, it has been noted that other measures of quantum chaos such as the OTOC also show a similar behavior between chaotic and interacting integrable spin chains [60, 81].

# E Computational basis classical Fisher information

## E.1 Full system

In this appendix, we derive the late-time saturation value of the classical Fisher information for measurements in computational basis on the full system $\{|\alpha\rangle\}_{\alpha=1}^{d}$. $d$ is the total Hilbert

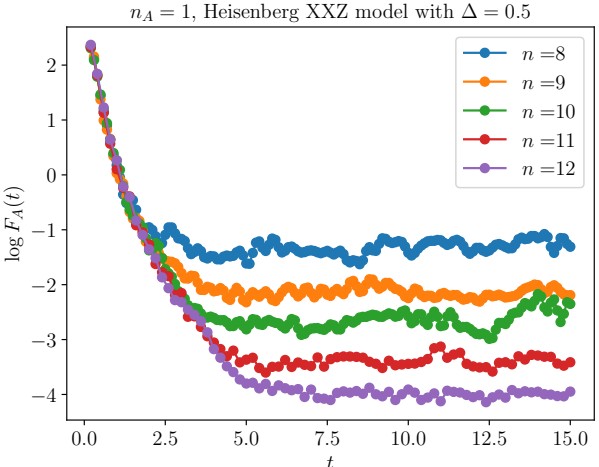

Figure 13: Behavior of the subsystem QFI $F_A(t)$ for a subsystem of size $n_A = 1$ in the Heisenberg XXZ model with $\Delta = 0.5$, for various full system sizes $n$ ranging from 8 to 12.

space dimension. Using (3), note that we have

$$f^{\text{comp}}(t) = \sum_{\alpha=1}^{d} \frac{1}{|\langle\alpha|\psi(t)\rangle|^2} |\langle\alpha|[H,|\psi(t)\rangle\langle\psi(t)|]|\alpha\rangle|^2 \tag{E.1}$$

$$= 2\sum_{\alpha=1}^{d} \langle\alpha|H|\psi(t)\rangle\langle\psi(t)|H|\alpha\rangle - 2\,\text{Re}\left(\sum_{\alpha=1}^{d} \langle\alpha|H|\psi(t)\rangle\langle\alpha|H|\psi(t)\rangle\langle\psi(t)|\alpha\rangle / \overline{\langle\psi(t)|\alpha\rangle}\right)$$

$$= 2\langle\psi_0|H^2|\psi_0\rangle - 2\,\text{Re}\left(\sum_{\alpha=1}^{d} \langle\alpha|H|\psi(t)\rangle\langle\alpha|H|\psi(t)\rangle\langle\psi(t)|\alpha\rangle / \overline{\langle\psi(t)|\alpha\rangle}\right),$$

where the overline indicates complex conjugation. Like in previous calculations, we approximate the late-time state as a typical Haar random state $|\psi_{\text{Haar}}\rangle$. For the present calculation, we do not need to use the entanglement structure of this state, and can simply use the fact that the coefficients $z_\nu \equiv \langle\nu|\psi_{\text{Haar}}\rangle$ can be treated as independent complex Gaussian random variables with zero mean and variance $1/d$, so that we have (In this appendix, we will use the notation $\langle\ldots\rangle$ for the Gaussian averages.)

$$\langle z_\mu \bar{z}_\nu\rangle = \delta_{\mu\nu}/d\,, \qquad \langle z_\mu z_\nu\rangle = 0\,, \qquad \langle \bar{z}_\nu/z_\nu\rangle = 0\,, \tag{E.2}$$

where the last equality follows from that $\bar{z}_\nu/z_\nu$ is a pure phase, and a complex Gaussian random variable has a uniformly distributed phase.

Using (E.1), we have the following average of $f^{\text{comp}}$ over such states:

$$f^{\text{comp}} = \frac{2}{d}\text{Tr}[H^2] - 2\,\text{Re}\left(\sum_{\alpha,\beta,\gamma=1}^{d} H_{\alpha\beta}H_{\alpha\gamma}z_\beta z_\gamma \bar{z}_\alpha/z_\alpha\right). \tag{E.3}$$

We organize the sum in the second term above as follows:

$$\sum_{\alpha,\beta,\gamma=1}^{d} H_{\alpha\beta}H_{\alpha\gamma}z_\beta z_\gamma \bar{z}_\alpha/z_\alpha = \sum_{\alpha,\beta=1}^{d} H_{\alpha\beta}H_{\alpha\alpha}z_\beta \bar{z}_\alpha + \sum_{\alpha,\gamma=1}^{d} H_{\alpha\alpha}H_{\alpha\gamma}z_\gamma \bar{z}_\alpha$$
$$+ \sum_{\beta,\gamma\neq\alpha}^{d} H_{\alpha\beta}H_{\alpha\gamma}z_\beta z_\gamma \bar{z}_\alpha/z_\alpha - \sum_{\alpha=1}^{d} H_{\alpha\alpha}^2 |z_\alpha|^2\,, \tag{E.4}$$

where we sum over $\gamma = \alpha$, $\beta = \alpha$, and $\beta \neq \alpha \wedge \gamma \neq \alpha$ respectively, and in the last term we subtract the double-counted $\gamma = \beta = \alpha$ case.

Taking the average, we have

$$\langle z_\beta \bar{z}_\alpha\rangle = \delta_{\beta\alpha}/d\,, \qquad \langle z_\beta z_\gamma \bar{z}_\alpha/z_\alpha\rangle = \langle z_\beta z_\gamma\rangle\langle\bar{z}_\alpha/z_\alpha\rangle = 0\,, \tag{E.5}$$

where the factorization in the second equality follows from independence of the different $z_\alpha$.

We hence obtain

$$f^{\text{comp}} = \frac{2}{d}\text{Tr}[H^2] - \frac{2}{d}\sum_{\alpha=1}^{d} H_{\alpha\alpha}^2\,. \tag{E.6}$$

For a local Hamiltonian, both terms are $\mathcal{O}(n)$, explaining the extensive late-time value of $f^{\text{comp}}$ observed for the spin chain model in Fig. 9.

### E.2 Subsystem

Now consider the computational basis CFI for a subsystem, $f_A^{\text{comp}}$. We have

$$
\begin{aligned}
f_A^{\text{comp}} &= \sum_{\alpha=1}^{d_A} \frac{|\langle\alpha|\operatorname{Tr}_{\bar{A}}[H,|\psi\rangle\langle\psi|]|\alpha\rangle|^2}{\langle\alpha|\operatorname{Tr}_{\bar{A}}|\psi\rangle\langle\psi||\alpha\rangle} \\
&= \sum_{\alpha=1}^{d_A} \langle\alpha|\operatorname{Tr}_{\bar{A}}|\psi\rangle\langle\psi||\alpha\rangle^{-1} \\
&\times \sum_{i,j=1}^{d_{\bar{A}}} 2\Big[\langle\alpha i|H|\psi\rangle\langle\psi|\alpha i\rangle\langle\alpha j|\psi\rangle\langle\psi|H|\alpha j\rangle - \operatorname{Re}\big(\langle\alpha i|H|\psi\rangle\langle\psi|\alpha i\rangle\langle\alpha j|H|\psi\rangle\langle\psi|\alpha j\rangle\big)\Big],
\end{aligned}
\tag{E.7}
$$

where we use $\alpha$ and $i$ respectively (and more generally Greek and English small letters respectively) to denote the computational bases in $A$ and $\bar{A}$, and $|\alpha j\rangle \equiv |\alpha\rangle_A |j\rangle_{\bar{A}}$.

In terms of the matrix elements of $H$ in the computational basis, we have

$$
\begin{aligned}
f_A^{\text{comp}} &= 2\sum_{\alpha=1}^{d_A} \langle\alpha|\operatorname{Tr}_{\bar{A}}|\psi\rangle\langle\psi||\alpha\rangle^{-1} \\
&\times \Bigg( \sum_{\gamma,\omega=1}^{d_A} \sum_{i,j,l,p=1}^{d_{\bar{A}}} H_{\alpha i,\gamma l} H_{\omega p,\alpha j} \langle\gamma l|\psi\rangle\langle\psi|\alpha i\rangle\langle\alpha j|\psi\rangle\langle\psi|\omega p\rangle \\
&\quad - \sum_{\nu,\theta=1}^{d_A} \sum_{i,j,n,t=1}^{d_{\bar{A}}} \operatorname{Re}\big(H_{\alpha i,\nu n} H_{\alpha j,\theta t} \langle\nu n|\psi\rangle\langle\psi|\alpha i\rangle\langle\theta t|\psi\rangle\langle\psi|\alpha j\rangle\big)\Bigg).
\end{aligned}
\tag{E.8}
$$

In this appendix, we will assume in all remaining equations that $A$ is always the subsystem *larger than half* of the full system. Then recall the following approximation for the late-time state with a flat spectrum,

$$
|\psi\rangle = \frac{1}{\sqrt{d_{\bar{A}}}} \sum_{a=1}^{d_{\bar{A}}} |\xi^a\rangle_A |\phi^a\rangle_{\bar{A}},
\tag{E.9}
$$

where $\{|\xi^a\rangle\}_{a=1}^{d_{\bar{A}}}$ and $\{|\phi^a\rangle\}_{a=1}^{d_{\bar{A}}}$ are independent Haar random pure states of dimension $d_A$ and $d_{\bar{A}}$ respectively. It follows that

$$
\langle\alpha i|\psi\rangle = \frac{1}{\sqrt{d_{\bar{A}}}} \sum_{a=1}^{d_{\bar{A}}} z_\alpha^a x_i^a,
\tag{E.10}
$$

where $\{z_\alpha^a\}_{\alpha,a}$ and $\{x_i^a\}_{a,i}$ are independent complex Gaussians with zero mean and variance $1/d_A$ and $1/d_{\bar{A}}$ respectively. Using (E.9), we have that the denominator $\langle\alpha|\operatorname{Tr}_{\bar{A}}|\psi\rangle\langle\psi||\alpha\rangle$ is equal to

$$
\langle\alpha|\operatorname{Tr}_{\bar{A}}|\psi\rangle\langle\psi||\alpha\rangle = \frac{1}{d_{\bar{A}}} \sum_{a=1}^{d_{\bar{A}}} \langle\alpha \mid \xi^a\rangle\langle\xi^a \mid \alpha\rangle = \frac{1}{d_{\bar{A}}} \sum_{a=1}^{d_{\bar{A}}} z_\alpha^a \bar{z}_\alpha^a =: s_\alpha,
\tag{E.11}
$$

and $s_\alpha$ has mean $1/d_A$.

In this appendix, we will again use overlines to indicate complex conjugation, and $\langle\ldots\rangle$ to indicate Gaussian averages. The computational-basis CFI reads,

$$
\begin{aligned}
f_A^{\text{comp}} &= \frac{2}{d_{\bar{A}}^2} \sum_{\alpha,\gamma,\omega=1}^{d_A} \sum_{i,j,l,p=1}^{d_{\bar{A}}} \sum_{a,b,c,d=1}^{d_{\bar{A}}} H_{\alpha i,\gamma l} H_{\omega p,\alpha j}\, z_\gamma^a x_l^a \bar{z}_\alpha^b \bar{x}_i^b z_\alpha^c x_j^c \bar{z}_\omega^d \bar{x}_p^d / s_\alpha && (=: I_1) \\
&\quad - \frac{2}{d_{\bar{A}}^2} \sum_{\alpha,\nu,\theta=1}^{d_A} \sum_{i,j,n,t=1}^{d_{\bar{A}}} \sum_{a,b,c,d=1}^{d_{\bar{A}}} \operatorname{Re}\Big(H_{\alpha i,\nu n} H_{\alpha j,\theta t}\, z_\nu^a x_n^a \bar{z}_\alpha^b \bar{x}_i^b z_\theta^c x_t^c \bar{z}_\alpha^d \bar{x}_j^d / s_\alpha\Big) && (=: I_2).
\end{aligned}
\tag{E.12}
$$

Now we work on the first term $I_1$. Taking the average of the Gaussians,

$$\sum_{a,b,c,d=1}^{d_{\bar{A}}} \langle z_\gamma^a x_l^a \bar{z}_\alpha^b \bar{x}_i^b z_j^c x_j^c \bar{z}_\omega^d \bar{x}_p^d / s_\alpha \rangle = \frac{1}{d_{\bar{A}}^2} \sum_{a,c=1}^{d_{\bar{A}}} \langle z_\gamma^a \bar{z}_\alpha^a z_\alpha^c \bar{z}_\omega^c / s_\alpha \rangle \delta_{il} \delta_{jp} + \frac{1}{d_A} \delta_{\gamma\omega} \delta_{lp} \delta_{ij}. \quad (E.13)$$

We have

$$\begin{aligned}
I_1 &= \frac{2}{d_{\bar{A}}^2} \sum_{\alpha,\gamma,\omega=1}^{d_A} \sum_{i,j,l,p=1}^{d_{\bar{A}}} H_{\alpha i,\gamma l} H_{\omega p,\alpha j} \left( \frac{1}{d_{\bar{A}}^2} \sum_{a,c=1}^{d_{\bar{A}}} \langle z_\gamma^a \bar{z}_\alpha^a z_\alpha^c \bar{z}_\omega^c / s_\alpha \rangle \delta_{il} \delta_{jp} + \frac{1}{d_A} \delta_{\gamma\omega} \delta_{lp} \delta_{ij} \right) \\
&= \frac{2}{d_{\bar{A}}^4} \sum_{\alpha,\gamma,\omega=1}^{d_A} \sum_{i,j=1}^{d_{\bar{A}}} H_{\alpha i,\gamma i} H_{\omega j,\alpha j} \sum_{a,c=1}^{d_{\bar{A}}} \langle z_\gamma^a \bar{z}_\alpha^a z_\alpha^c \bar{z}_\omega^c / s_\alpha \rangle + \frac{2}{d_{\bar{A}}^2 d_A} \mathrm{Tr} H^2.
\end{aligned} \quad (E.14)$$

Let's break the first term above into two cases, $\gamma, \omega \neq \alpha$ and $\omega = \gamma = \alpha$, (other cases yield zero.)

$$\begin{aligned}
&\frac{2}{d_{\bar{A}}^4} \sum_{\gamma,\omega\neq\alpha}^{d_A} \sum_{i,j=1}^{d_{\bar{A}}} H_{\alpha i,\gamma i} H_{\omega j,\alpha j} \sum_{a,c=1}^{d_{\bar{A}}} \langle \bar{z}_\alpha^a z_\alpha^c / s_\alpha \rangle \langle z_\gamma^a \bar{z}_\omega^c \rangle + \frac{2}{d_{\bar{A}}^4} \sum_{\alpha=1}^{d_A} \sum_{i,j=1}^{d_{\bar{A}}} H_{\alpha i,\alpha i} H_{\alpha j,\alpha j} \sum_{a,c=1}^{d_{\bar{A}}} \langle z_\alpha^a \bar{z}_\alpha^a z_\alpha^c \bar{z}_\alpha^c / s_\alpha \rangle \\
&= \frac{2}{d_A d_{\bar{A}}^3} \sum_{\gamma\neq\alpha}^{d_A} \sum_{i,j=1}^{d_{\bar{A}}} H_{\alpha i,\gamma i} H_{\gamma j,\alpha j} + \frac{2}{d_A d_{\bar{A}}^2} \sum_{\alpha=1}^{d_A} \sum_{i,j=1}^{d_{\bar{A}}} H_{\alpha i,\alpha i} H_{\alpha j,\alpha j} \\
&= \frac{2}{d_A d_{\bar{A}}^3} \mathrm{Tr}_A H_A^2 + \left( \frac{2}{d_A d_{\bar{A}}^2} - \frac{2}{d_A d_{\bar{A}}^3} \right) \sum_{\alpha=1}^{d_A} (H_A)_{\alpha\alpha}^2.
\end{aligned} \quad (E.15)$$

We hence have

$$I_1 = \frac{2}{d_A d_{\bar{A}}^2} \mathrm{Tr} H^2 + \frac{2}{d_A d_{\bar{A}}^3} \mathrm{Tr}_A H_A^2 + \left( \frac{2}{d_A d_{\bar{A}}^2} - \frac{2}{d_A d_{\bar{A}}^3} \right) \sum_{\alpha=1}^{d_A} (H_A)_{\alpha\alpha}^2. \quad (E.16)$$

Now we proceed to $I_2$. Taking the average of the Gaussians,

$$\sum_{a,b,c,d=1}^{d_{\bar{A}}} \langle z_\nu^a x_n^a \bar{z}_\alpha^b \bar{x}_i^b z_\theta^c x_t^c \bar{z}_\alpha^d \bar{x}_j^d / s_\alpha \rangle = \frac{1}{d_{\bar{A}}^2} \sum_{a,c=1}^{d_{\bar{A}}} \langle z_\nu^a \bar{z}_\alpha^a z_\theta^c \bar{z}_\alpha^c / s_\alpha \rangle \delta_{ni} \delta_{jt} + \frac{1}{d_{\bar{A}}^2} \sum_{a,b=1}^{d_{\bar{A}}} \langle z_\nu^a \bar{z}_\alpha^b z_\theta^b \bar{z}_\alpha^a / s_\alpha \rangle \delta_{nj} \delta_{it}. \quad (E.17)$$

For $I_2$ defined in (E.12), we have (we anticipate that the result will be real and drop the "Re"):

$$\begin{aligned}
I_2 &= -\frac{2}{d_{\bar{A}}^4} \sum_{\alpha,\nu,\theta=1}^{d_A} \sum_{i,j,n,t=1}^{d_{\bar{A}}} H_{\alpha i,\nu n} H_{\alpha j,\theta t} \left( \sum_{a,c=1}^{d_{\bar{A}}} \langle z_\nu^a \bar{z}_\alpha^a z_\theta^c \bar{z}_\alpha^c / s_\alpha \rangle \delta_{ni} \delta_{jt} + \sum_{a,b=1}^{d_{\bar{A}}} \langle z_\nu^a \bar{z}_\alpha^b z_\theta^b \bar{z}_\alpha^a / s_\alpha \rangle \delta_{nj} \delta_{it} \right) \\
&= -\frac{2}{d_{\bar{A}}^4} \sum_{\alpha,\nu,\theta=1}^{d_A} \sum_{i,j=1}^{d_{\bar{A}}} H_{\alpha i,\nu i} H_{\alpha j,\theta j} \sum_{a,c=1}^{d_{\bar{A}}} \langle z_\nu^a \bar{z}_\alpha^a z_\theta^c \bar{z}_\alpha^c / s_\alpha \rangle \\
&\quad -\frac{2}{d_{\bar{A}}^4} \sum_{\alpha,\nu,\theta=1}^{d_A} \sum_{i,j=1}^{d_{\bar{A}}} H_{\alpha i,\nu j} H_{\alpha j,\theta i} \sum_{a,b=1}^{d_{\bar{A}}} \langle z_\nu^a \bar{z}_\alpha^b z_\theta^b \bar{z}_\alpha^a / s_\alpha \rangle.
\end{aligned} \quad (E.18)$$

We now break the sums into two cases, $\nu, \theta \neq \alpha$ and $\nu = \theta = \alpha$, (other cases yield zero)

$$
\begin{aligned}
I_2 = &-\frac{2}{d_{\bar{A}}^4} \sum_{\nu,\theta \neq \alpha}^{d_A} \sum_{i,j=1}^{d_{\bar{A}}} H_{\alpha i, \nu i} H_{\alpha j, \theta j} \sum_{a,c=1}^{d_{\bar{A}}} \langle z_\nu^a \bar{z}_\alpha^a z_\theta^c \bar{z}_\alpha^c / s_\alpha \rangle \\
&- \frac{2}{d_{\bar{A}}^4} \sum_\alpha^{d_A} \sum_{i,j=1}^{d_{\bar{A}}} H_{\alpha i, \alpha i} H_{\alpha j, \alpha j} \sum_{a,c=1}^{d_{\bar{A}}} \langle z_\alpha^a \bar{z}_\alpha^a z_\alpha^c \bar{z}_\alpha^c / s_\alpha \rangle \\
&- \frac{2}{d_{\bar{A}}^4} \sum_{\nu,\theta \neq \alpha=1}^{d_A} \sum_{i,j=1}^{d_{\bar{A}}} H_{\alpha i, \nu j} H_{\alpha j, \theta i} \sum_{a,b=1}^{d_{\bar{A}}} \langle z_\nu^a \bar{z}_\alpha^b z_\theta^b \bar{z}_\alpha^a / s_\alpha \rangle \\
&- \frac{2}{d_{\bar{A}}^4} \sum_{\alpha=1}^{d_A} \sum_{i,j=1}^{d_{\bar{A}}} H_{\alpha i, \alpha j} H_{\alpha j, \alpha i} \sum_{a,b=1}^{d_{\bar{A}}} \langle z_\alpha^a \bar{z}_\alpha^b z_\alpha^b \bar{z}_\alpha^a / s_\alpha \rangle \\
= &-\frac{2}{d_A d_{\bar{A}}^2} \sum_{\alpha=1}^{d_A} \sum_{i,j=1}^{d_{\bar{A}}} \left( H_{\alpha i, \alpha i} H_{\alpha j, \alpha j} + H_{\alpha i, \alpha j} H_{\alpha j, \alpha i} \right) = -\frac{2}{d_A d_{\bar{A}}^2} \sum_{\alpha=1}^{d_A} \left[ (H_A)_{\alpha\alpha}^2 + \mathrm{Tr}_{\bar{A}}(\langle \alpha | H | \alpha \rangle)_{\bar{A}}^2 \right].
\end{aligned}
\tag{E.19}
$$

In conclusion,

$$
\begin{aligned}
f_A^{\mathrm{comp}} = &\frac{2}{d_A d_{\bar{A}}^2} \mathrm{Tr} H^2 + \frac{2}{d_A d_{\bar{A}}^3} \mathrm{Tr}_A H_A^2 - \frac{2}{d_A d_{\bar{A}}^3} \sum_{\alpha=1}^{d_A} (H_A)_{\alpha\alpha}^2 - \frac{2}{d_A d_{\bar{A}}^2} \sum_{\alpha=1}^{d_A} \mathrm{Tr}_{\bar{A}}(\langle \alpha | H | \alpha \rangle)_{\bar{A}}^2 \\
&\approx \mathcal{O}(n/d_{\bar{A}}) + \mathcal{O}(n_A/d_{\bar{A}}) + \mathcal{O}(n_A/d_{\bar{A}}) + \mathcal{O}(n_A/d_{\bar{A}}).
\end{aligned}
\tag{E.20}
$$

Similarly, we can also calculate the CFI for $\bar{A}$:

$$
\begin{aligned}
f_{\bar{A}}^{\mathrm{comp}} = &\frac{2}{d_A^2 d_{\bar{A}}} \mathrm{Tr} H^2 + \frac{2}{d_A^2 d_{\bar{A}}^2} \mathrm{Tr}_{\bar{A}} H_{\bar{A}}^2 - \frac{2}{d_A^2 d_{\bar{A}}^2} \sum_{\alpha=1}^{d_{\bar{A}}} (H_{\bar{A}})_{\alpha\alpha}^2 - \frac{2}{d_A^2 d_{\bar{A}}} \sum_{\alpha=1}^{d_{\bar{A}}} \mathrm{Tr}_A(\langle \alpha | H | \alpha \rangle)_A^2 \\
&\approx \mathcal{O}(n/d_A) + \mathcal{O}(n_{\bar{A}}/d_A) + \mathcal{O}(n_{\bar{A}}/d_A) + \mathcal{O}(n_{\bar{A}}/d_A).
\end{aligned}
\tag{E.21}
$$

## F   Trace distance between classical probability distributions

Consider the task of distinguishing $|\psi\rangle$ (a Haar state on $n$-qubit Hilbert space $\mathcal{H}$) and the maximally mixed state $\rho_{\mathrm{mm}}$ on $\mathcal{H}$. Let $A$ be a subsystem with $n_A$ qubits and $\bar{A}$ be its complement with $n_{\bar{A}}$ qubits. We denote the Hilbert space dimensions as $d_A \equiv 2^{n_A}, d_{\bar{A}} \equiv 2^{n_{\bar{A}}}, d \equiv d_A d_{\bar{A}}$. Suppose that the experimentalists who need to perform the task can only perform measurements in the computational basis of $A$. Then, the total variation distance or trace distance (TD) between the classical probability distributions on $d_A$ bit-strings,

$$
\mathrm{TD}_\psi(d, d_{\bar{A}}) \equiv \sum_{i_A=0}^{d_A-1} |p_{i_A} - p'_{i_A}| = \sum_{i_A=0}^{d_A-1} \left| \langle i_A | \mathrm{Tr}_{\bar{A}}(|\psi\rangle\langle\psi|) | i_A \rangle - d_A^{-1} \right|,
\tag{F.1}
$$

would be a good information-theoretic measure of the distinguishability of $|\psi\rangle$ and $\rho_{\mathrm{mm}}$ under this restriction. We will average this quantity over Haar-random states $|\psi\rangle$, and find $\mathrm{TD}(d, d_{\bar{A}}) \equiv \overline{\mathrm{TD}_\psi(d, d_{\bar{A}})}$ as a function of $d, d_{\bar{A}}$.

We first state the results, leaving the details to the following two subsections. We first consider the case where we have access to measurements on the full system, namely $d_A = d$ and $d_{\bar{A}} = 1$. We find (see Appendix F.1 for the derivation):

$$
\mathrm{TD}(d) \equiv \mathrm{TD}(d, 1) = 2(1 - d^{-1})^d, \qquad f(+\infty) = 2e^{-1} \approx 0.735759.
\tag{F.2}
$$

Next, for $d_A < d$, we find (see Appendix F.2 for derivation):

$$\text{TD}(d, d_{\tilde{A}}) = 2\frac{d_{\tilde{A}}^{d_{\tilde{A}}-1}(d-d_{\tilde{A}})^{d-d_{\tilde{A}}}\Gamma(d)}{d^d \Gamma(d_{\tilde{A}})\Gamma(d-d_{\tilde{A}})},$$

$$\text{TD}(+\infty, d_{\tilde{A}}) = 2\frac{d_{\tilde{A}}^{d_{\tilde{A}}-1}}{\Gamma(d_{\tilde{A}})}e^{-d_{\tilde{A}}} = 2(2\pi)^{-1/2}d_{\tilde{A}}^{-1/2} + O(d_{\tilde{A}}^{-3/2}). \tag{F.3}$$

This is tells us that: (1) When the experimentalists have access to the full system, then the trace distance is $\mathcal{O}(1)$ in thermodynamic limit, indicating that a typical Haar random pure state and a maximally mixed state can be distinguished well. This is similar to the conclusion we arrived at by evaluating the CFI of the computational basis for the full system, $f^{\text{comp}}$, in Sec. E. (2) When the experimentalists even slightly lose control of the full system, they cannot distinguish a typical Haar random pure state and the maximally mixed state well, in the sense that the trace distance in thermodynamic limit is of order $\mathcal{O}(2^{-n_{\tilde{A}}/2})$, which is exponentially suppressed in the volume of the complementary subsystem to which the experimentalists have no access. This exponential decaying behaviour is again similar to that of the computational basis CFI of a subsystem, $f_A^{\text{comp}}$.

## F.1 Full system

Let $|\psi\rangle$ be a Haar-random state in the $d$-dimensional Hilbert space $\mathcal{H}$, and $\{|i\rangle, i = 0, \ldots, d-1\}$ be a fixed orthonormal basis. In this section we want to calculate:

$$\text{TD}(d) = \sum_{i=0}^{d-1}\overline{|\langle i|\psi\rangle\langle\psi|i\rangle - d^{-1}|} = d \cdot \overline{|\langle 0|\psi\rangle\langle\psi|0\rangle - d^{-1}|}. \tag{F.4}$$

Defining a random variable $x = \langle 0|\psi\rangle\langle\psi|0\rangle$, we first calculate its moments $\overline{x^n}$. We use the following formula which calculates the moments of the density matrix:

$$\overline{|\psi\rangle\langle\psi|^{\otimes n}} = \frac{\sum_{\pi\in S_n}\pi}{\sum_{\pi\in S_n}\text{Tr}(\pi)} = \frac{\sum_{\pi\in S_n}\pi}{\sum_{\pi\in S_n}d^{\ell(\pi)}}, \tag{F.5}$$

where $\pi$ is the permutation operator acting on the $n$-copy Hilbert space $\mathcal{H}^{\otimes n}$, and $\ell(\pi)$ is the number of minimal cycles in $\pi$. More precisely, we can define Cayley distance $d_{Cayley}(\pi, \sigma)$ on the permutation group, then $\ell(\pi) = n - d_{Cayley}(\pi, e)$ where $e$ is the identity permutation.

Using the formula

$$\sum_{\pi\in S_n}d^{\ell(\pi)} = \frac{(d-1+n)!}{(d-1)!}, \tag{F.6}$$

we obtain:

$$\overline{x^n} = \langle 0^{\otimes n}|\overline{|\psi\rangle\langle\psi|^{\otimes n}}|0^{\otimes n}\rangle = \frac{\sum_{\pi\in S_n}\langle 0^{\otimes n}|\pi|0^{\otimes n}\rangle}{\frac{(d-1+n)!}{(d-1)!}} = \frac{n!}{\frac{(d-1+n)!}{(d-1)!}} = \binom{d+n-1}{n}^{-1}, \tag{F.7}$$

where we notice that $\langle 0^{\otimes n}|\pi|0^{\otimes n}\rangle = 1, \forall \pi \in S_n$. Next, we want to calculate the probability density distribution $\rho(x)$ of $x$. (Note that $x$ itself is a Born-rule probability, which has now become are random variable over different choices of the random pure states $|\psi\rangle$, and we are calculating the probability distribution of $x$ resulting from that of $|\psi\rangle$.) We use the resolvent method, defining

$$R(\lambda) = \overline{(\lambda - x)^{-1}}, \qquad \rho(\lambda) = -\pi^{-1}\text{Im}[R(\lambda + i0^+)]. \tag{F.8}$$

The resolvent can be calculated using the moments to be

$$R(\lambda) = \lambda^{-1}\sum_{n=0}^{\infty}\lambda^{-n}\overline{x^n} = \lambda^{-1}\sum_{n=0}^{\infty}\lambda^{-n}\binom{d+n-1}{n}^{-1} = \lambda^{-1}{}_2F_1(1,1,d,\lambda^{-1}), \qquad \text{(F.9)}$$

where ${}_2F_1(a,b,c,z)$ is the Hypergeometric function. We find that the resulting probability density is

$$\rho(\lambda) = (d-1)(1-\lambda)^{d-2}, \qquad \lambda \in [0,1]. \qquad \text{(F.10)}$$

This can be checked for $d = 2,3$ using the explicit form of the hypergeometric function: $\lambda^{-1}{}_2F_1(1,1,2,\lambda^{-1}) = -\log(1-\lambda^{-1})$ and $\lambda^{-1}{}_2F_1(1,1,3,\lambda^{-1}) = 2 + 2(\lambda-1)\log(1-\lambda^{-1})$. More generally, by comparing the functional form (F.10) to the numerical evaluation of the RHS of (F.8) in Mathematica, we found exact agreement. In the large $d$ limit, this becomes the Porter-Thomas distribution:

$$p(\lambda) \approx d e^{-\lambda d}. \qquad \text{(F.11)}$$

Using this form of $\rho(\lambda)$, we obtain:

$$\begin{aligned} \text{TD}(d) &= d\int_0^1 d\lambda \cdot \rho(\lambda)|\lambda - d^{-1}| = d\int_0^1 d\lambda \cdot (d-1)(1-\lambda)^{d-2}|\lambda - d^{-1}| \\ &= 2(1-d^{-1})^d. \end{aligned} \qquad \text{(F.12)}$$

Taking $d \to \infty$, we find:

$$\text{TD}(\infty) = 2e^{-1} \approx 0.735759. \qquad \text{(F.13)}$$

For consistency, in the remaining part of this subsection, we report an independent calculation of (F.10) for $d = 2,3$ using the Bloch vector method.

**Calculation using Bloch vector at $d = 2$.** As an independent check, let us consider $d = 2$. The result (F.10) predicts that $\rho(\lambda) = 1, \lambda \in [0,1]$, which is a constant. This seems unusual, but can be confirmed using Bloch vector. A qubit state is given by:

$$|\psi\rangle = \cos(\theta/2)|0\rangle + e^{i\phi}\sin(\theta/2)|1\rangle, \qquad \text{(F.14)}$$

so in this case, we have $x = \langle 0|\psi\rangle\langle\psi|0\rangle = \cos^2(\theta/2)$. The uniform distribution on the Bloch sphere is $\rho(\theta,\phi) = \frac{1}{4\pi}\sin\theta$. Integrating out $\phi$, we obtain $\rho(\theta) = \frac{1}{2}\sin\theta$. Changing variables to $x$, we have:

$$\rho(x) = \rho(\theta)\bigg/\left|\frac{dx}{d\theta}\right| = \frac{\frac{1}{2}\sin\theta}{2\cos(\theta/2)\sin(\theta/2)\frac{1}{2}} = 1. \qquad \text{(F.15)}$$

**Calculation using Bloch vector at $d = 3$.** For SU(3), the Bloch vector is more complicated, but the calculation is still viable. The Haar measure of SU(3) is given by:

$$dV \propto \sin(2\beta)\sin(2\theta)\sin^2\theta\sin(2b)\, d\alpha\, d\beta\, d\gamma\, da\, db\, dc\, d\theta\, d\phi, \qquad \text{(F.16)}$$

where the eight Euler angles are within the range:

$$\alpha,\gamma,a,c \in [0,\pi], \qquad \beta,b,\theta \in [0,\pi/2], \qquad \phi \in [0,2\pi]. \qquad \text{(F.17)}$$

The SU(3) unitary parametrized by the Euler angles is given by:

$$U = e^{i\lambda_3\alpha}e^{i\lambda_2\beta}e^{i\lambda_3\gamma}e^{i\lambda_5\theta}e^{i\lambda_3a}e^{i\lambda_2b}e^{i\lambda_3c}e^{i\lambda_8\phi}, \qquad \text{(F.18)}$$

where $\lambda_{1\sim 8}$ are eight ($3\times 3$) Gell-Mann matrices for generator of $SU(3)$ in fundamental representation (same convention as in Wikipedia). Acting with $U$ on an arbitrary state, say, $(1,0,0)$, we obtain a Bloch vector:

$$|\psi\rangle = U\begin{pmatrix} 1 \\ 0 \\ 0 \end{pmatrix} = \begin{pmatrix} e^{i(-a+c+\alpha-\gamma+\phi/\sqrt{3})}(e^{i(2a+2\gamma)}\cos b\cos\beta\cos\theta - \sin b\sin\beta) \\ e^{i(-a+c-\alpha-\gamma+\phi/\sqrt{3})}(-e^{i(2a+2\gamma)}\cos b\sin\beta\cos\theta - \sin b\cos\beta) \\ -e^{i(a+c+\phi/\sqrt{3})}\cos b\sin\theta \end{pmatrix}. \qquad \text{(F.19)}$$

For convenience, we study the third component, namely define $x = \cos^2 b\sin^2\theta$. The marginal distribution on $(b,\theta)$ is given by $\rho(b,\theta) = 2\sin(2b)\sin(2\theta)\sin^2\theta$. Then the distribution of $x$ is given by:

$$\rho(x) = \int_0^{\pi/2}\int_0^{\pi/2} db\, d\theta\, \rho(b,\theta)\delta(x - \cos^2 b\sin^2\theta) = 2(1-x). \qquad \text{(F.20)}$$

## F.2 Subsystem

We now consider a subsystem of dimension $d_A$, and use $\{|i\rangle, i_A = 0,\dots,d_A-1\}$ to denote an orthonormal basis. We now want to calculate:

$$\mathrm{TD}(d,d_{\bar{A}}) = \sum_{i_A=0}^{d_A-1}\overline{\left|\langle i_A|\mathrm{Tr}_{\bar{A}}(|\psi\rangle\langle\psi|)|i_A\rangle - d_A^{-1}\right|} = d_A\cdot\overline{\left|\langle 0_A|\mathrm{Tr}_{\bar{A}}(|\psi\rangle\langle\psi|)|0_A\rangle - d_A^{-1}\right|}. \qquad \text{(F.21)}$$

Similar to the previous subsection, we define a random variable $x \equiv \langle 0_A|\mathrm{Tr}_{\bar{A}}(|\psi\rangle\langle\psi|)|0_A\rangle$. We first calculate its moments:

$$\overline{x^n} = \overline{\langle 0_A^{\otimes n}|\mathrm{Tr}_{\bar{A}_1,\dots,\bar{A}_n}\left(|\psi\rangle\langle\psi|^{\otimes n}\right)|0_A^{\otimes n}\rangle} = \frac{\sum_{\pi\in S_n}\langle 0_A^{\otimes n}|\pi_A|0_A^{\otimes n}\rangle\mathrm{Tr}(\pi_{\bar{A}})}{\sum_{\pi\in S_n}\mathrm{Tr}(\pi_A\pi_{\bar{A}})} = \frac{\sum_{\pi\in S_n}d_{\bar{A}}^{\ell(\pi)}}{\sum_{\pi\in S_n}d^{\ell(\pi)}}$$

$$= \binom{d_{\bar{A}}+n-1}{n}\cdot\binom{d+n-1}{n}^{-1}. \qquad \text{(F.22)}$$

Next, we find the resolvent:

$$R(\lambda) = \overline{(\lambda-x)^{-1}} = \lambda^{-1}\sum_{\lambda=0}^{\infty}\lambda^{-n}\overline{x^n} = \lambda^{-1}{}_2F_1(1,d_{\bar{A}},d,\lambda^{-1}). \qquad \text{(F.23)}$$

Using $\rho(\lambda) = -\pi^{-1}\mathrm{Im}[R(\lambda+i0^+)]$, we obtain the pdf of $x$, again by using Mathematica:

$$\rho(\lambda) = \frac{\Gamma(d)}{\Gamma(d-d_{\bar{A}})\Gamma(d_{\bar{A}})}(1-\lambda)^{d-d_{\bar{A}}-1}\lambda^{d_{\bar{A}}-1}, \qquad \lambda\in[0,1]. \qquad \text{(F.24)}$$

This is known as the Erlang distribution, and was previously derived for this context in [82]. As a consistency check, one can show that the averaged probability distribution is uniform:

$$\int_0^1 d\lambda\,\rho(\lambda)\lambda = d_A^{-1}. \qquad \text{(F.25)}$$

Now, we can calculate the trace distance:

$$\mathrm{TD}(d,d_{\bar{A}}) = d_A\int_0^1 d\lambda\,\rho(\lambda)|\lambda-d_A^{-1}| = 2\frac{d_{\bar{A}}^{d_{\bar{A}}-1}(d-d_{\bar{A}})^{d-d_{\bar{A}}}\Gamma(d)}{d^d\Gamma(d_{\bar{A}})\Gamma(d-d_{\bar{A}})}. \qquad \text{(F.26)}$$

Now, in order to study the scaling of $f(d, d_{\bar{A}})$, we first take $d = \infty$, and expand around large but finite $d_{\bar{A}}$:

$$\text{TD}(\infty, d_{\bar{A}}) = 2\frac{d_{\bar{A}}^{d_{\bar{A}}-1}}{\Gamma(d_{\bar{A}})}e^{-d_{\bar{A}}} = 2(2\pi)^{-1/2}d_{\bar{A}}^{-1/2} + O\left(d_{\bar{A}}^{-3/2}\right). \tag{F.27}$$

In the last equality, we used the Stirling formula. We therefore see that the trace distance $\text{TD}(\infty, d_{\bar{A}}) \sim d_{\bar{A}}^{-1/2} \sim 2^{-n_{\bar{A}}/2}$ exponentially decays with the volume of the $\bar{A}$ subsystem.

# G  State distinguishability in time estimation code

In this appendix, we will derive the result for $F_H(\rho_R, \sigma_R) = \text{Tr}\sqrt{\rho_R}\sqrt{\sigma_R}$ discussed in (36). Recall that $\rho_R = \text{Tr}_B |\psi\rangle\langle\psi|$ and $\sigma_R = \text{Tr}_B |\xi\rangle\langle\xi|$. Instead of tuning the relative sizes of $R$ and $B$, we label the bipartition as $A\bar{A}$ such that $A \leq \bar{A}$. By calculating the $F_H(\rho_A, \sigma_A)$ and $F_H(\rho_{\bar{A}}, \sigma_{\bar{A}})$ and identifying $R$ as $A$ or $\bar{A}$ respectively depending on whether we are before or after the Page time, we can obtain (36) for any size of $R$ and $B$.

To simplify calculations, we again use the following simple flat-spectrum random pure state,

$$|\psi\rangle = \frac{1}{\sqrt{d_A}}\sum_{i=1}^{d_A} |\phi_i\rangle_A U_{ib} |b\rangle_{\bar{A}}. \tag{G.1}$$

The flat entanglement spectrum gives us that

$$\sqrt{\rho_A} \approx \sqrt{d_A}\rho_A, \qquad \sqrt{\rho_{\bar{A}}} \approx \sqrt{d_A}\rho_{\bar{A}}. \tag{G.2}$$

It is tricky to evaluate the operator-square-root for $\sigma_A$ and $\sigma_{\bar{A}}$. We shall go further by approximating that the entanglement spectrum of $\xi$ is also approximately flat. This is because a local Hamiltonian cannot drastically change the maximal non-local entanglement across the cut.

$$\sqrt{\sigma_A} \approx \sqrt{d_A}\sigma_A, \qquad \sqrt{\sigma_{\bar{A}}} \approx \sqrt{d_A}\sigma_{\bar{A}}. \tag{G.3}$$

We then have

$$
\begin{aligned}
F_H(\rho_A, \sigma_A) &\approx d_A \text{Tr}\rho_A\sigma_A \\
&= \frac{d_A}{d_{\bar{A}}^2\sigma^2}\sum_{i,j,k=1}^{d_A}\sum_{a,b,c=1}^{d_{\bar{A}}} U_{ib}\bar{U}_{jc} \langle\phi_k|_A \langle a|_{\bar{A}} \bar{H} |\phi_i\rangle_A |b\rangle_{\bar{A}} \langle\phi_j|_A \langle c|_{\bar{A}} \bar{H} |\phi_k\rangle_A |a\rangle_{\bar{A}}.
\end{aligned}
\tag{G.4}
$$

Using the Weingarten calculus,

$$\mathbb{E}_U U_{ib}\bar{U}_{jc} = \frac{1}{d_{\bar{A}}}\delta_{ij}\delta_{bc}, \tag{G.5}$$

yields

$$F_H(\rho_A, \sigma_A) \approx \frac{1}{d_A d_{\bar{A}}\sigma^2}\text{Tr}\bar{H}^2 \approx 1, \tag{G.6}$$

where

$$
\begin{aligned}
\text{Tr}\bar{H}^2 &= \text{Tr}(H^2 + \langle H\rangle_\psi^2 \mathbf{1} - 2H\langle H\rangle_\psi) \approx dn - (\text{Tr}H)^2/d = dn, \\
\sigma^2 &= \text{Tr}H^2/d - (\text{Tr}H/d)^2 \approx n.
\end{aligned}
\tag{G.7}
$$

We see that for the smaller subsystem $A$, the noisy codewords are indistinguishable.

Let us now evaluate $F_H(\rho_{\bar{A}}, \sigma_{\bar{A}})$,

$$F_H(\rho_{\bar{A}}, \sigma_{\bar{A}}) \approx d_A \text{Tr}\, \rho_{\bar{A}} \sigma_{\bar{A}} \tag{G.8}$$

$$= \frac{d_A}{d_A^2 \sigma^2} \sum_{i,j,k,l=1}^{d_A} \sum_{a,b,c,d=1}^{d_{\bar{A}}} U_{ib} \bar{U}_{jc} U_{kd} \bar{U}_{ka} \langle l|_A \langle a|_{\bar{A}} \bar{H} |\phi_i\rangle_A |b\rangle_{\bar{A}} \langle \phi_j|_A \langle c|_{\bar{A}} \bar{H} |l\rangle_A |d\rangle_{\bar{A}}\,.$$

Using the Weingarten calculus,

$$\mathbb{E}_U U_{ib} \bar{U}_{jc} U_{kd} \bar{U}_{ka} = \frac{1}{d_{\bar{A}}^2 - 1} (\delta_{ij} \delta_{bc} \delta_{kk} \delta_{da} + \delta_{ik} \delta_{ba} \delta_{kj} \delta_{dc})\,, \tag{G.9}$$

yields

$$F_H(\rho_{\bar{A}}, \sigma_{\bar{A}}) \approx \frac{d_A \text{Tr}\bar{H}^2 + \text{Tr}\bar{H}_A^2}{d_A d_{\bar{A}}^2 \sigma^2} \approx \frac{d_A dn + d d_{\bar{A}} n_A}{d d_{\bar{A}} n} \approx \frac{n_A}{n}\,, \tag{G.10}$$

where

$$\text{Tr}\bar{H}_{\bar{A}}^2 = \text{Tr}(H_{\bar{A}}^2 + d_{\bar{A}}^2 \langle H\rangle_\psi \mathbf{1}_{\bar{A}} - 2d_{\bar{A}} H_{\bar{A}} \langle H\rangle_\psi) \approx d d_{\bar{A}} n_A \approx \frac{n_A}{n}\,. \tag{G.11}$$

Hence, for the larger subsystem $\bar{A}$, the fidelity scales linearly with the system size being traced out.

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
