# Peer review of "Estimating time in quantum chaotic systems and black holes"

_SciPost Physics, doi:SciPost Phys. 19, 095 (2025)_

## Round 1 · Referee Report · Anonymous (Referee 1) · 2025-7-17

Strengths

  1. Introduce a new metric to probe quantum chaos and thermalization dynamics, based on estimating time from local / low-complexity measurements.
  2. The paper is largely clear and well-written.

Weaknesses

  1. Many of the claims are based on numerics only, and it is often unclear what results can be derived analytically for some specific models of dynamics, and what results were just inferred from numerics.
  2. Results for integrable systems are limited to non-interacting (free fermions) systems, and aren't representative of generic integrable systems.
  3. Sections 4,5,6 feel a bit disconnected from the rest of the paper, and could be either appendices or be integrated better with the rest of the narrative.

Report

In this paper, the authors introduce a new probe of universal quantum chaotic dynamics using an interesting idea of "time estimation", which they quantify using a Quantum Fisher Information metric. While the Quantum Fisher Information has been used in this context before, the authors' perspective is new and quite illuminating. They provide a universal picture of how this quantity behaves in generic chaotic quantum systems, largely based on numerics, and also relating it to evaporating black holes.

I think this is a very good paper, that can be published in Scipost Physics. I have a few suggestions listed below that I believe could improve the paper before publication.

Requested changes

  1. While the paper is well-written overall, I felt that sections 4-6 were a bit disconnected from the main narrative. Perhaps some of the content of these sections could be integrated in the rest of the text in a better way. For example, I think section 5 (which provides a practical implementation of the "estimating time" idea) could come a lot earlier to illustrate the authors' main idea.

  2. The authors argue that their quantity is useful to distinguish integrable and chaotic behavior. However, that claim is based on taking a non-interacting free fermion chain as a model of "integrability". This can be very misleading, as free fermion systems are very much not representative of interacting integrable systems, which can have very different behavior in quantities like OTOCs. I would ask the authors to either change the terminology from "integrable" to "free fermions", or add numerics on interacting integrable systems.

  3. Are all universal results on the scaling of the subsystem QFI inferred from numerical observations? Some of the results follow from the Brownian dynamics model described in an appendix for example. It might be useful to clarify which result (if any) can be established analytically (even for some toy models), and which results are purely based on small system numerics.

Recommendation

Ask for minor revision

---

## Round 1 · Referee Report · Anonymous (Referee 2) · 2025-7-25

Strengths

1. Framing late-time dynamics as a time-estimation task quantified through the quantum/classical Fisher information (QFI/CFI) is original and insightful.
2. The dependence of the time evolution and late-time scaling laws of the QFI on subsystem size and on whether the dynamics are chaotic or integrable is clearly exposed.
3. The combination of spin chains, Lindbladians, random Schmidt-basis states, and Brownian GUE toy models provides rich and convincing interpretations of the observed phenomena.
4. The contrast drawn between QFI and CFI, and the experiment on estimating time are timely and well motivated.

Weaknesses

1. It is sometimes unclear which statements are empirically observed in finite-size spin chains, which follow from toy models/Haar averages, and which are rigorously derived (often only in appendices under assumptions such as local and traceless Hamiltonians, and negligible boundary contributions). This obscures the degree of universality and generality beyond 1D, low-rank spin chains.
2. The black-hole section is under-specified and partially unjustified.
a. The gravitational setup is not fully stated: the gravity-plus-matter theory, spacetime asymptotics, the preparation and the age of the black hole should be specified.
b. Eq. (21) appears to be an expectation inferred from Eq. (20), yet it is used as the key step in deriving Eq. (26).
c. It is unclear why the volume of the photon gas should be proportional to the evaporation time in Eq. (24). Instead, the photon energy in Eq. (22) should be tied to the energy lost by the black hole.
d. Black hole evaporation is a nonequilibrium process. The manuscript seems to model the evaporation during a short window δt ≪ δt_evap as nearly equilibrium. How can this be justified or checked?

Report

The authors propose a new time-estimation task based on measurements on subsystems and relate the corresponding uncertainty to the Fisher information (quantum or classical, depending on the measurement basis). They primarily compute these Fisher informations numerically in chaotic and integrable spin chains, construct effective models to explain the behavior, and extract universal scaling laws. In chaotic models, the QFI of a sufficiently large subsystem shows sharply different behavior before and after the Page time.
The work is conceptually strong and potentially impactful for both quantum many-body physics and black-hole information. I find it suitable for publication in SciPost, subject to the requested changes below.
The authors are also asked to address two specific questions:
1. In the analytical derivations, the boundary contribution is assumed negligible compared to the bulk, |∂S| ≪ n_S. Does this imply that the results do not extend to all-to-all Hamiltonians, such as the SYK model?
2. I agree that entanglement entropy is insensitive to rotations of the support of the reduced density matrix on a subsystem. If I understand correctly, these rotations only set the overall energy-density-squared scale of the QFI, while the universal terms depend solely on n_A and n_{\bar A} and are thus captured by the entanglement entropy. Consequently, once the dynamics generate a 2-design at late times, the precise form of the Hamiltonian becomes unimportant. Is this why random pure states and the Brownian GUE toy model—both independent of microscopic Hamiltonian—work so well for computing the QFI?

Requested changes

1. In connection with Weakness 1, clearly distinguish results obtained numerically from those derived analytically, and highlight all assumptions used in the derivations in the main text.
2. In connection with Weakness 2, spell out the gravitational setup in detail and provide further justification for Eqs. (21)–(25).
3. Correct the typo “chaotiic” on line 372 to “chaotic”.

Recommendation

Ask for minor revision

  • validity: high
  • significance: high
  • originality: top
  • clarity: good
  • formatting: excellent
  • grammar: excellent

Author:  Shreya Vardhan  on 2025-08-18  [id 5737]

(in reply to Report 2 on 2025-07-25)

In response to the specific questions in this report (see the resubmission for a full list of changes in response to the other comments):

  1. "In the analytical derivations, the boundary contribution is assumed negligible compared to the bulk, $|\delta S| \ll n_S$. Does this imply that the results do not extend to all-to-all Hamiltonians, such as the SYK model?"

It is true that the result in equation (22) does not apply to the SYK model. See the comment added in footnote 7 on page 16 of the resubmitted version about how one can derive the analogous result for SYK.

  1. "I agree that entanglement entropy is insensitive to rotations of the support of the reduced density matrix on a subsystem. If I understand correctly, these rotations only set the overall energy-density-squared scale of the QFI, while the universal terms depend solely on $n_A$ and $n_{\bar A}$ and are thus captured by the entanglement entropy."

It is not true that the final result, say in equation (22), can be expressed in terms of entanglement entropies of subsystems. Note that the entanglement entropies for random pure states are always proportional to $\text{min}(n_A, n_{\bar A})$, and never provide information about the size of the larger of the two subsystems. In contrast, the second line of (22) is proportional to the size of the larger subsystem. This is a concrete way of seeing that the QFI has access to information about the system beyond that captured by the entanglement entropy.

"Consequently, once the dynamics generate a 2-design at late times, the precise form of the Hamiltonian becomes unimportant."

It is not immediately clear whether the relevant time scale for saturation of the QFI is the one where the time-evolution unitary forms a 2-design: note that the $U$ and $V$ that appear in equation (21), whose 2-design properties used, are not not quite the same as the time-evolution unitary. Instead, these are the unitaries that relate some simple reference vectors in $A$ and $\bar A$ to the Schmidt vectors of the time-evolved state in $A$ and $\bar A$.

"Is this why random pure states and the Brownian GUE toy model—both independent of microscopic Hamiltonian—work so well for computing the QFI?"

We expect that the reason why random pure states work well for estimating the late-time QFI (and in particular, match well with the spin chain numerics) is that in the spin chain system, we have chosen an ensemble of initial random product states, which are expected to equilibrate to infinite temperature and resemble random pure states at late times.

---

## Round 2 · Referee Report · Anonymous (Referee 2) · 2025-8-31

Report

The authors have provided a detailed response that fully addresses my two questions and implements the suggested changes. I am satisfied with the revision and recommend its publication in SciPost.

Recommendation

Publish (surpasses expectations and criteria for this Journal; among top 10%)

---

## Round 2 · Author Response

Thank you for facilitating the feedback from the referees. We are grateful to the referees for their careful reading and helpful comments on the manuscript, and their positive evaluation of the strengths of the paper. Below, we address specific weakness and questions raised by the referees and list the changes made in response. See "list of changes," as well as "reply to referee 2" for the two specific questions mentioned in that report.

---

## Round 2 · List of Changes

REPLY TO REFEREE 1:

  1. "While the paper is well-written overall, I felt that sections 4-6 were a bit disconnected from the main narrative. Perhaps some of the content of these sections could be integrated in the rest of the text in a better way. For example, I think section 5 (which provides a practical implementation of the "estimating time" idea) could come a lot earlier to illustrate the authors' main idea."

(i) We agree with the referee's point that the maximum likelihood estimate (MLE), which provides the practical implementation of the idea of estimating time, should have been introduced at an earlier point in the text instead of in Section 5. Accordingly, we have added the definition of the maximum likelihood estimate in Sec.~2 in the added red text on page 7. We then refer to this definition in Sections 4 and 5 where we need to refer to the MLE (we have also highlighted the related changes in those sections in red text on page 21 and page 22).

(ii) We have added some text, highlighted in red, at the beginning of Sec. 4. to more clearly summarize the result of the previous section. This added text also makes it clear that we are now returning to the setup of general unitary chaotic quantum many-body systems, as opposed to the specific black hole example of the previous subsection.

(iii) To better integrate Section 6, we have made it clearer in the first paragraph of Sec. 6 that we are now further interpreting a key result from Section 3 in terms of error correction. We have also anticipated the discussion of Sec. 6 in the general summary of properties of the subsystem QFI at the beginning of Sec. 3, in the highlighted red text on page 12.

  1. "The authors argue that their quantity is useful to distinguish integrable and chaotic behavior. However, that claim is based on taking a non-interacting free fermion chain as a model of "integrability". This can be very misleading, as free fermion systems are very much not representative of interacting integrable systems, which can have very different behavior in quantities like OTOCs. I would ask the authors to either change the terminology from "integrable" to "free fermions", or add numerics on interacting integrable systems."

We thank the referee for this suggestion. We have now checked the behavior of the subsystem QFI in an example of an interacting integrable system, the Heisenberg XXZ spin chain. This case is discussed the new Appendix~D, which we refer to at the end of Section 3.1 in the main text. The behavior in this interacting integrable system turns out to be similar to that in chaotic systems. We have therefore added the caveat about ``free fermion integrable systems'' in the main text in the following places: (1) below point 2 on page 4, (2) in point 1 on page 10, (3) at the end of Sec. 3.1 on page 14, (4) in the caption of Fig. 6, (5) at the beginning of Sec. 7, and (6) at the end of section 7.1.

  1. "Are all universal results on the scaling of the subsystem QFI inferred from numerical observations? Some of the results follow from the Brownian dynamics model described in an appendix for example. It might be useful to clarify which result (if any) can be established analytically (even for some toy models), and which results are purely based on small system numerics.''

We agree with the referee that we should more clearly distinguish between results that are obtained numerically and those that are found analytically. For the scaling of the subsystem QFI (and the subsystem CFI), the dependence of the {\it late time saturation values} on system sizes in all cases can be obtained from analytic calculations in random pure states, and these agree with the numerical simulations in chaotic spin chains. For the intermediate time values, the evidence is mostly numerical, other than the Brownian GUE toy model discussed in Appendix C. In the new version, we have tried to make these points clearer from the beginning by making the following changes:

(i) We have better clarified which results are analytic and which ones are numerical in the added red text on page 4 in the introduction.

(ii) We have also added clarifications along these lines in the added red text on page 10 in the summary of results at the beginning of Section 3. In particular, we have moved the summary of the Brownian GUE result of Appendix C to these numbered points, instead of mentioning it at the end of the introductory part of Section 3.

REPLY TO REFEREE 2:

  1. " It is sometimes unclear which statements are empirically observed in finite-size spin chains, which follow from toy models/Haar averages, and which are rigorously derived (often only in appendices under assumptions such as local and traceless Hamiltonians, and negligible boundary contributions). This obscures the degree of universality and generality beyond 1D, low-rank spin chains."

"In connection with Weakness 1, clearly distinguish results obtained numerically from those derived analytically, and highlight all assumptions used in the derivations in the main text."

This suggestion is similar to comment 3 of Referee 1; please see the changes listed above in response to that comment.

Regarding the comment about clarifying all assumptions in the main text:

(i) We have further emphasized our assumption of geometric locality above equation (22), and added a footnote about non-local cases on page 16.

(ii) We have edited equation (22) so that it no longer assumes that the Hamiltonian is traceless. The new expressions are proportional to the energy variance in the infinite temperature state instead of simply being proportional to $\Tr[H^2]$. We have also made edits in Appendix B, highlighted in red, to get rid of the traceless assumption in the derivation of equation (22).

  1. "The black-hole section is under-specified and partially unjustified.``In connection with Weakness 2, spell out the gravitational setup in detail and provide further justification for Eqs. (21)–(25)."

We are grateful to the referee for this set of suggestions, which have helped significantly improve our discussion of black hole evaporation. We have made edits throughout section 3.4 in red text to take into account more realistic aspects of the evaporation process.

See below for specific edits addressing the referee's comments:

"(a) The gravitational setup is not fully stated: the gravity-plus-matter theory, spacetime asymptotics, the preparation and the age of the black hole should be specified."

--> We thank the referee for pointing out that we should have explicitly stated these assumptions. All of our formulas for Hawking temperature and other assumptions rely on using the same setup as in Hawking's 1975-1976 papers (Refs. 13 and 14) deriving Hawking radiation: we are in 3+1-dimensional asymptotically flat space, and the black hole is formed from spherically symmetric collapse of an initial pure state, which is the vacuum state at past infinity. We have clarified this setup in the introduction, on page 5, as well as in the red text in Sec. 3.4 above equation (24). For the matter content, we can either assume a massless scalar field as in Hawking's calculation, or gravitons and photons like in Page's more realistic estimates in Ref. 16; the scaling of the results with $G_N$ and $M$ is unaffected by this choice. We have also clarified this point around equation (24).

"(b.) Eq. (23) appears to be an expectation inferred from Eq. (22), yet it is used as the key step in deriving Eq. (35)."

--> In order to apply our results to the evaporating black hole setup, it is necessary to use a finite-temperature version of the random pure state result in equation (22), and equation (23) is the most natural generalization of (22) to finite temperature. With the edits to equation (22) in the new version to include Hamiltonians with non-zero trace, we hope that equation (23) seems more intuitive. The logic here is similar to Page's assumption in Ref. 16 that the infinite-temperature Hilbert space dimensions that appear in the random pure state calculation of Ref. 11 can be generalized in terms of finite-temperature thermodynamic entropies and applied to the black hole setup.

"(c) It is unclear why the volume of the photon gas should be proportional to the evaporation time in Eq. (24). Instead, the photon energy in Eq. (22) should be tied to the energy lost by the black hole."

--> We thank the referee for making this important point. The assumption that the volume of the photon gas is proportional to the time is sometimes used to heuristically justify the initial linear increase with time of the entanglement entropy of the radiation, like in Ref. 49, but it does violate energy conservation. This assumption should not be used in our estimate for $F_A(t)$, where the role of energy conservation is important. We have corrected the estimate for the energy variance (as well as our model for the state of the radiation, as we explain further in the next point) in the new version of Section 3.4. The essential feature of the result for $F_A(t)$ which we wanted to emphasize, that it is proportional to $1/G_N$ after the Page time, is unchanged by this correction.

"d. Black hole evaporation is a nonequilibrium process. The manuscript seems to model the evaporation during a short window $\delta t \ll {\delta t}_{\rm evap}$ as nearly equilibrium. How can this be justified or checked?"

--> One concrete way to see that the state of the radiation does not change in Hawking's calculation on time scales shorter than ${\delta t}_{\rm evap}$ is to note that the rate of emission of particles into the radiation, $dN/dt$, is proportional $ \frac{1}{G_N M}$, so that the time between the emission of two particles is on average $t_{evap}=\frac{1}{G_N M}$. This emission rate was calculated by Page in Ref. 15 (see in particular section IV of this reference), which we have added to the new version of our draft and referred to on page 5 in the introduction. Between the emission of two particles, neither the temperature nor the number of degrees of freedom in the radiation changes from the semiclassical gravity calculation, so the state appears not to change at all. This is a useful point to clarify, and we have added it in the red text on page 5 as well as in Sec. 3.4.

--> More broadly, even though black hole evaporation is a nonequilibrium process, the assumption that the state can be treated as a thermalized state corresponding to the Hawking temperature $T_H$ on time scales much shorter than $t_{\rm evap}$ is standard in the literature. For example, Hawking uses this as a central assumption in Refs. 13 and 14 while modeling the evaporation process with a series of {\it time-independent} Schwarzschild solutions with gradually decreasing masses, which corresponds to a series of thermal equilibriums at gradually increasing temperature. Similarly, Page applied results from random pure states (which can be seen as equilibrated pure states at infinite temperature) in order to deduce entanglement entropies of the black hole and radiation. Again, in order to derive the non-trivial time-depedence, Page assumed that the temperature of the equilibrium state, and the number of degrees of freedom in the black hole and radiation, was gradually changing during the evaporation process. But for each temperature and each stage in the evaporation process, he made use of the equilibrium result from random pure states. The same assumption underlies our use of equation (23) to derive equation (35).

--> One point which did need to be corrected in our analysis along these lines is that instead of assuming that the state of radiation is $e^{-\beta H_R}/ Z_{\beta}$ at the time when the black hole temperature is $1/\beta$, we should instead have a tensor product over canonical ensembles for particles emitted at different times, which have different temperatures. The corrected model for the equilibrium state of the radiation and its consequences are discussed around equation (27).

  1. ``Correct the typo “chaotiic” on line 372 to “chaotic”."

We have corrected the typo on page 12.

Please see the "reply to referee 2" for responses to the two additional questions mentioned in this report.

---

## Editorial Decision

published